# Compositional Generalization Requires Linear, Orthogonal Representations in Vision Embedding Models

Arnas Uselis [1]  Andrea Dittadi [2 3]  Seong Joon Oh [1]

## Abstract

Compositional generalization, the ability to recognize familiar parts in novel contexts, is a defining property of intelligent systems, yet modern models, despite massive training sets, see only a tiny fraction of the combinatorial input space. We ask what structure representations must have to support generalization to unseen combinations. We formalize three desiderata (divisibility, transferability, stability) and show they impose necessary geometric constraints under standard training: representations must decompose linearly into per-concept components, orthogonal across concepts. This grounds the Linear Representation Hypothesis as a necessary consequence of compositional generalization, and yields dimension bounds linking the number of composable concepts to embedding geometry. Empirically, across CLIP, SigLIP, and DINO, we find partial linear factorization with low-rank near-orthogonal per-concept factors, and the degree of this structure correlates with compositional generalization on unseen combinations. As models continue to scale, these conditions predict the geometry they may converge to. Code: https://github.com/oshapio/necessary-compositionality.

## 1. Introduction

Modern vision systems are trained on a tiny, biased subset of a combinatorial space of visual concepts, like objects, attributes, relations in different contexts. Despite this, we expect them to perform well in the wild on novel recombinations of familiar concepts, an expectation tied to the view that systematic generalization, the ability to recombine learned constituents, is a hallmark of intelligence (Fodor & Pylyshyn, 1988). Yet a large body of empirical work shows

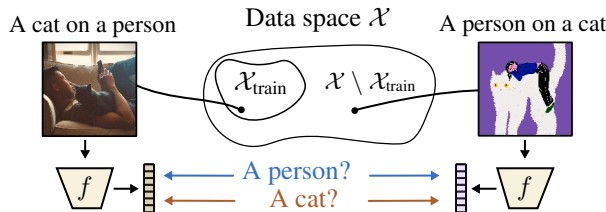

Figure 1. **What enables compositional generalization in vision-language embedding models?** Training data contain common configurations (left: a cat on a person) but lack rare ones (right: a person on a cat). Yet the same text-based queries, e.g. "A photo of a person", must work on both, even when the latter was never seen during training. We investigate what properties encoder $f$ must satisfy for such transfer to succeed.

that even high-performing neural models often struggle with systematicity when train/test combinations mismatch (Lake & Baroni, 2018; Keysers et al., 2020; Hupkes et al., 2022; Uselis et al., 2025). At the same time, large vision-language models such as CLIP (Radford et al., 2021) and its variants are trained on web-scale datasets (e.g., LAION-400M (Schuhmann et al., 2021a)) and achieve impressive zero-shot transfer on many tasks (Radford et al., 2021; Zhai et al., 2022). However, these systems often fail when test images contain unusual combinations of familiar concepts (Xu et al., 2022; Bao et al., 2024; Thrush et al., 2022; Abbasi et al., 2024; Yuksekgonul et al., 2023; Ma et al., 2023). Fig. 1 illustrates this tension for CLIP-like architectures: an image encoder $f$ produces embeddings on which linear classifiers (e.g. from the text encoder) classify concepts, but the training data $\mathcal{X}_{\text{train}}$ covers only common compositions from the full data space $\mathcal{X}$, while models must answer the same queries correctly on rare compositions from $\mathcal{X} \setminus \mathcal{X}_{\text{train}}$. Given how rarely, if at all, such compositions appear in training, we ask: *assuming that compositional generalization succeeds, what properties must the representations have to accommodate it?*

We argue for *non-negotiable, model-agnostic* properties that any neural-network-based system claiming compositional generalization must satisfy. We state three desiderata: *divisibility*, *transferability*, and *stability*. These desiderata formalize that (i) all parts of an input should be accessible to a simple readout; (ii) readouts trained on a tiny but diverse subset should transfer to unseen combinations; and

[1]Tübingen AI Center, University of Tübingen [2]Helmholtz Munich [3]Technical University of Munich. Correspondence to: Arnas Uselis <arnas.uselis@uni-tuebingen.de>.

*Proceedings of the $43^{rd}$ International Conference on Machine Learning*, Seoul, South Korea. PMLR 306, 2026. Copyright 2026 by the author(s).

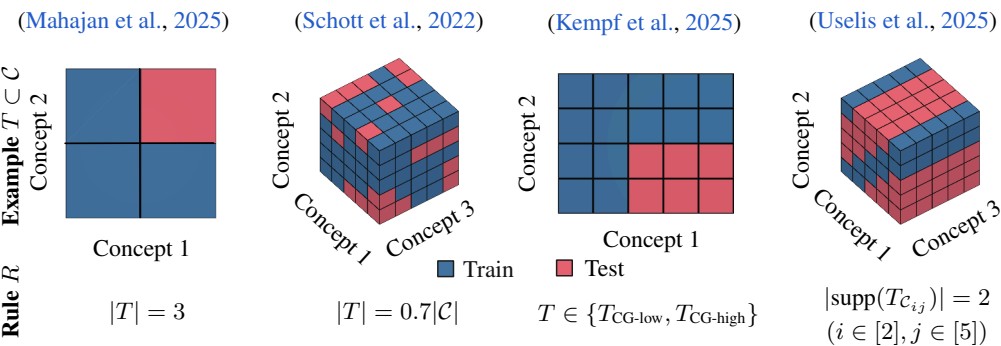

*Figure 2.* **Interpreting previous works' sampling designs $T$ and validity rules $R$.** Each panel shows a concept space as a grid (2D or 3D), with blue cells denoting training combinations and red cells denoting held-out test combinations.

(iii) training on any valid subset should yield robust generalization. Our scope is the common setting where predictions are linear in the embedding $f$: CLIP-style zero-shot classifiers, linear probing, and cases where a fixed non-linear head is folded into the encoder.

Our key finding is that these desiderata *necessitate* a specific geometry: *linear factorization* with *near-orthogonal* concept directions. This establishes what any model *must* achieve to compositionally generalize under standard training, providing a concrete target for future design. Moreover, it offers theoretical grounding for the *Linear Representation Hypothesis* (Mikolov et al., 2013; Elhage et al., 2022) – the linear structure widely observed in neural representations is a *necessary consequence* of compositional generalization.

Our contributions are: (1) **Defining desiderata.** We define three desiderata: *divisibility*, *transferability*, *stability*, and formalize compositional generalization in their terms. (2) **Structural necessity.** Under gradient descent with the cross-entropy loss, these desiderata imply *linear factorization*: embeddings decompose into per-concept sums with orthogonal difference directions. (3) **Empirical grounding.** Across CLIP and SigLIP families, we find strong evidence of factorization, near-orthogonality, low-rank per-concept geometry, and correlation with compositional generalization accuracy, suggesting that the current state-of-the-art vision models are converging to the necessary geometric properties.

## 2. Related work

**Compositional generalization.** Prior work establishes *sufficient* conditions for compositional generalization under specific assumptions on data-generating processes or representations (Wiedemer et al., 2023; Mahajan et al., 2025; Uselis et al., 2025). In contrast, we study *necessary* conditions: we ask what must be true of a model's embeddings *if* it transfers from a restricted subset to the full space under our desiderata.

**Geometry of learned representations.** Empirical studies document linear subspaces in VLMs (Trager et al., 2023),

the Linear Representation Hypothesis in LLMs (Park et al., 2023), and near-orthogonal feature encodings (Elhage et al., 2022). In parallel to our work, (Fel et al., 2025) map task-relevant concept families in DINOv2 and propose a Minkowski-sum hypothesis for token space, highlighting that strong geometric structure may arise from architecture as well as from downstream demands. In contrast, we show that under practice-driven desiderata and standard training, linearity and orthogonality are *necessary*, not merely observed.

**Disentangled and object-centric representations.** This line of work proposes desiderata and training schemes, with mixed evidence linking these to compositional generalization (Eastwood & Williams, 2018; Montero et al., 2021; Dittadi et al., 2022). We ask the complementary question: if a model *does* generalize compositionally, what must its embeddings satisfy? See Appendix A for a detailed discussion.

## 3. Setup: A framework for compositionality

We begin by detailing key desiderata for embedding models that contend to be compositionally generalizing, motivated from a practical perspective: models must distinguish any concept combination, transfer from a subset of the concept space to the full space, and do so robustly across valid subsets.

### 3.1. Setup: concept spaces and data collection process

We interpret the world as a product of concepts: any input $x_c \in \mathcal{X}$ (e.g., an image) has an associated tuple of concepts $c \in \mathcal{C}$, describing its constituent parts and properties. This is a reasonable way to describe a large portion of the world. For example, current large-scale datasets (e.g., image-caption pairs) provide noisy natural-language descriptions that can be decomposed into *discrete* concept values. Clearly, a single concept tuple cannot capture all aspects of the world, e.g. how attributes bind to objects or how different objects relate spatially. Still, an intelligent system should at least be able to tell apart basic concepts (such as objects and their attributes), even without modeling their relations. In other

words, concept spaces may not capture the full compositional structure of the world, but any model of the world must involve them in some form. Importantly, we do not assume *how* the concept values are distributed (e.g. being independent), only *what* they represent.

**Definition 1** (Concept space). Suppose we have $k$ concepts, where concept $i$ takes $n_i$ possible values. Let $C_i = [n_i]$ for $i \in [k]$. The *concept space* is the Cartesian product

$$C = C_1 \times C_2 \times \cdots \times C_k. \tag{1}$$

In most of the paper we use $n_i = n$ for all $i$, so $C = [n]^k$ and $|C| = n^k$. We index inputs by concept tuples: for each $c \in C$ we assume an associated $x_c \in \mathcal{X}$ (e.g., a natural image) realizing $c$.

Data-related components for compositional generalization involve three notions: (1) the total variation of the data, (2) the concepts we aim to learn and expect the model to capture, and (3) the data that is actually collectible. We capture (1) by the concept space $C$ (Definition 1). For (2), the targets that we aim to capture are all concepts and their values in $C$, reflecting that foundation models attempt to align with all present concepts.[1] For (3), we formalize collectability constraints through a validity class that specifies which training supports are valid, indicating which concept combinations may appear in training–taking into account cases that many combinations are not collectible (e.g. "a pink cat on the moon" isn't common). We formalize this below.

**Considering data collection.** We are interested in models that support efficient compositional generalization from a subset of the concept space. To formalize this notion, we specify a validity class $\mathcal{T} \subseteq 2^C$ of valid training sets, where $2^C$ denotes the power set of $C$, and a validity rule $R : 2^C \to \{0, 1\}$ that specifies whether a given training set is valid. This setup captures the natural question of which training sets we use and for which we expect generalization.

**Definition 2** (Training support, validity class, and training dataset). Let $C$ be the concept space. A *training support* is any subset $T \subseteq C$. *Validity class* is a collection $\mathcal{T} \subseteq 2^C$ whose members are called *valid training sets*. The class $\mathcal{T}$ specifies which training sets are observable. Validity class $\mathcal{T}$ is specified by a *validity rule* $R : 2^C \to \{0, 1\}$ through $\mathcal{T} = \{T \subseteq C : R(T) = 1\}$. A *training dataset* for a training set $T$ is $D_T = \{(x_c, c) : c \in T\}$.

We note that there are many validity rules used in practice. For example, if we can collect any subset of size $N < |C|$, then $R(T) = 1$ whenever $|T| = N$. Fig. 2 illustrates common choices: Mahajan et al. (2025) use training supports

---

[1] More generally, the world may involve additional factors beyond the concepts represented in $C$. Our framework does not require $C$ to be exhaustive: $C$ can be viewed as the subset of concepts we model explicitly, with any remaining variation treated as nuisance.

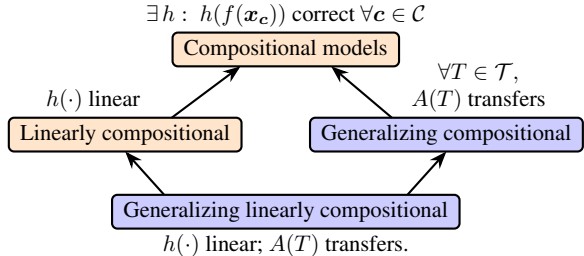

*Figure 3.* **Relationship between (generalizing) compositional models.** Divisibility (orange) and transferability (blue) requirements.

that cover every concept value; Schott et al. (2022) use random samples covering 70% of all combinations; Kempf et al. (2025) specify a small set of allowed supports; and Uselis et al. (2025) use supports whose joint marginals cover at least two values per concept. Note that these validity rules apply to concept supports rather than individual datapoints.

### 3.2. Compositional representations and models

Given the concept space and the training supports, we now make precise how we expect models to learn.

**Scope of models.** We study embedding models: these cover modern foundation models like CLIP and SigLIP (Tschannen et al., 2025; Zhai et al., 2023), supervised-learning models, self-supervised models like DINO (Caron et al., 2021; Siméoni et al., 2025). At inference the models we study are *non-contextual*: the representation of an input depends only on that input (no dependence on other test examples, prompts, or the batch). Formally, the encoder is a map $f : \mathcal{X} \to \mathcal{Z}$, with $z = f(x)$ (optionally $\ell_2$-normalized).

**Readout class (linear vs. non-linear).** Usually, encoders $f$ are associated with either a downstream or readout model $h$ that takes $z = f(x)$ and outputs per-concept logits $h(z) \in \mathbb{R}^{k \times n}$ using argmax classification rule (see Definition 3). This covers zero-shot use of text features as linear classifiers, standard linear probing, and the affine last layer in most neural classifiers. If $h$ is non-linear in a neural network, we absorb the layers preceding the linear layer $g$ into the encoder ($\tilde{f} = g \circ f$) and analyze the resulting affine layer. The definition below keeps the readout $h$ general to allow future extensions beyond linear heads, but all results in this paper consider the linear case, without such restrictions, a high-capacity readout could make any injective encoder appear compositional by memorization.

**Definition 3** ((Linearly) compositional model). An encoder $f : \mathcal{X} \to \mathcal{Z}$ is *compositional w.r.t.* $C$ if there exists $h : \mathcal{Z} \to \mathbb{R}^{k \times n}$ such that, for all $c \in C$ and all $i \in [k]$,

$$c_i = \underset{j \in [n]}{\arg\max}\, h(f(x_c))_{i,j}. \tag{2}$$

It is *linearly compositional* if $h$ can be taken affine $h(z) = Wz + b$. We refer to $h$ as the *readout*.

## 3.3. Compositional generalization and desiderata

Given the ingredients (concept space $\mathcal{C}$, encoder $f$, and training-support family $\mathcal{T}$), we now define a learning rule $A$ and state three desiderata for compositional generalization: *divisibility*, *transferability*, and *stability*. We emphasize that these desiderata are on the NN-based models that exhibit generalization, as defined below, not on the representations, as studied in disentangled representation learning.

**Considering training.** We view a learning algorithm as a map

$$A : D_T \mapsto h_T, \qquad h_T \in \mathcal{H} \subseteq \{h : \mathcal{Z} \to \mathbb{R}^{k \times n}\},$$

from a dataset supported on $T \subseteq \mathcal{C}$ to a readout in a chosen hypothesis class. In practice, $A$ is typically (stochastic) gradient descent on a cross-entropy or contrastive objective, covering contrastive vision-language encoders (e.g., CLIP, SigLIP), standard supervised classifiers, and linear probes on self-supervised vision encoders like DINO.

**Desiderata for compositional generalization.** Suppose we train a downstream readout $h_T = A(D_T)$ on some $T \in \mathcal{T}$. What should $h_T$ satisfy? We argue for three practically-motivated properties.

First, every combination of concept values should be *classifiable* by the readout: for any $\boldsymbol{c} \in \mathcal{C}$, the corresponding region of the representation space of $f$ is nonempty: there exists at least one $\boldsymbol{z}$ that $h_T$ assigns the concept values $\boldsymbol{c}$. Otherwise, generalization to the full grid is impossible. We refer to this property as *Divisibility*.

**Desideratum 1** (Divisibility). A compositional model (Definition 3) can only exist if the readout has sufficient capacity to represent every concept combination. That is, for a readout $h : \mathcal{Z} \to \mathbb{R}^{k \times n}$, every concept tuple must have a nonempty decision region:

$$\forall \boldsymbol{c} \in \mathcal{C} : \bigcap_{i=1}^{k} \mathcal{R}_{i, c_i}(h) \neq \varnothing, \tag{3}$$
$$\text{where } \mathcal{R}_{ij}(h) = \{\boldsymbol{z} \in \mathcal{Z} : \arg\max_{j' \in [n]} h(\boldsymbol{z})_{i,j'} = j\}.$$

Divisibility is necessary but not sufficient: it guarantees that the space is divisible, but does not imply that the readout will be correct. We therefore ask that, for every training set, the learned readout transfers to the full grid; we call this *Transferability*.

**Desideratum 2** (Transferability). For every $T \in \mathcal{T}$, the trained readout $h_T = A(D_T)$ correctly classifies all possible combinations of the concept space:

$$\forall \boldsymbol{c} \in \mathcal{C}, \, \forall i \in [k] : \quad \underset{j \in [n]}{\arg\max} \, h_T\big(f(\boldsymbol{x_c})\big)_{i,j} = c_i. \tag{4}$$

Note that Transferability implies Divisibility. We state Divisibility explicitly because it highlights a capacity requirement: the embedding space must be able to represent all concept combinations.

Third, consider readouts learned from different valid supports $T \in \mathcal{T}$. Divisibility and Transferability do not say anything about the behavior of the classification decisions. Intuitively: if an input depicts a "cat", retraining on another valid support should not change the preference towards "dog". We refer to this as *Stability*.

**Desideratum 3** (Stability). For any $T, T' \in \mathcal{T}$, any point $\boldsymbol{x} \in \mathcal{X}$, and any $i \in [k]$, the per-concept posteriors agree across supports:

$$p_i^{(T)}(j \mid f(\boldsymbol{x})) = \frac{\exp(h_T(f(\boldsymbol{x}))_{i,j})}{\sum_{k=1}^{n} \exp(h_T(f(\boldsymbol{x}))_{i,k})},$$
$$p_i^{(T)}(\cdot \mid f(\boldsymbol{x})) = p_i^{(T')}(\cdot \mid f(\boldsymbol{x})). \tag{5}$$

We view Stability as an idealization: once the training support is sufficiently informative, retraining on any other valid support should not change the model's per-concept predictions (and ideally its calibrated posteriors). In practice, this may only hold approximately; relaxing Stability to allow small distributional deviations across supports is a natural direction for future work. We discuss the role of Stability and what can fail without it in Appendix F.

**Defining compositional generalization.** We now tie the ingredients into a single tuple $\Pi = (f, \mathcal{H}, A, \mathcal{T})$, which we use as the object that specifies the entire compositional-generalization setup: the encoder, the readout class, the learning rule, and the family of valid training supports. We specify compositional generalization as a process of learning readouts that generalize over *all* $T \in \mathcal{T}$ and satisfy Desiderata 1–3.

**Definition 4** (Compositional generalization). $\Pi = (f, \mathcal{H}, A, \mathcal{T})$ *exhibits compositional generalization* if, for every $T \in \mathcal{T}$ with $h_T = A(D_T)$, Divisibility (Desideratum 1) and Transferability (Desideratum 2) hold on the full grid, and the posteriors are Stable across valid retrainings (Desideratum 3) for all pairs $T, T' \in \mathcal{T}$. We say that $\Pi$ exhibits *linear compositional generalization* when the readout hypothesis class is linear.

We illustrate the relationship between (linear) models and their compositional counterparts in Fig. 3. In practice one could consider relaxed or average-case variants; however, we here are interested in "ideal" representations that support compositional generalization under any data sample.

## 3.4. Instantiating the framework with CLIP

We instantiate the framework in the dual-encoder, vision-language setting in the style of CLIP models: images and texts are embedded into a shared space and trained to align, with captions acting as noisy descriptions of concept tuples.

**Encoders.** Let $f : \mathcal{X} \to \mathcal{Z}$ be the image encoder and $g : \mathcal{Y} \to \mathcal{Z}$ the text encoder. At inference both are typically $\ell_2$-normalized so that inner products are cosine similarities: $\|f(\boldsymbol{x})\| = \|g(\boldsymbol{y})\| = 1$.

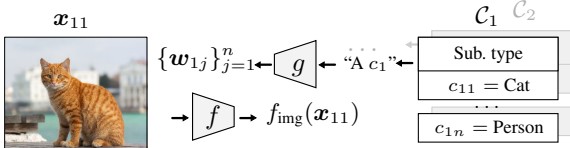

*Figure 4.* **Instantiating the framework with CLIP-like embedding models for analysis.**

**Prompts as linear probes.** Zero-shot classification uses text features as linear classifiers. For each concept $i \in [k]$ and value $j \in [n]$, we can choose a prompt $p_{i,j}$ (e.g., "a photo of a cat") and define a probe vector $\boldsymbol{w}_{i,j} := g(p_{i,j}) \in \mathcal{Z}$. Stacking these gives a readout

$$h(\boldsymbol{z}) = \left[\ \boldsymbol{w}_{i,j}^{\intercal}\boldsymbol{z}\ \right]_{i,j} \in \mathbb{R}^{k \times n}.$$

Here $f$ is the representation model, while $h$ is a linear readout whose weights come from the text encoder. Training in CLIP-like models can be viewed as learning a readout model where the *same* set of text-derived probes serves across many images; prompts often mention only parts of an image, so the system is implicitly asked to recognize objects and attributes regardless of which other concepts co-occur. We illustrate this process in Fig. 4; for a high-level schematic, see Fig. 1 in the introduction.

**The question we study.** Given a concept space $\mathcal{C}$, what structure must $\boldsymbol{z} = f(\boldsymbol{x}_{\boldsymbol{c}})$ have so that a single set of probes $\{\boldsymbol{w}_{i,j}\}$ (whether fixed by $g$ or learned as linear probes) satisfies our desiderata (Desiderata 1–3) on the full $\mathcal{C}$? In other words, what constraints does zero-shot, probe-based classification place on the geometry of image representations in order to support compositional generalization?

# 4. Implications of compositionality on representations

We now ask what our desiderata imply for representations in common training regimes, both as necessary constraints and as sufficient conditions, and what minimum embedding dimension they require.

## 4.1. Geometry of $f$ under common training settings

We instantiate $A$ as gradient descent on the binary cross-entropy (logistic) loss. As in Section 3.4, the readout $h$ is linear in the embedding $\boldsymbol{z}_{\boldsymbol{c}} = f(\boldsymbol{x}_{\boldsymbol{c}})$ (using either text-encoder-derived probes or learned linear heads). Under these assumptions, the representation space $\mathcal{Z}$ must exhibit both linearity and cross-concept orthogonality. This conclusion holds under at least two validity regimes: (1) when more than half of all possible combinations are observed, $|T| = 2^{k-1} + 1$, and (2) when only a small, carefully chosen set of datapoints is observed, $|T| = 1 + k$.

**Proposition 1** (Compositional generalization implies linear factorization). Let $\Pi = (f, \mathcal{H}, A, \mathcal{T})$ be the tuple instantiated in Definition 4, with linear heads $\mathcal{H}$ and $A$

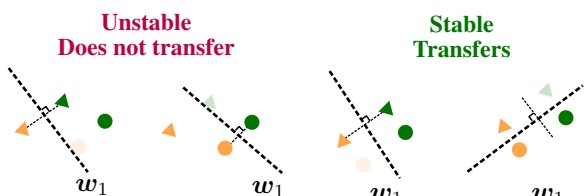

*Figure 5.* **Stable and unstable examples of feature representations.** The left panel shows an unstable configuration, where depending on the sample, the readout either does not transfer or does so unstably. The right panel shows a stable configuration.

given by GD+CE. Suppose the validity rule is one of: (i) $R(T) = 1$ iff $|T| = 2^{k-1} + 1$ (random sampling, more than half the grid), or (ii) $R(T) = 1$ iff $|T| = 1 + k$ and $T$ consists of an anchor point together with $k$ points each differing from it in exactly one concept. Assume Desiderata 1–3 are satisfied. Then, under the binary grid $\mathcal{C}_i = \{0,1\}$ with $\mathcal{Z} = \{\boldsymbol{z}_{\boldsymbol{c}} : \boldsymbol{c} \in [2]^k\} \subset \mathbb{R}^d$, there exist $\{\boldsymbol{u}_{i,0}, \boldsymbol{u}_{i,1} \in \mathbb{R}^d\}_{i=1}^k$ such that for every $\boldsymbol{c} \in [2]^k$ the following holds:

1. *(Linearity)* $\boldsymbol{z}_{\boldsymbol{c}} = \sum_{i=1}^k \boldsymbol{u}_{i,c_i}$.

2. *(Cross-concept orthogonality)* $(\boldsymbol{u}_{i,1} - \boldsymbol{u}_{i,0}) \perp (\boldsymbol{u}_{j,1} - \boldsymbol{u}_{j,0})$ for all $i,j \in [k]$ with $(i \neq j)$.

*Proof sketch.* On linearly separable data (guaranteed by Definition 3), GD on the logistic loss converges in direction to the max-margin solution (Soudry et al., 2024). Stability plus max-margin makes each training point a support vector, yielding prediction invariance when other concepts change. Flipping a concept then produces an additive shift, and per-concept weights are proportional to the segment connecting positive and negative classes, from which orthogonality follows. The minimal-dataset case is argued similarly via cross datasets of size $1 + k$ containing only pairs differing in one concept.

Intuitively, linear factorization means that a combination space of $n^k$ elements can be explained using only $n \cdot k$ factors. The orthogonality condition says that factors of concept values belonging to different concepts (e.g., "red" and "square") are orthogonal to each other, but no requirement is placed on the factors of concept values belonging to the same concept (e.g., "red" and "blue"). We illustrate the stable and unstable examples of feature representations in Fig. 5. Additionally, we note that linear factorization in itself is not trivial: the fact that $n^k$ datapoints can be explained using $n \cdot k$ factors does not have to hold for any linearly compositional model. We illustrate this with examples in Appendix G.4.

The datapoint requirement can be interpreted as operating in either (i) a minimal-learning regime for extrapolating to the whole grid (as in Compositional Risk Minimization framework (Mahajan et al., 2025)), where $|T| = 1 + k(n-1)$ suffices to extrapolate to the whole grid, or (ii) a large-sample

$k = 2, n = 20$ $\qquad$ $k = 3, n = 12$

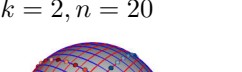 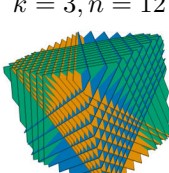

*Figure 6.* **Example geometries of linearly compositional models. Left:** 2 concepts ($n = 20$ each) on a sphere. Each colored stripe is the argmax boundary for one concept value; their intersections yield $20^2$ combination cells. **Right:** 3 concepts ($n = 12$ each) in 3D. Colored planes show argmax boundaries; their intersections carve out $12^3$ combination cells. Each boundary is colored according to the concept it belongs to.

regime in which random sampling yields near-complete coverage of the concept space.

**Empirical evaluation on synthetic data.** The necessary conditions above concern models that already support compositional generalization; networks trained from scratch may not. Extending Uselis et al. (2025) beyond their two-concept setting, our synthetic experiments under standard classification losses show that linearity and orthogonality emerge, especially as the number of concepts grows (Figs. 43 and 44; setup in Appendix H.7). In Appendix E.2 we additionally analyze an idealized setting where logits take two values ($h(\boldsymbol{z}) \in \{\alpha, \beta\}$), showing strong dimensionality requirements and additive representations that preserve all probe scores (Proposition 7).

> **Takeaway §4.1.** Under gradient descent with cross-entropy loss, compositional generalization with stable transfer requires linear factorization and orthogonality of cross-concept factors.

**Sufficiency.** The converse of Proposition 1 also holds: in the same binary GD+CE regime, linear factorization with cross-concept orthogonality is sufficient for compositional generalization (Proposition 4 and appendix C.1).

### 4.2. Packing and minimum dimension

So far we have established necessary and sufficient conditions for the representation space $\mathcal{Z}$ to exhibit compositional generalization. However, it is not clear what exactly the capacity constraints are on the representation space to support it. Here, we ask a basic capacity question: what is the minimum embedding dimension $d$ needed to support Divisibility (Desideratum 1), i.e. realize all possible $n^k$ combinations? The following result gives a tight lower bound.

**Proposition 2** (Minimum dimension for linear probes)**.** For $k$ concepts, each with $n$ values, suppose there exist linear probes that correctly classify each concept value for all $n^k$ combinations from embeddings $f(\boldsymbol{x}) \in \mathbb{R}^d$. Then necessarily $d \geq k$.

The bound is independent of the number of values $n$ per concept, requiring only that any two values per factor be dis-

tinguishable (so it holds for discrete and continuous factors alike). Fig. 6 illustrates two divisibility examples (sphere and Euclidean), with additional visualizations in Fig. 19. As $k$ grows for fixed $d$, factors within each concept must lie in progressively lower-dimensional subspaces, approaching near-collinearity (Fig. 14).

**Empirical results and the dependence on representation space and the loss function.** The result in Proposition 2 is a geometric lower bound and does not depend on a specific loss or representation space. Empirically, we find that CE in Euclidean space reaches this bound most closely, BCE typically requires higher dimension (approximately $2k$), and spherical geometry adds roughly one extra dimension (see the setting in Appendix H.7, and Fig. 45 for the empirical results). In Appendix E.2 we discuss an idealized case where a model's logits are constrained to take two values, yet satisfy perfect classification, and show that this necessarily requires $d \geq 1 + k(n-1)$ in Proposition 6.

> **Takeaway §4.2.** The minimum embedding dimension scales with the number of concepts $k$, not the number of values $n$ per concept ($d \geq k$). When $k$ grows relative to a fixed $d$, per-concept subspaces must become increasingly low-rank, approaching near-collinearity.

## 5. Surveying necessary conditions in pretrained models

Previous section established necessary conditions for compositional generalization: representation space must be linearly factorized and the factors must be orthogonal across concepts (Proposition 1). In this section, we ask: how far are modern pretrained models from these necessary conditions?

Our theory is built on binary concept values, but some of the concepts in the datasets we consider are multi-valued. Instead of repeatedly sampling binary values and testing factorization on these, we adopt the natural multivalued extension of the necessary structure. Concretely, we test whether representations admit an approximate additive decomposition of the form

$$\boldsymbol{z_c} \approx \sum_{i=1}^{k} \boldsymbol{u}_{i,c_i}, \quad (\boldsymbol{u}_{i,a} - \boldsymbol{u}_{i,b}) \perp (\boldsymbol{u}_{j,a'} - \boldsymbol{u}_{j,b'}), \quad (6)$$

for all $\boldsymbol{c} \in [n]^k$, all $i \neq j$, and all $a, b, a', b' \in [n]$, i.e., an additive per-concept factorization with cross-concept orthogonality of difference directions. This form reduces to the binary case when $n = 2$.

**Main qualitative result.** We give intuition for both conditions in (6) using a pretrained DINOv3 model on dSprites (Fig. 7). The figure shows that the idealized linearity and orthogonality conditions are visible in the global PCA view **(a)**, and that the same pattern is observed when some concepts are fixed and others are varied **(b)**, consistent with approximate additive factorization and cross-concept or-

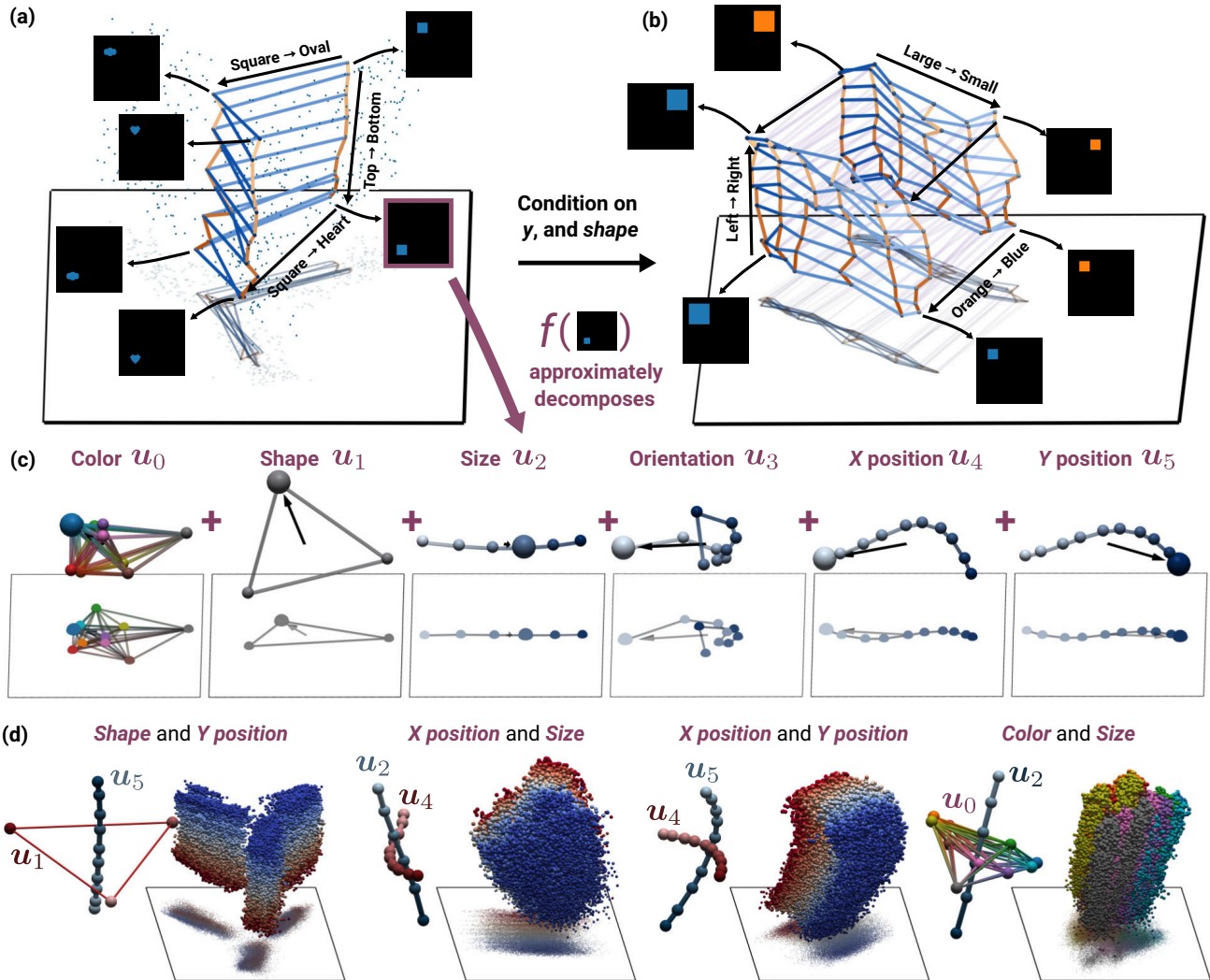

*Figure 7.* **Summary of the linearity + orthogonality hypothesis, illustrated in DINOv3 on dSprites. (a)** PCA projection of DINOv3 embeddings over all dSprites combinations. Changing shape or $y$-position produces near-constant direction shifts (linearity), and the two directions are nearly orthogonal. **(b)** After fixing shape to square and $y$ to one value, PCA on the remaining subset shows that horizontal position $x$ (left $\rightarrow$ right), size (large $\rightarrow$ small), and color (orange $\leftrightarrow$ blue) each vary along near-constant directions and form a grid-like structure. **(c)** Embeddings exhibit approximate linear factorization: each embedding decomposes as a sum of one factor per concept, $z_c \approx \sum_i u_{i,c_i}$ (illustrated for the highlighted sample, with arrows pointing to the selected factor in each panel). The recovered factors are typically low-rank ($\leq$ 3D), so these 3D plots capture most of their structure. **(d)** For each pair of concepts, the left mini-panel shows the two sets of factors, illustrating near-orthogonality across concepts. The right mini-panel shows all datapoints projected onto the span of those two factors; the projected points organize into a grid aligned with the factor directions, consistent with additive decomposition. The corresponding quantitative results are reported in Sections 5.1 and 5.3 and appendix H.3; full qualitative results in Appendix H.6.

thogonality. Any datapoint can approximately be expressed as a sum of the factor vectors $u$ **(c)**, and they often take low-rank structure, as well as satisfy orthogonality **(d)**. The remainder of this section quantifies these observations. For full qualitative results, see Appendix H.6.

Concretely, we quantitatively evaluate whether pretrained models exhibit linear factorization (Section 5.1), whether it correlates with compositional generalization (Section 5.2), whether factors are orthogonal across concepts (Section 5.3), and what geometric structure they exhibit (Appendix H.3).

**Models and datasets.** We evaluate the CLIP, SigLIP, and DINO model families on three compositional datasets (PUG-Animal, dSprites, MPI3D), plus ImageNet-AO; see Appendix G.5 for the full checkpoint and model details.

**Recovering the factors from representations.** Assuming that a linear factorization exists in the representations of a model $f$ as detailed in Section 4.1, we can recover the factors $\{u_{i,j}\}_{i\in[k],j\in[n]}$ by averaging over all the datapoints that share a particular concept value (Trager et al., 2023). For analysis purposes it is sufficient to recover the centered factors. That is, given all centered em-

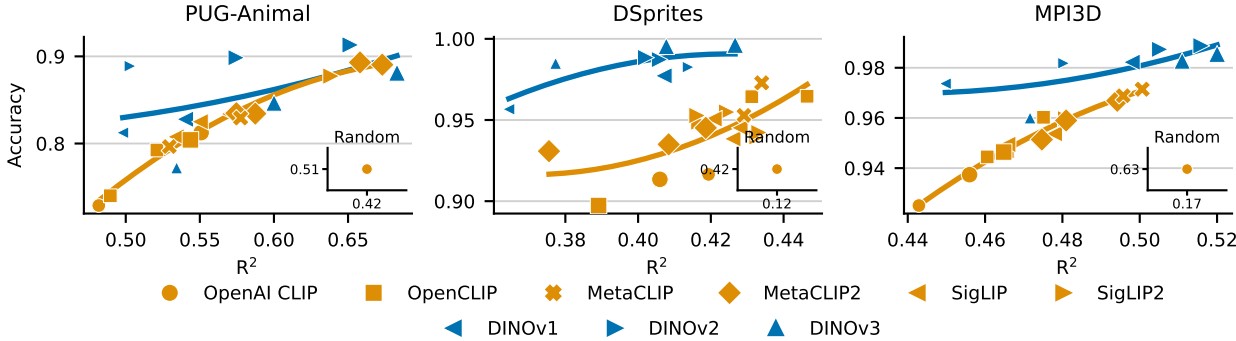

*Figure 8.* **Linearity in embeddings correlates with compositional generalization across VLMs and self-supervised vision models.** Across three datasets, we plot compositional generalization accuracy against projected $R^2$ (our linear-factorization score) for a broad set of pretrained encoders, including vision-language models (OpenAI CLIP, OpenCLIP, SigLIP, MetaCLIP, MetaCLIP2) and pure vision backbones (DINOv1–v3). Each marker is a model variant; higher projected $R^2$ is consistently associated with higher compositional generalization performance.

beddings $\{f(\boldsymbol{x_c})\}_{\boldsymbol{c}\in[n]^k}$, the factors can be recovered as $\boldsymbol{u}_{i,j} = \frac{1}{|\{\boldsymbol{c}\in[n]^k:c_i=j\}|} \sum_{\boldsymbol{c}\in[n]^k:c_i=j} f(\boldsymbol{x_c})$.

### 5.1. Linear factorization in pre-trained models

To assess the extent of linearity in the embeddings, we measure a whitened $R^2$ score on the probe span. We (1) project onto the probe span to discard information the embeddings may possess beyond the dataset concepts, and (2) whiten the embedding space so that $R^2$ is not inflated by a few dominant directions. Concretely, given the recovered approximate factors $\{\boldsymbol{u}_{i,j}\}_{i\in[k],j\in[n]}$, the $R^2$ score is computed as

$$R^2 = 1 - \frac{\sum_{\boldsymbol{x_c}\in\mathcal{D}}\left\|f(\boldsymbol{x_c}) - \sum_{i=1}^{k}\boldsymbol{u}_{i,c_i}\right\|_2^2}{\sum_{\boldsymbol{x_c}\in\mathcal{D}}\|f(\boldsymbol{x_c}) - \bar{\boldsymbol{f}}\|_2^2}, \qquad (7)$$

where $\mathcal{D}$ is the dataset, and $\bar{\boldsymbol{f}}$ is the mean embedding. Note that a score of 1.0 indicates perfect linearity. We provide intuition of linear factorization and its relation to the $R^2$ in Appendix G.3, additional justification of whitening in Appendix G.2.

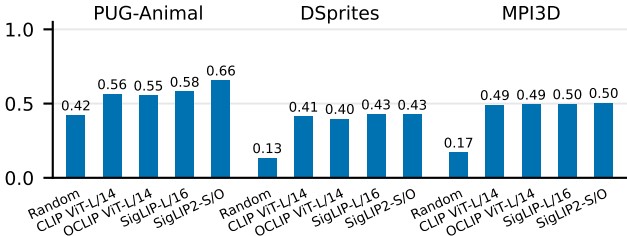

*Figure 9.* **Linear factorization partly explains current models' embedding spaces.** Bar plots of whitened $R^2$ on three datasets with varying concept/value counts.

**Results.** Fig. 9 shows projected $R^2$ scores across models and datasets. Among all datasets, each model's $R^2$ score is consistently above the random baseline (0.4-0.6 vs. 0.12-0.42, respectively). This suggests that embeddings are partially captured by a sum of per-concept components, while still leaving some information unexplained. Additionally, we observe that $R^2$ scores are similar in scale across models. The same linearity trend holds in the zero-shot setting when

using text encoders as probes on both PUG-Animal and ImageNet-AO (Appendix H.4 and Figs. 28 and 32).

Importantly, we note that the $R^2$ scores, while consistently above random, are far from perfect, indicating that current models only partially satisfy the linear factorization predicted by our theory.

> **Takeaway §5.1.** Embeddings exhibit partial linear factorization, explaining a moderate fraction of the variance via per-concept components. The gap from perfect scores highlights a divergence from the ideal embedding structure theory predicts.

### 5.2. Compositional generalization and linear factorization

We ask whether the *degree* of linear factorization predicts compositional generalization.

**Metrics and setup.** For each dataset/model, we train linear probes on 10% of all concept combinations and evaluate on the held-out 10% unseen compositions (cf. sampling discussion in Section 4.1). This corresponds to a validity rule $R(T) = 1$ if $|T| = 0.1\,n^k$. We compute *Projected $R^2$* on *whitened* $\boldsymbol{P_W}f(\boldsymbol{x})$, where $\boldsymbol{P_W}$ projects onto the span of the probe weights $\boldsymbol{W}$ (Section 5.1) and pair it with a *compositional accuracy* score on the held-out compositions. We use a randomly-initialized OpenCLIP ViT-L/14 model as a baseline by training linear probes on the embeddings. We use linear probing rather than zero-shot classification to avoid prompt-specification issues, nonetheless, the same conclusions hold in zero-shot experiments on both PUG-Animal and ImageNet-AO (Appendix H.4).

Compositional accuracy is computed by training one linear classifier per concept, then averaging each classifier's accuracy on the held-out combinations. For example, dSprites has 6 concepts (shape, orientation, $x$ position, $y$ position, size, and color); we train 6 classifiers and report their mean accuracy on unseen combinations.

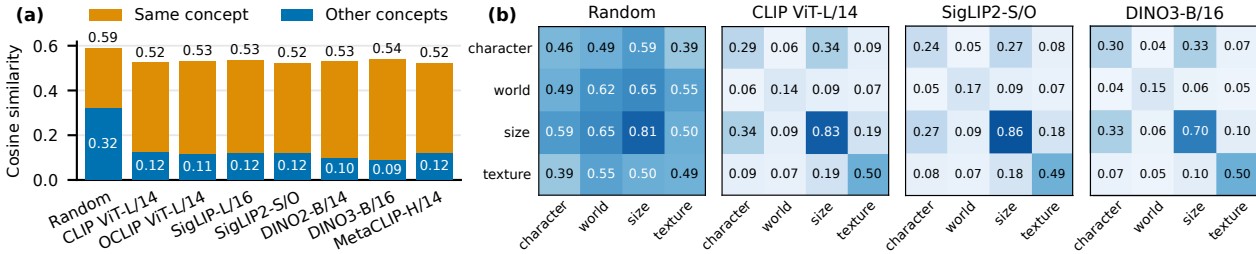

*Figure 10.* **Pre-trained models exhibit strong within-concept direction similarity and partial orthogonality across concepts. (a)** Aggregated within-concept direction similarity over datasets. **(b)** Pairwise average cosine across concepts. Lower values indicate greater orthogonality between factor vectors. Full orthogonality results are reported in Appendix H.2.

**Results.** Across all datasets *higher Projected $R^2$ coincides with higher compositional accuracy* (Fig. 8). The randomly initialized OpenCLIP ViT-L/14 baseline consistently occupies the low-$R^2$/low-accuracy corner, indicating the effect is not a dimensionality or scale artifact, and follows the rationale of sanity checks in interpretability (Méloux et al., 2025). The same trend is observed in zero-shot text-probe experiments on both PUG-Animal and ImageNet-AO (Appendix H.4 and Figs. 28 and 32). The correlation persists across train fractions $|S|/N \in \{0.001, \dots, 0.999\}$ and on the full $32{\times}32$ dSprites grid (Appendices H.1, H.5.1, H.5.2 and H.5.4), consistent with our theory: models whose embeddings are closer to a linear factorization also generalize better to unseen concept combinations.

> **Takeaway §5.2:** Linear factorization in pre-trained models correlates positively with compositional generalization performance.

### 5.3. Orthogonality of factors

Our theory (Proposition 1) predicts that per-concept difference vectors should be orthogonal *across* concepts under linear factorization, in generalizing linearly compositional models. We empirically test this prediction by testing orthogonality in two ways: (1) within-concept and (2) across-concept. We defer the details to Appendix H.2.

**Results.** Pretrained encoders exhibit consistently higher direction similarity within concepts than across concepts (Fig. 10): within-concept similarity (a) is around $\approx 0.53$-$0.55$, whereas cross-concept similarity (b) is $\approx 0.09$-$0.12$. The randomly-initialized encoder also exhibits this pattern; however, the across-concept similarity is higher (0.32 on average) compared to pre-trained models. The same within-vs-across pattern is also observed in zero-shot text-probe experiments on both PUG-Animal and ImageNet-AO (Figs. 29 and 33). Error bars across concept pairs and a re-run on the full $32{\times}32$ dSprites grid are reported in Appendix H.5.3.

> **Takeaway §5.3:** Pretrained models exhibit partial cross-concept orthogonality, substantially more so than randomly initialized encoders, suggesting that training drives factor directions toward the geometry predicted by our theory.

**Dimensionality of factors.** Per-concept factors in pretrained models are typically low-dimensional: ordinal and continuous concepts are captured by $\leq 4$ PCs (often $\leq 2$), while discrete concepts show higher effective rank. Factor geometry is also similar across model families (CLIP, OpenCLIP, SigLIP, SigLIP2), in line with the Platonic Representation Hypothesis (Huh et al., 2024). See Appendix H.3 for full results.

## 6. Discussion and conclusion

In this work, we formalized compositional generalization through three practically motivated desiderata (divisibility, transferability, stability) and showed they force representations into linear factorization with cross-concept orthogonality, with a minimum embedding dimension $d \geq k$. This reframes the Linear Representation Hypothesis as a necessary consequence of compositional generalization rather than just an empirical observation.

Empirically, across modern VLMs and compositional datasets, current representations partially satisfy this structure, and the degree of factorization correlates with compositional generalization on unseen combinations. The pattern holds for both vision-language models and self-supervised DINO encoders. The $R^2$–accuracy correlation gives a practical diagnostic for assessing compositional capability directly in representation space; the gap from perfect factorization may explain current models' failures on compositional benchmarks, and our conditions predict the geometry they may converge to as they scale.

**Limitations and future work.** Our theory emphasizes worst-case stability over valid training supports; relaxing this to average-case or approximate stability is a natural direction that may better match some practical settings. Characterizing how different training supports change the implied geometry could turn these necessary conditions into a practical design guide for data collection and model building. Our theory assumes a fixed encoder and considers retraining across different training supports; in practice the encoder is trained once on a single dataset; understanding the impact of this assumption on the necessary conditions is an interesting direction for future work.

## Acknowledgments

We would like to thank Divyat Mahajan for useful discussions and insights, as well as comments on an early version of this work. We also thank Alexander Rubinstein and Darina Koishigarina for insightful discussions, and Simon Buchholz for useful comments. This work was supported by the Tübingen AI Center. Arnas Uselis was supported by the International Max Planck Research School for Intelligent Systems (IMPRS-IS). Seong Joon Oh was supported by the Institute for Information & Communications Technology Planning & Evaluation (IITP) grant funded by the Korea government (MSIT) (RS-2019-II190075, Artificial Intelligence Graduate School Program gram(KAIST)).

## Impact statement

This paper presents work whose goal is to advance the field of machine learning. There are many potential societal consequences of our work, none of which we feel must be specifically highlighted here.

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

# Appendix

## Contents

# A. Extended related work

**Compositional generalization.** Research on compositional generalization investigates how models can systematically combine concepts. On the objective side, approaches such as Compositional Feature Alignment (Wang, 2025) and Compositional Risk Minimization (Mahajan et al., 2025) study how model training objectives, and model architecture (Jarvis et al., 2024) affect compositional generalization. On the representational side, kernel analyses characterize when certain compositional structures in embeddings yield generalization theoretically (Lippl & Stachenfeld, 2025), and empirical work investigates the role of disentangled representations for compositional generalization (Montero et al., 2021; Dittadi et al., 2021; Liang et al., 2025). On the data side, recent work probes whether and how scaling and data coverage improve compositional behavior (Uselis et al., 2025; Schott et al., 2022; Kempf et al., 2025). Abbasi et al. (2024) investigate CLIP's ability to recognize unlikely attribute-object combinations, finding that CLIP models still fall short on such tasks.

Other works establish formal sufficient conditions for when particular model classes can achieve compositional generalization, e.g., generative models whose data are produced by a differentiable rendering process and whose training distribution provides compositional support over latent factors (Wiedemer et al., 2023), constrained decoders paired with explicit inversion (Brady et al., 2025), conditional diffusion models with local conditional scores (Bradley, 2025), discriminative models whose inputs are drawn from an additive energy distribution (Mahajan et al., 2025), or linearly factorized representations (Uselis et al., 2025). In contrast, we do not impose specific structure on the data-generating process or on the learned representations. Instead, we ask what properties are implied *if* a model transfers from a restricted subset of the data space to the full space under our desiderata. Within this setting, our results can be interpreted as providing *necessary* conditions for compositional generalization for models that satisfy these desiderata.

**Geometry of learned representations.** A large literature studies the shape of learned features. In VLMs, Trager et al. (2023) report compositional linear subspaces, while in LLMs the *Linear Representation Hypothesis* (LRH) is examined mechanistically and statistically (Jiang et al., 2024; Park et al., 2023). Extending LRH, Engels et al. (2025) show that features can be multi-dimensional rather than rank-1, and Roeder et al. (2020) analyze identifiability constraints. Sparse-autoencoder probes provide evidence for monosemantic or selectively remapped features in VLMs (Pach et al., 2025; Zaigrajew et al., 2025; Lim et al., 2025). Beyond nominal labels, ordinal/ordered concepts motivate the rankability of embeddings (Sonthalia et al., 2025). More broadly, capacity limits for embedding-based retrieval emphasize geometric bottlenecks (Weller et al., 2025a). (Elhage et al., 2022) investigated empirically how neural networks can represent more features than there are dimensions in two-layer auto-encoder models. They found a tendency to encode features near-orthogonally with respect to neurons. Abbasi et al. (2024) find evidence of disentanglement in CLIP models. In contrast to these works, which document linear or near-orthogonal structure empirically, we show that under practice-driven desiderata and standard training, linearity and orthogonality are *necessary*.

Concurrent to our study of compositional generalization constraints in embedding spaces, Fel et al. (2025) conduct a large-scale, empirical concept analysis of DINOv2 using sparse autoencoders, showing strong task-dependent structure in which different downstream tasks recruit different concept families. They further propose a "Minkowski Representation Hypothesis", in which token activations behave as a Minkowski sum of convex polytopes, emphasizing both the interpretability consequences and the non-uniqueness/identifiability limitations of such decompositions. While their focus is token-level geometry and interpretability (rather than CLS-centric compositional probing and necessary conditions), their observations offer complementary evidence that modern vision encoders exhibit rich internal geometric structure beyond a purely unstructured embedding space.

**Data, objectives, and training effects on geometry.** Data distribution strongly shapes zero-shot behavior; concept frequency during pretraining predicts multimodal performance (Udandarao et al., 2024). On the objective side, BCE vs. CE can induce different feature geometries (Li et al., 2025), and contrastive/InfoNCE objectives exhibit characteristic similarity patterns (Lee et al., 2025). Convergence perspectives argue that the *objective* drives canonical representational forms (Huh et al., 2024), and objective choice has been tied to representational similarity across datasets (Ciernik et al., 2025).

**Binding, explicit structure injection, and concept identification.** Work on *binding* asks whether models maintain factored world states (Feng et al., 2025), and CLIP has been observed to show uni-modal binding (Koishigarina et al., 2025). Surveys and empirical studies examine binding limits and emergent symbolic mechanisms (Campbell et al., 2025; Assouel et al., 2025). Other approaches inject structure directly, e.g., hyperbolic image-text embeddings and entailment learning (Pal et al., 2024; Desai et al., 2024), or pursue concept identification at the causal/foundation interface and object-centric pipelines (Rajendran et al., 2024; Mamaghan et al., 2024).

**Relation to disentangled and object-centric representations.** Work on disentangled representations largely focuses on specifying desiderata for internal codes (e.g., disentanglement, completeness, informativeness) and proposing metrics or training schemes to satisfy them, often with the informal motivation that such structure should help downstream generalization (Bengio et al., 2014; Eastwood & Williams, 2018; Higgins et al., 2018). Closely related, object-centric representation learning injects an inductive bias that scenes should be modeled as compositions of objects, which can be viewed as a structured form of factorization/binding that may support compositional transfer and robustness (Greff et al., 2020). Recent studies directly probe how these properties relate to out-of-distribution or compositional generalization, often with mixed or limited evidence (Watters et al., 2019; Dittadi et al., 2021; 2022; Montero et al., 2021; 2024; Kapl et al., 2026). We instead ask a complementary question: if a discriminative model *does* exhibit compositional generalization when learned from a subset of the data space, what must necessarily be true of its embeddings?

A large body of literature has studied the usefulness and implications of learning disentangled representations in an unsupervised way (Bengio et al., 2014; Lake et al., 2017). Most commonly, the goal is to learn a generative model, usually through a VAE (Kingma & Welling, 2014), that can compress the data in a disentangled manner, in a way that allows to reconstruct these representations. While shown to be impossible without additional assumptions (Locatello et al., 2019), under weak supervision learning is possible (Shu et al., 2020; Locatello et al., 2020). Measuring the degree of disentanglement in these models is in itself non-trivial and various metrics have been proposed, e.g. by measuring disentanglement by performing interventions on the representations (Higgins et al., 2017; Kim & Mnih, 2018).

The DCI framework (Eastwood & Williams, 2018) proposes desiderata of properties disentangled representations should satisfy, namely disentanglement, completeness, and informativeness, and proposes a metric to measure them. Some works also consider what constitutes a good disentanglement (Higgins et al., 2018) and propose a conceptual framing of meaning behind disentangled representations with respect to the data generative process in terms of group actions of transformations.

Abbasi et al. (2024) investigate the role of representation disentanglement in compositional generalization in CLIP models. Using metrics such as DCI, they find that CLIP models with more disentangled text and image representations exhibit higher compositional OOD accuracy on their attribute-object dataset (ImageNet-AO). This work is complementary to ours. Their study explores correlations between disentanglement and compositional generalization by probing CLIP embeddings with respect to the adjective and noun components present in the inputs. For instance, they estimate "attribute" and "object" subspaces by feeding isolated adjectives or nouns into the text encoder, or by generating isolated attributes/objects via a text-to-image model and embedding them with CLIP. However, this approach assumes that CLIP's embedding space is additively decomposed with respect to individual words, an assumption that is not guaranteed to hold. Indeed, Yamada et al. (2024) show that word embeddings in language models are often highly entangled with associated concepts. In contrast, our necessary condition does not rely on word-level decomposition. We posit that models achieving perfect downstream compositional performance must possess linearly factorized representations that separate per-concept components, independent of how an encoder processes individual words. In short, our work provides principled motivation for analyses of representational decomposition, whereas Abbasi et al. (2024) offer an empirical correlation study based on CLIP's emergent disentanglement.

Lippl & Stachenfeld (2025) investigate when a particular form of compositionally structured representations, specifically representations whose similarity depends only on how many underlying components two inputs share, supports downstream compositional generalization. Using kernel theory, they characterize exactly which tasks linear readouts on top of such representations can solve, showing that these models are fundamentally restricted to conjunction-wise additive functions. In contrast, we focus on a specific subclass of compositional tasks: identifying factors of inputs that never co-occur during training. While Lippl & Stachenfeld (2025) characterize what kinds of generalization are possible under a compositional representational structure, we ask the complementary question: given perfect downstream performance on such a task, what representational structure must the model necessarily possess under the desiderata we specify?

# B. Necessary conditions (proof of Proposition 1)

In this section, we prove Proposition 1 and state the auxiliary lemmas used in the argument.

Our default training-support regime is the one used in the main result: valid supports satisfy $|T| = 2^{k-1} + 1$. In the proof below, we work with cross-datasets $\mathcal{D}^{\boldsymbol{c}}$ (Definition 6), i.e., datasets centered at $\boldsymbol{c}$ that vary one concept at a time, because this makes the core geometric argument transparent. We write $\mathcal{D}$ for the full dataset of all $n^k$ combinations and $\mathcal{D}^{\boldsymbol{c}}$ for a cross-dataset centered at $\boldsymbol{c}$.

Although most proofs in this section are written in the binary setting for clarity, we state intermediate claims in a form that extends directly to general $n$ where possible. Concretely, in the binary setting $\mathcal{C}_i = \{0, 1\}$. Any learned quantity carries a superscript indicating the training set, e.g., $\boldsymbol{w}_i^{(\mathcal{D})}$ or $\boldsymbol{w}_i^{(\mathcal{D}^{\boldsymbol{c}})}$. For concept $i$ and training set $S$, we use the logistic score form

$$h_i^{(S)}(\boldsymbol{z}) := (\boldsymbol{w}_i^{(S)})^{\mathsf{T}}\boldsymbol{z} + b_i^{(S)}, \qquad p_i^{(S)}(\boldsymbol{z}) = \sigma(h_i^{(S)}(\boldsymbol{z})),$$

and we drop the superscript when the dataset is clear from context.

**Definition 5** (Intervention on a concept value). For any concept index $i \in [k]$, target value $j \in [n]$, and concept vector $\boldsymbol{c} \in [n]^k$, we define the intervened index and representation

$$\boldsymbol{c}(i \to j) := (c_1, \ldots, c_{i-1}, j, c_{i+1}, \ldots, c_k), \quad \boldsymbol{z}_{\boldsymbol{c}(i \to j)} := \boldsymbol{z}_{\boldsymbol{c}} \text{ with concept } i \text{ set to } j.$$

In case of multiple interventions, they compose componentwise, e.g. for distinct $i, m \in [k]$, $\boldsymbol{c}(i \to j, m \to l) = (c_1, \ldots, c_{i-1}, j, c_{i+1}, \ldots, c_{m-1}, l, c_{m+1}, \ldots, c_k)$.

**Definition 6** (Cross dataset at $\boldsymbol{c}$). Given a concept space $\mathcal{C} = \mathcal{C}_1 \times \cdots \times \mathcal{C}_k$, we say that a dataset $\mathcal{D}^{\boldsymbol{c}}$ is a cross-dataset at $\boldsymbol{c} \in [n]^k$ if:

1. It contains only samples that vary one concept at a time around the center $\boldsymbol{c}$:

$$\mathcal{D}^{\boldsymbol{c}} = \big\{(c_1', c_2, \ldots, c_k) : c_1' \in [n]\big\} \cup \cdots \cup \big\{(c_1, c_2, \ldots, c_{k-1}, c_k') : c_k' \in [n]\big\} = \{\boldsymbol{c}\} \cup \bigcup_{i=1}^{k} \bigcup_{a \in [n] \setminus \{c_i\}} \{\boldsymbol{c}(i \to a)\}.$$

2. Its size is $1 + k(n-1)$,

**Definition 7** (Dataset marginal counts). For any dataset $S \subseteq \{(\boldsymbol{z}_{\boldsymbol{c}}) : \boldsymbol{c} \in [n]^k\}$ (e.g., $S = \mathcal{D}$ or $S = \mathcal{D}^{\boldsymbol{c}}$), concept $i \in [k]$, and value $j \in [n]$, the marginal count of value $j$ in $S$ is

$$N_{i,j}(S) := \big|\{\boldsymbol{c} \in [n]^k : (\boldsymbol{z}_{\boldsymbol{c}}) \in S, c_i = j\}\big|.$$

When $S$ is clear, we abbreviate $N_{i,j} := N_{i,j}(S)$.

**Remark 1** (Marginal counts: full vs cross-datasets). For the full dataset $\mathcal{D}$, the marginal counts are balanced:

$$N_{i,j}(\mathcal{D}) = n^{k-1} \quad \text{for all } i \in [k], \ j \in [n].$$

For a cross-dataset $\mathcal{D}^{\boldsymbol{c}}$ as in Definition 6, the marginal counts satisfy

$$N_{i,j}(\mathcal{D}^{\boldsymbol{c}}) = \begin{cases} 1 + (k-1)(n-1), & j = c_i, \\ 1, & j \neq c_i. \end{cases}$$

*Proof.* In $\mathcal{D}$, fixing $c_i' = j$ leaves $n^{k-1}$ free coordinates. In $\mathcal{D}^{\boldsymbol{c}}$: varying concept $i$ contributes one point for each $j \neq c_i$; the center contributes one more with $c_i' = c_i$; varying any other concept $r \neq i$ adds $(n-1)$ points with $c_i' = c_i$, across $(k-1)$ such concepts, totaling $(k-1)(n-1)$. $\qquad\square$

**Definition 8** (Binary complement notation). In the binary case ($\mathcal{C}_i = \{0, 1\}$), we write $\bar{c}_i := 1 - c_i$ for the complement value of concept $i$. As shorthand for an intervention to the complement, we write $\boldsymbol{c}(\bar{c}_i) := \boldsymbol{c}(i \leftarrow \bar{c}_i)$.

To make use of Stability in the binary case, we use the identifiability of the sigmoid.

**Lemma 1** (Binary equal probabilities imply equal affine parameters). For a concept index $i$ and two training supports $S, S'$, let

$$h_i^{(S)}(\boldsymbol{z}) := (\boldsymbol{w}_i^{(S)})^{\mathsf{T}}\boldsymbol{z} + b_i^{(S)}, \quad h_i^{(S')}(\boldsymbol{z}) := (\boldsymbol{w}_i^{(S')})^{\mathsf{T}}\boldsymbol{z} + b_i^{(S')}.$$

Assume that for every $\boldsymbol{z} \in \mathbb{R}^d$,

$$\sigma\left(h_i^{(S)}(\boldsymbol{z})\right) = \sigma\left(h_i^{(S')}(\boldsymbol{z})\right),$$

where $\sigma(t) = 1/(1 + e^{-t})$. Then

$$\boldsymbol{w}_i^{(S)} = \boldsymbol{w}_i^{(S')} \qquad \text{and} \qquad b_i^{(S)} = b_i^{(S')}.$$

*Proof.* Since $\sigma$ is injective,

$$h_i^{(S)}(z) = h_i^{(S')}(z) \qquad \forall z \in \mathbb{R}^d.$$

Hence

$$\left(\boldsymbol{w}_i^{(S)} - \boldsymbol{w}_i^{(S')}\right)^\mathsf{T} \boldsymbol{z} + \left(b_i^{(S)} - b_i^{(S')}\right) = 0 \qquad \forall \boldsymbol{z} \in \mathbb{R}^d.$$

Taking $\boldsymbol{z} = \boldsymbol{0}$ gives $b_i^{(S)} = b_i^{(S')}$, and then

$$\left(\boldsymbol{w}_i^{(S)} - \boldsymbol{w}_i^{(S')}\right)^\mathsf{T} \boldsymbol{z} = 0 \qquad \forall \boldsymbol{z} \in \mathbb{R}^d,$$

implies $\boldsymbol{w}_i^{(S)} = \boldsymbol{w}_i^{(S')}$, since the equation holds for any point in $\mathbb{R}^d$. $\square$

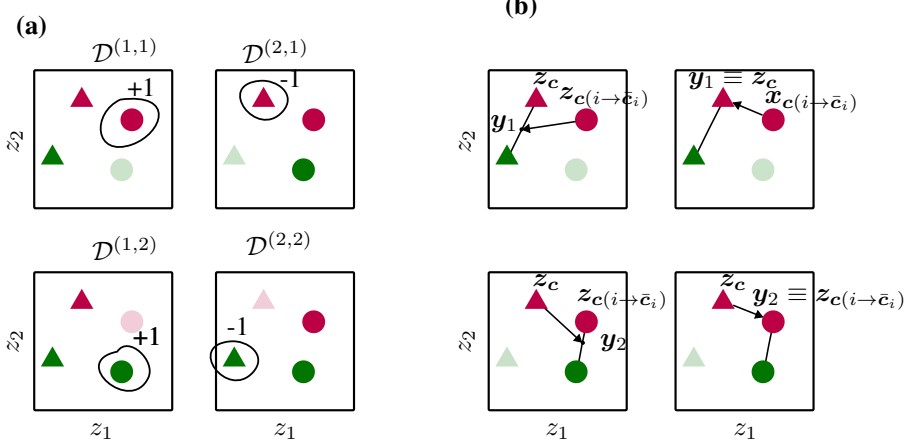

*Figure 11.* **Illustration of the invariance lemma (left) and the main proposition (right). (a)** The invariance lemma (Lemma 3): any point $z_c$ can be made a support vector by an appropriate choice of a pair of datasets. **(b)** The main proposition (Proposition 1): linearity and orthogonality under GD+BCE follow because, for each concept $i$, the SVM support vectors correspond to counterfactual pairs that differ only in the $i$-th concept. Concretely, when the minority support point is $z_{c(i \to \bar{c}_i)}$, the majority-class support vector $y_1$ under $\mathcal{D}^{(1,1)}$ reduces to $z_c$, and analogously for $y_2$.

**Lemma 2** (Bi-directional tight support vectors). For binary concepts $\mathcal{C}_i = \{0,1\}$ for all $i \in [k]$, consider a pair of cross-datasets $\mathcal{D}^c$ and $\mathcal{D}^{c(i \to \bar{c}_i)}$ for some $i \in [k]$ and the corresponding SVM solutions $\{\boldsymbol{w}_i, b_i\}$ and $\{\boldsymbol{w}_i', b_i'\}$ for these datasets, respectively. Then, $z_{c(i \to \bar{c}_i)}$ is a tight support vector for concept $i$ under $(\boldsymbol{w}_i, b_i)$ trained on $\mathcal{D}^c$, and $z_c$ is a tight support vector under $(\boldsymbol{w}_i', b_i')$ trained on $\mathcal{D}^{c(i \to \bar{c}_i)}$, i.e. the following hold:

$$\begin{aligned} (\boldsymbol{w}')_i^\mathsf{T} z_c + b_i' &= y_i(\boldsymbol{c}) \\ \boldsymbol{w}_i^\mathsf{T} z_{c(i \to \bar{c}_i)} + b_i &= y_i(\boldsymbol{c}(i \to \bar{c}_i)), \end{aligned} \qquad (8)$$

where $y_i(\boldsymbol{c}) \in \{+1, -1\}$ is the label of $z_c$ with respect to concept $i$.

*Proof.* For the second equality: by Remark 1, $N_{i,c_i}(\mathcal{D}^c) = 1 + (k-1)(n-1)$ and $N_{i,\bar{c}_i}(\mathcal{D}^c) = 1$. Hence both classes are non-empty in $\mathcal{D}^c$; standard hard-margin SVM argument then gives at least one support vector in each class achieving equality at the margin (Cortes & Vapnik, 1995). Repeating the same argument for $\mathcal{D}^{c(i \to \bar{c}_i)}$ gives the first equation. $\square$

**Lemma 3** (Invariance to irrelevant concepts). Assume each concept is binary, $\mathcal{C}_i = \{0,1\}$ for all $i \in [k]$, and Desiderata 1–3 hold, and consider either the rule $|T| = 2^{k-1} + 1$ or the cross-dataset design (Definition 6). Then, for any $i \in [k]$ and any $\boldsymbol{c}, \boldsymbol{c}' \in [2]^k$ with $c_i = c_i'$, it holds that

$$P(C_i = c_i \mid z_c) = P(C_i = c_i \mid z_{c'}). \qquad (9)$$

*Proof.* We encode the $i$-label by $y_i(z) \in \{+1, -1\}$ with $y_i(z) = +1$ iff $C_i(z) = 1$ and $-1$ otherwise, and let

$$h_i(z) := \boldsymbol{w}_i^\mathsf{T} z + b_i \qquad (10)$$

By Stability (Desideratum 3) and Lemma 1, the affine separator $(\boldsymbol{w}_i, b_i)$ is the same across valid cross-datasets. For any $\boldsymbol{c} \in [2]^k$, consider the cross-dataset $\mathcal{D}^{c(i \to \bar{c}_i)}$. By Remark 1, in this dataset concept $i$ has count 1 for value $c_i$ and count $1 + (k-1)(n-1)$ for value $\bar{c}_i$, so

the unique minority example with respect to concept $i$ is $\boldsymbol{z_c}$. By Lemma 2, both classes have tight support vectors. Since the minority class has exactly one point, that point $\boldsymbol{z_c}$ is the tight support vector for its class. Hence

$$y_i(\boldsymbol{z_c})h_i(\boldsymbol{z_c}) = 1. \tag{11}$$

The same argument applies to $\boldsymbol{c}(i \to \bar{c}_i)$ in $\mathcal{D}$, where it follows that $y_i(\boldsymbol{z_{c(i \to \bar{c}_i)}})\, h_i(\boldsymbol{z_{c(i \to \bar{c}_i)}}) = 1$.

Since this was performed for any $\boldsymbol{c}$, it follows that

$$h_i(\boldsymbol{z}) = \begin{cases} +1, & \text{if } C_i(\boldsymbol{z}) = 1, \\ -1, & \text{if } C_i(\boldsymbol{z}) = 0, \end{cases} \quad \text{on the whole grid } \{\boldsymbol{z_c} : \boldsymbol{c} \in [2]^k\}.$$

Hence $h_i(\boldsymbol{z})$ depends only on $C_i(\boldsymbol{z})$ and not on the other concepts. Since $P(C_i = 1 \mid \boldsymbol{z}) = \sigma(h_i(\boldsymbol{z})) = \frac{1}{1+e^{-h_i(\boldsymbol{z})}}$ (and $P(C_i = 0 \mid \boldsymbol{z}) = 1 - P(C_i = 1 \mid \boldsymbol{z})$), the conditional probability $P(C_i = c_i \mid \boldsymbol{z_c})$ is constant over all $\boldsymbol{c}$ with fixed $c_i$. In particular, for any $\boldsymbol{c}, \boldsymbol{c}'$ with $c_i = c_i'$,

$$P(C_i = c_i \mid \boldsymbol{z_c}) = P(C_i = c_i' \mid \boldsymbol{z_{c'}}).$$

$\square$

Next, we state an important property of SVMs on two separable sets and adapt it to our case, where one of the elements in the set is a singleton.

**Lemma 4** (SVM geometry for separable sets)**.** Given a set of points $\mathcal{Y} := \{\boldsymbol{y}_i\}_{i=1}^{N}$ with $\boldsymbol{y}_i \in \mathbb{R}^d$, and a point $\boldsymbol{z} \in \mathbb{R}^d$, let $(\boldsymbol{w}, b)$ be the optimal hard-margin SVM separator between classes $\mathcal{Y}$ and $\{\boldsymbol{z}\}$, with the canonical scaling $\boldsymbol{w}^\intercal \boldsymbol{x} + b = \pm 1$ on support vectors. Then:

1. There exist coefficients $\{\lambda_i\}_{i=1}^{N}$ with $\lambda_i \geq 0$ and $\sum_i \lambda_i = 1$ such that

$$\boldsymbol{w}^\intercal \left( \sum_i \lambda_i \boldsymbol{y}_i \right) + b = -1, \tag{12}$$

$$\boldsymbol{w}^\intercal \boldsymbol{z} + b = +1. \tag{13}$$

2. The weight $\boldsymbol{w}$ is related to the shortest-segment between the convex hulls of the two classes as

$$\frac{2}{\|\boldsymbol{w}\|^2}\boldsymbol{w} = \boldsymbol{z} - \sum_i \lambda_i \boldsymbol{y}_i. \tag{14}$$

*Proof.* This is a standard geometric characterization of hard-margin SVMs: $\boldsymbol{w}$ is parallel to the shortest segment joining the convex hulls of the two classes, and under canonical margin scaling the two supporting hyperplanes are at signed distance $1/\|\boldsymbol{w}\|$ from the decision hyperplane, hence the support-point displacement equals $\frac{2}{\|\boldsymbol{w}\|^2}\boldsymbol{w}$ (Bennett & Bredensteiner, 2000). $\square$

We now establish the main result on the resulting geometry of linearly generalizable compositional models. Intuition of it, together with the invariance lemma above is presented in Fig. 11.

**Proposition 1** (Compositional generalization implies linear factorization)**.** Let $\Pi = (f, \mathcal{H}, A, \mathcal{T})$ be the tuple instantiated in Definition 4, with linear heads $\mathcal{H}$ and $A$ given by GD+CE. Suppose the validity rule is one of: (i) $R(T) = 1$ iff $|T| = 2^{k-1} + 1$ (random sampling, more than half the grid), or (ii) $R(T) = 1$ iff $|T| = 1 + k$ and $T$ consists of an anchor point together with $k$ points each differing from it in exactly one concept. Assume Desiderata 1–3 are satisfied. Then, under the binary grid $\mathcal{C}_i = \{0, 1\}$ with $\mathcal{Z} = \{\boldsymbol{z_c} : \boldsymbol{c} \in [2]^k\} \subset \mathbb{R}^d$, there exist $\{\boldsymbol{u}_{i,0}, \boldsymbol{u}_{i,1} \in \mathbb{R}^d\}_{i=1}^{k}$ such that for every $\boldsymbol{c} \in [2]^k$ the following holds:

1. *(Linearity)* $\boldsymbol{z_c} = \sum_{i=1}^{k} \boldsymbol{u}_{i,c_i}$.

2. *(Cross-concept orthogonality)* $(\boldsymbol{u}_{i,1} - \boldsymbol{u}_{i,0}) \perp (\boldsymbol{u}_{j,1} - \boldsymbol{u}_{j,0})$ for all $i, j \in [k]$ with $(i \neq j)$.

*Proof.* Although Proposition 1 is stated for the $|T| = 2^{k-1} + 1$ regime, the geometric mechanism is easiest to see on the cross-dataset construction $\mathcal{D}^{\boldsymbol{c}}$ (Definition 6), which isolates one-concept flips around a center point $\boldsymbol{c}$ (in the binary case, $|\mathcal{D}^{\boldsymbol{c}}| = 1 + k$). We therefore present the proof using cross-datasets to keep the SVM geometry and the stability constraints transparent.

**Linearity**.

The idea is to show that for a pair of cross-datasets that share the datapoints in negative class, the shortest distance from a single point in the positive class to the convex set of the positive points is achieved by considering a flip in one of the concepts. We make this concrete below.

Consider any point $\boldsymbol{z_c}$ and its corresponding cross-dataset $\mathcal{D}^{\boldsymbol{c}}$. For any concept $i \in [k]$, let $\boldsymbol{z_{c(i \to \bar{c}_i)}}$ be the counterfactual point obtained by flipping concept $i$ to $\bar{c}_i$, and consider the neighboring cross-dataset $\mathcal{D}^{\boldsymbol{c}(i \to \bar{c}_i)}$.

Note that for the concept $i$ it holds that:

1. Under $\mathcal{D}^{\boldsymbol{c}} = \{\boldsymbol{z}_{\boldsymbol{c}}\} \cup \{\boldsymbol{z}_{\boldsymbol{c}(j \to \bar{c}_j)} : j \in [k]\}$. For each concept $i$, the marginal counts are

$$N_{i,c_i}(\mathcal{D}^{\boldsymbol{c}}) = k, \qquad N_{i,\bar{c}_i}(\mathcal{D}^{\boldsymbol{c}}) = 1 \tag{15}$$

(by Remark 1). Thus $\boldsymbol{z}_{\boldsymbol{c}(j \to \bar{c}_j)}$ is the unique minority example for concept $j$ (with label $\bar{c}_j$), and

$$\mathcal{Y}_1 := \mathcal{D}^{\boldsymbol{c}} \setminus \{\boldsymbol{z}_{\boldsymbol{c}(i \to \bar{c}_i)}\} \tag{16}$$

is the set of $k$ majority examples (with label $c_i$).

2. Under

$$\mathcal{D}^{\boldsymbol{c}(i \to \bar{c}_i)} := \{\boldsymbol{z}_{\boldsymbol{c}(i \to \bar{c}_i)}\} \cup \{\boldsymbol{z}_{\boldsymbol{c}}\} \cup \{\boldsymbol{z}_{\boldsymbol{c}(i \to \bar{c}_i, \ell \to \bar{c}_\ell)} : \ell \in [k] \setminus \{i\}\},$$

for any $\ell \neq i$ the counts are unchanged: $N_{\ell,c_\ell}(\mathcal{D}^{\boldsymbol{c}(i \to \bar{c}_i)}) = k$ and $N_{\ell,\bar{c}_\ell}(\mathcal{D}^{\boldsymbol{c}(i \to \bar{c}_i)}) = 1$, but for concept $i$ they swap: $N_{i,\bar{c}_i}(\mathcal{D}^{\boldsymbol{c}(i \to \bar{c}_i)}) = k$ and $N_{i,c_i}(\mathcal{D}^{\boldsymbol{c}(i \to \bar{c}_i)}) = 1$. Thus $\boldsymbol{z}_{\boldsymbol{c}}$ is now the unique minority example for concept $i$ (label $c_i$). We denote

$$\mathcal{Y}_2 = \mathcal{D}^{\boldsymbol{c}(i \to \bar{c}_i)} \setminus \{\boldsymbol{z}_{\boldsymbol{c}}\} \tag{17}$$

as the majority examples for concept $i$.

Let the pair of majority support vectors for $\mathcal{D}^{\boldsymbol{c}}$ and $\mathcal{D}^{\boldsymbol{c}(i \to \bar{c}_i)}$ be $\boldsymbol{y}_1$ and $\boldsymbol{y}_2$ respectively. By Lemma 4, we can write

$$\boldsymbol{y}_1 = \lambda_i \boldsymbol{z}_{\boldsymbol{c}} + \sum_{j \in [k] \setminus \{i\}} \lambda_j \boldsymbol{z}_{\boldsymbol{c}(j \to \bar{c}_j)} \quad \text{and} \quad \boldsymbol{y}_2 = \gamma_i \boldsymbol{z}_{\boldsymbol{c}(i \to \bar{c}_i)} + \sum_{j \in [k] \setminus \{i\}} \gamma_j \boldsymbol{z}_{\boldsymbol{c}(i \to \bar{c}_i, j \to \bar{c}_j)} \tag{18}$$

for some convex combinations $\lambda_j \geq 0$ with $\sum_j^k \lambda_j = 1$ and $\gamma_j \geq 0$ with $\sum_j^k \gamma_j = 1$.

Additionally, note that by Lemma 3 it holds for any point $\boldsymbol{z}_{\boldsymbol{c}'}$ for concept $j \in [k]$ that

$$\boldsymbol{w}_j^\intercal \boldsymbol{z}_{\boldsymbol{c}'} + b_j = y_j(\boldsymbol{c}'), \tag{19}$$

where we use a shorthand $y_j(\boldsymbol{c}) = 1$ if $c_j = 1$ and $y_j(\boldsymbol{c}) = -1$ otherwise.

Then, by Lemma 4 it holds that the support vectors are aligned with the shortest segment between the convex sets (pairs of $\boldsymbol{z}_{\boldsymbol{c}(i \to \bar{c}_i)}$ and $\boldsymbol{y}_1$, and $\boldsymbol{z}_{\boldsymbol{c}}$ and $\boldsymbol{y}_2$)

$$\boldsymbol{z}_{\boldsymbol{c}(i \to \bar{c}_i)} + y_i(\boldsymbol{c}) \frac{2}{||\boldsymbol{w}_i||^2} \boldsymbol{w}_i = \boldsymbol{y}_1 \quad \text{and} \quad \boldsymbol{z}_{\boldsymbol{c}} - y_i(\boldsymbol{c}) \frac{2}{||\boldsymbol{w}_i||^2} \boldsymbol{w}_i = \boldsymbol{y}_2, \tag{20}$$

where clearly $y_i(\boldsymbol{c}(i \to \bar{c}_i)) = -y_i(\boldsymbol{c})$. From this, it follows that

$$\boldsymbol{y}_1 - \boldsymbol{z}_{\boldsymbol{c}(i \to \bar{c}_i)} = \boldsymbol{z}_{\boldsymbol{c}} - \boldsymbol{y}_2. \tag{21}$$

Now, for any $j \neq i$ it follows that

$$\boldsymbol{w}_j^\intercal \boldsymbol{y}_1 + b_j = \boldsymbol{w}_j^\intercal \left( \lambda_i \boldsymbol{z}_{\boldsymbol{c}} + \sum_{l \in [k] \setminus \{i\}} \lambda_l \boldsymbol{z}_{\boldsymbol{c}(l \to \bar{c}_l)} \right) + b_j \tag{22}$$

$$= \lambda_i \boldsymbol{w}_j^\intercal \boldsymbol{z}_{\boldsymbol{c}} + \sum_{l \in [k] \setminus \{i\}} \lambda_l \boldsymbol{w}_j^\intercal \boldsymbol{z}_{\boldsymbol{c}(l \to \bar{c}_l)} + \sum_l^k \lambda_l b_j \tag{23}$$

$$= \lambda_i (\boldsymbol{w}_j^\intercal \boldsymbol{z}_{\boldsymbol{c}} + b_j) + \sum_{l \in [k] \setminus \{i\}} \lambda_l (\boldsymbol{w}_j^\intercal \boldsymbol{z}_{\boldsymbol{c}(l \to \bar{c}_l)} + b_j) \tag{24}$$

$$= \lambda_i y_j(\boldsymbol{c}) + \sum_{l \in [k] \setminus \{i,j\}} \lambda_l y_j(\boldsymbol{c}(l \to \bar{c}_l)) + \lambda_j y_j(\boldsymbol{c}(j \to \bar{c}_j)) \tag{25}$$

$$= \lambda_i y_j(\boldsymbol{c}) + \left( \sum_{l \in [k] \setminus \{i,j\}} \lambda_l \right) y_j(\boldsymbol{c}) - \lambda_j y_j(\boldsymbol{c}) \tag{26}$$

$$= (1 - \lambda_j) y_j(\boldsymbol{c}) - \lambda_j y_j(\boldsymbol{c}) = (1 - 2\lambda_j) y_j(\boldsymbol{c}), \tag{27}$$

where we used the fact that $\lambda$ are convex combinations in the second equality, and the fact that in the paired dataset $k$-concept values remain the same when flipping any other concept than $k$.

By repeating the same calculation as (22) for $\boldsymbol{y}_2$, we get:

$$\boldsymbol{w}_j^\mathsf{T} \boldsymbol{y}_2 + b_j = (1 - 2\gamma_j) y_j(\boldsymbol{c}). \tag{28}$$

By (21) it follows that (again with $j \neq i$)

$$
\begin{aligned}
& \boldsymbol{w}_j^\mathsf{T}(\boldsymbol{y}_1 - \boldsymbol{z}_{\boldsymbol{c}(i \to \bar{c}_i)}) = \boldsymbol{w}_j^\mathsf{T}(\boldsymbol{z}_{\boldsymbol{c}} - \boldsymbol{y}_2) \\
\Rightarrow\quad & \boldsymbol{w}_j^\mathsf{T}\boldsymbol{y}_1 + b_j - \boldsymbol{w}_j^\mathsf{T}\boldsymbol{z}_{\boldsymbol{c}(i \to \bar{c}_i)} - b_j = \boldsymbol{w}_j^\mathsf{T}\boldsymbol{z}_{\boldsymbol{c}} + b_j - \boldsymbol{w}_j^\mathsf{T}\boldsymbol{y}_2 - b_j \\
\Rightarrow\quad & (1 - 2\lambda_j) y_j(\boldsymbol{c}) - y_j(\boldsymbol{c}) = y_j(\boldsymbol{c}) - (1 - 2\gamma_j) y_j(\boldsymbol{c}) \\
\Rightarrow\quad & 1 - 2\lambda_j - 1 = 1 - 1 + 2\gamma_j \\
\Rightarrow\quad & \lambda_j + \gamma_j = 0.
\end{aligned}
\tag{29}
$$

Clearly, since $\lambda_j$ and $\gamma_j$ are convex combinations and thus non-negative, (29) implies that $\lambda_j = \gamma_j = 0$.

By repeating this process for all $j \neq i$, we get that $\lambda_j = \gamma_j = 0$ for all $j \neq i$, and therefore $\lambda_i = \gamma_i = 1$. From this, it follows that $\boldsymbol{y}_1 = \boldsymbol{z}_{\boldsymbol{c}}$ and $\boldsymbol{y}_2 = \boldsymbol{z}_{\boldsymbol{c}(i \to \bar{c}_i)}$.

This means that

$$\boldsymbol{z}_{\boldsymbol{c}(i \to \bar{c}_i)} + y_i(\boldsymbol{c}) \frac{2}{||\boldsymbol{w}_i||^2} \boldsymbol{w}_i = \boldsymbol{z}_{\boldsymbol{c}} \quad \text{and} \quad \boldsymbol{z}_{\boldsymbol{c}} - y_i(\boldsymbol{c}) \frac{2}{||\boldsymbol{w}_i||^2} \boldsymbol{w}_i = \boldsymbol{z}_{\boldsymbol{c}(i \to \bar{c}_i)}, \tag{30}$$

and therefore the differences between $\boldsymbol{z}_{\boldsymbol{c}} - \boldsymbol{z}_{\boldsymbol{c}(i \to \bar{c}_i)}$ are independent of other concept variations. Because of that, we can write any datapoint $\boldsymbol{z}_{\boldsymbol{c}}$ as a sum of concept-specific values $\boldsymbol{u}_{i,c_i}(c_i \in [2])$. For instance, if we fix $\boldsymbol{0} = (0, \dots, 0) \in [2]^k$, we can express $\boldsymbol{z}_{\boldsymbol{c}}$ as, for example (up to a global linear shift per concept)

$$\boldsymbol{u}_{i,0} = \boldsymbol{z}_{\boldsymbol{0}}/k, \quad \boldsymbol{u}_{i,1} = \boldsymbol{z}_{\boldsymbol{0}}/k + \frac{2}{||\boldsymbol{w}_i||^2} \boldsymbol{w}_i,$$

$$\boldsymbol{z}_{\boldsymbol{c}} = \sum_{i=1}^{k} \boldsymbol{u}_{i,c_i}, \tag{31}$$

from here, we can write any datapoint $\boldsymbol{z}_{\boldsymbol{c}}$ as

$$\boldsymbol{z}_{\boldsymbol{c}} = \sum_{i=1}^{k} \boldsymbol{u}_{i,c_i} = \sum_{i \in [k]:c_i=0} \boldsymbol{z}_{\boldsymbol{0}}/k + \sum_{i \in [k]:c_i=1} (\boldsymbol{z}_{\boldsymbol{0}}/k + \frac{2}{||\boldsymbol{w}_i||^2} \boldsymbol{w}_i) = \boldsymbol{z}_{\boldsymbol{0}} + \sum_{i \in [k]:c_i=1} \frac{2}{||\boldsymbol{w}_i||^2} \boldsymbol{w}_i, \tag{32}$$

which establishes linearity.

**Orthogonality.** First, note that by invariance (Lemma 3) it holds that for any concept $i$, changes in concept values other than $i$ do not affect the prediction of concept $i$:

$$\boldsymbol{w}_i^\mathsf{T} \boldsymbol{z}_{\boldsymbol{c}} + b_i = \boldsymbol{w}_i^\mathsf{T} \boldsymbol{z}_{\boldsymbol{c}(j \to \bar{c}_j)} + b_i \tag{33}$$

But by linear factorization (31) it follows that

$$
\begin{aligned}
& \boldsymbol{w}_i^\mathsf{T} \boldsymbol{z}_{\boldsymbol{c}} + b_i = \boldsymbol{w}_i^\mathsf{T} \boldsymbol{z}_{\boldsymbol{c}(j \to \bar{c}_j)} + b_i \\
\Rightarrow\quad & \boldsymbol{w}_i^\mathsf{T}(\boldsymbol{z}_{\boldsymbol{c}} - \boldsymbol{z}_{\boldsymbol{c}(j \to \bar{c}_j)}) = 0 \\
\Rightarrow\quad & \boldsymbol{w}_i^\mathsf{T}\left(\boldsymbol{u}_{j,c_j} - \boldsymbol{u}_{j,\bar{c}_j}\right) = 0 \\
\Rightarrow\quad & \boldsymbol{w}_i^\mathsf{T}\left(\frac{2}{||\boldsymbol{w}_j||^2} \boldsymbol{w}_j\right) = 0 \\
\Rightarrow\quad & \boldsymbol{w}_i^\mathsf{T}\boldsymbol{w}_j = 0.
\end{aligned}
\tag{34}
$$

Then,

$$(\boldsymbol{u}_{i,c_i} - \boldsymbol{u}_{i,\bar{c}_i})^\mathsf{T}(\boldsymbol{u}_{j,c_j} - \boldsymbol{u}_{j,\bar{c}_j}) \propto \boldsymbol{w}_i^\mathsf{T}\boldsymbol{w}_j = 0. \tag{35}$$

More generally, orthogonality of one concept holds against the span of other concepts as well. For $\{\alpha_j \in \mathbb{R}\}_{j \neq i}$ it follows that

$$(\boldsymbol{u}_{i,c_i} - \boldsymbol{u}_{i,\bar{c}_i})^\mathsf{T} \left(\sum_{j \neq i} \alpha_j(\boldsymbol{u}_{j,c_j} - \boldsymbol{u}_{j,\bar{c}_j})\right) \propto \boldsymbol{w}_i^\mathsf{T} \left(\sum_{j \neq i} \alpha_j \boldsymbol{w}_j\right) = 0, \tag{36}$$

and therefore orthogonality holds against the span of other concepts differences. $\qquad\square$

# C. Sufficiency of linear factorization for compositional generalization

This section complements the main text's necessity results by showing a converse: under the same linear-head setting, linearly factored representations are sufficient to obtain compositional generalization under our desiderata.

We discuss two cases. First, in the binary setting, we show that linear factorization together with cross-concept orthogonality makes stable transfer automatic for GD+CE, even from small but diverse training supports (Appendix C.1). Second, in the general multi-valued setting we discuss the general case when the underlying features are linear, but not necessarily orthogonal across concepts, and the learning algorithm is not necessarily GD+CE (Appendix C.2). There, training with GD+CE does not necessarily satisfy the desiderata, but in principle classifiers can be constructed that do, precisely because the factors can be recovered.

**Proposition 3** (Sufficiency summary (informal)). Under linear readouts, two sufficiency statements hold.

1. *Binary necessity–sufficiency case.* Under the binary grid with GD+CE, if embeddings decompose as $z_c = \sum_i u_{i,c_i}$ with cross-concept orthogonality, then any training set with $|T| = 2^{k-1} + 1$ (or a cross dataset of size $1 + k$) suffices: the learned readout recovers every concept value on the full grid (transferability) and is invariant across valid $T$ (stability). Combined with Proposition 1, the geometry is both necessary and sufficient (Proposition 4).

2. *General constructive case.* In the multi-valued case, recoverable linear factors are sufficient to construct concept readouts in principle (Appendix C.2).

We believe that only the first case is of practical interest, since it assumes standard GD+CE training; together with the necessary conditions, it reinforces that linear, cross-concept orthogonal structure is a plausible target that current vision(-language) systems *should* exhibit if they are to generalize compositionally.

## C.1. Binary valued-case: sufficiency of SVMs

In this section, we show that linear factorization together with cross-concept orthogonality makes stable transfer automatic for GD+CE, even from small but diverse training supports. Since concepts are binary-valued, we can represent the readout for each concept by a single affine separator with parameters $(w_i, b_i)$, and we work with these parameters throughout (in the same way as Appendix B).

**Proposition 4** (Linear factorization implies compositional generalization). Consider the binary grid $\mathcal{C}_i = \{0, 1\}$ with representations $\mathcal{Z} = \{z_c : c \in [2]^k\} \subset \mathbb{R}^d$. Assume there exist $\{u_{i,0}, u_{i,1} \in \mathbb{R}^d\}_{i=1}^k$ such that for every $c \in [2]^k$:

1. *(Linearity)* $z_c = \sum_{i=1}^k u_{i,c_i}$.

2. *(Cross-concept orthogonality)* $(u_{i,1} - u_{i,0}) \perp (u_{j,1} - u_{j,0})$ for all $i \neq j$.

Then for any training set $T \subseteq [2]^k$ satisfying either of the following conditions:

1. Any set with $|T| = 2^{k-1} + 1$, or
2. A cross dataset $\mathcal{D}^c$ (i.e. the center $c$ and all one-value flips from $c$, Definition 6) with $|T| = 1 + k$.

It holds that gradient descent + cross-entropy loss trained on $T$ satisfy the desiderata on the entire grid: they recover every concept value on all $z_c$ (transferability) and are invariant across valid $T$ (stability).

*Proof.* Throughout the proof we use the standard result that the GD+CE converges to the max-margin SVM solution in the binary-class case (Soudry et al., 2024) and interpret the optimal solution as a weight vector $w_i$ and bias $b_i$ that separates the two classes.

**Full dataset case.**

First, we establish the exact weights produced by the linear probes.

For that, note that for any concept $i \in [k]$, the SVM solution is proportional to the shortest segment between the convex hulls of the points in the two classes, denoted side by side as

$$\mathcal{Y}_- := \{z_c : c \in [2]^k, \ c_i = 0\}, \qquad \mathcal{Y}_+ := \{z_c : c \in [2]^k, \ c_i = 1\}. \tag{37}$$

with the proportionality constant $\frac{2}{\|w_i\|^2}$ (Bennett & Bredensteiner, 2000). Equivalently, there exist convex combinations $y_- \in \mathrm{conv}(\mathcal{Y}_-)$ and $y_+ \in \mathrm{conv}(\mathcal{Y}_+)$ such that $w_i$ is parallel to the shortest segment $y_+ - y_-$, where

$$y_- = \sum_{c \in [2]^k, c_i = 0} \gamma_c z_c, \qquad y_+ = \sum_{c \in [2]^k, c_i = 1} \lambda_c z_c, \qquad \lambda, \gamma \succeq 0, \qquad \sum_{c \in [2]^k, c_i = 1} \lambda_c = 1, \qquad \sum_{c \in [2]^k, c_i = 0} \gamma_c = 1. \tag{38}$$

Importantly, these convex combinations are the support vectors of the SVM solution and correspond to the shortest distance between the two classes. To proceed, we first write the difference between any two points in the convex hulls and then lower bound the norm of the difference and arrive at the exact weight vector.

Note that for any $\boldsymbol{y}_+$ and $\boldsymbol{y}_-$ their difference can be written as:

$$\boldsymbol{y}_+ - \boldsymbol{y}_- = \sum_{\boldsymbol{c} \in [2]^k : c_i = 1} \lambda_{\boldsymbol{c}} \boldsymbol{z}_{\boldsymbol{c}} - \sum_{\boldsymbol{c} \in [2]^k : c_i = 0} \gamma_{\boldsymbol{c}} \boldsymbol{z}_{\boldsymbol{c}} \tag{40}$$

$$= \sum_{\boldsymbol{c} \in [2]^k : c_i = 1} \lambda_{\boldsymbol{c}} \left( \sum_{j \neq i} \boldsymbol{u}_{j,c_j} + \boldsymbol{u}_{i,1} \right) - \sum_{\boldsymbol{c} \in [2]^k : c_i = 0} \gamma_{\boldsymbol{c}} \left( \sum_{j \neq i} \boldsymbol{u}_{j,c_j} + \boldsymbol{u}_{i,0} \right) \tag{41}$$

$$= \sum_{\boldsymbol{c} \in [2]^k : c_i = 1} \lambda_{\boldsymbol{c}} \boldsymbol{u}_{i,1} - \sum_{\boldsymbol{c} \in [2]^k : c_i = 0} \gamma_{\boldsymbol{c}} \boldsymbol{u}_{i,0} + \sum_{\boldsymbol{c} \in [2]^k : c_i = 1} \lambda_{\boldsymbol{c}} \sum_{j \neq i} \boldsymbol{u}_{j,c_j} - \sum_{\boldsymbol{c} \in [2]^k : c_i = 0} \gamma_{\boldsymbol{c}} \sum_{j \neq i} \boldsymbol{u}_{j,c_j} \tag{42}$$

$$= \boldsymbol{u}_{i,1} - \boldsymbol{u}_{i,0} + \sum_{\boldsymbol{c} \in [2]^k : c_i = 1} \lambda_{\boldsymbol{c}} \sum_{j \neq i} \boldsymbol{u}_{j,c_j} - \sum_{\boldsymbol{c} \in [2]^k : c_i = 0} \gamma_{\boldsymbol{c}} \sum_{j \neq i} \boldsymbol{u}_{j,c_j}, \tag{44}$$

where the last equality uses the fact that the sum runs over all concept combinations, and therefore sums to 1. Note that the last term in (44) can be written as

$$\sum_{\boldsymbol{c} \in [2]^k : c_i = 1} \lambda_{\boldsymbol{c}} \sum_{j \neq i} \boldsymbol{u}_{j,c_j} - \sum_{\boldsymbol{c} \in [2]^k : c_i = 0} \gamma_{\boldsymbol{c}} \sum_{j \neq i} \boldsymbol{u}_{j,c_j} \tag{45}$$

$$= \sum_{\boldsymbol{c} \in [2]^k : c_i = 1} (\lambda_{\boldsymbol{c}} - \gamma_{\boldsymbol{c}(i \to 0)}) \left( \sum_{j \neq i} \boldsymbol{u}_{j,c_j} \right) \tag{46}$$

$$= \sum_{j \neq i} \sum_{\boldsymbol{c} \in [2]^k : c_i = 1} (\lambda_{\boldsymbol{c}} - \gamma_{\boldsymbol{c}(i \to 0)}) \boldsymbol{u}_{j,c_j} \tag{47}$$

$$= \sum_{j \neq i} \left( \left( \sum_{\boldsymbol{c} \in [2]^k : c_i = 1, c_j = 1} (\lambda_{\boldsymbol{c}} - \gamma_{\boldsymbol{c}(i \to 0)}) \right) \boldsymbol{u}_{j,1} + \left( \sum_{\boldsymbol{c} \in [2]^k : c_i = 1, c_j = 0} (\lambda_{\boldsymbol{c}} - \gamma_{\boldsymbol{c}(i \to 0)}) \right) \boldsymbol{u}_{j,0} \right), \tag{48}$$

but since $\sum_{\boldsymbol{c} \in [2]^k : c_i = 1} (\lambda_{\boldsymbol{c}} - \gamma_{\boldsymbol{c}(i \to 0)}) = 0$, it follows that for any $j \neq i$,

$$\sum_{\boldsymbol{c} \in [2]^k : c_i = 1, c_j = 1} (\lambda_{\boldsymbol{c}} - \gamma_{\boldsymbol{c}(i \to 0)}) + \sum_{\boldsymbol{c} \in [2]^k : c_i = 1, c_j = 0} (\lambda_{\boldsymbol{c}} - \gamma_{\boldsymbol{c}(i \to 0)}) = 0 \tag{49}$$

$$\Rightarrow \sum_{\boldsymbol{c} \in [2]^k : c_i = 1, c_j = 1} (\lambda_{\boldsymbol{c}} - \gamma_{\boldsymbol{c}(i \to 0)}) = - \sum_{\boldsymbol{c} \in [2]^k : c_i = 1, c_j = 0} (\lambda_{\boldsymbol{c}} - \gamma_{\boldsymbol{c}(i \to 0)}) \tag{50}$$

We thus denote $\Delta_j := \sum_{\boldsymbol{c} \in [2]^k : c_i = 1, c_j = 1} (\lambda_{\boldsymbol{c}} - \gamma_{\boldsymbol{c}(i \to 0)})$. The full expression of (40) can be written compactly as

$$\boldsymbol{y}_+ - \boldsymbol{y}_- = \boldsymbol{u}_{i,1} - \boldsymbol{u}_{i,0} + \sum_{j \neq i} \Delta_j \boldsymbol{u}_{j,1} - \Delta_j \boldsymbol{u}_{j,0} \tag{51}$$

$$= \boldsymbol{u}_{i,1} - \boldsymbol{u}_{i,0} + \sum_{j \neq i} \Delta_j (\boldsymbol{u}_{j,1} - \boldsymbol{u}_{j,0}). \tag{52}$$

Recall that by assumption, $\boldsymbol{u}_{i,1} - \boldsymbol{u}_{i,0} \perp \boldsymbol{u}_{j,1} - \boldsymbol{u}_{j,0}$ for all $j \neq i$, so it follows that

$$(\boldsymbol{u}_{i,1} - \boldsymbol{u}_{i,0})^\top \left( \sum_{j \neq i} \Delta_j (\boldsymbol{u}_{j,1} - \boldsymbol{u}_{j,0}) \right) = \sum_{j \neq i} \Delta_j (\boldsymbol{u}_{i,1} - \boldsymbol{u}_{i,0})^\top (\boldsymbol{u}_{j,1} - \boldsymbol{u}_{j,0}) = 0. \tag{53}$$

Therefore, $\boldsymbol{u}_{i,1} - \boldsymbol{u}_{i,0} \perp \sum_{j \neq i} \Delta_j (\boldsymbol{u}_{j,1} - \boldsymbol{u}_{j,0})$. This allows us to apply the Pythagorean theorem when computing the distance between $\boldsymbol{y}_+$ and $\boldsymbol{y}_-$:

$$\|\boldsymbol{y}_+ - \boldsymbol{y}_-\|^2 = \|\boldsymbol{u}_{i,1} - \boldsymbol{u}_{i,0}\|^2 + \left\| \sum_{j \neq i} \Delta_j (\boldsymbol{u}_{j,1} - \boldsymbol{u}_{j,0}) \right\|^2 \tag{54}$$

$$\geq \|\boldsymbol{u}_{i,1} - \boldsymbol{u}_{i,0}\|^2. \tag{55}$$

Thus, any two points in their respective convex hulls are at least as far apart as the distance between $\boldsymbol{u}_{i,1} - \boldsymbol{u}_{i,0}$. We make use of this result in computing the SVM solution by picking two points $\boldsymbol{y}_+$ and $\boldsymbol{y}_-$ that have exactly the shortest possible distance between them, thus

establishing them as the support vectors. Conveniently, any two "counterfactual" points do: for any $c \in [2]^k$ by picking $y_+ = z_{c(i \to 1)}$ and $y_- = z_{c(i \to 0)}$, we have that

$$\|z_{c(i \to 1)} - z_{c(i \to 0)}\|^2 = \|u_{i,1} - u_{i,0}\|^2.$$

Since this computation is independent of the particular choice of the concept $i$, it holds for any concept. As such, we can write the weight vector $w_i$ as

$$w_i = \frac{2}{\|u_{i,1} - u_{i,0}\|^2}(u_{i,1} - u_{i,0}). \tag{56}$$

To show that classification works, note that

$$z_c = \sum_{j=1}^{k} \left(\widetilde{u}_{j,c_j} + \bar{u}_j\right) = \sum_{j} \widetilde{u}_{j,c_j} + \sum_{j} \bar{u}_j, \tag{57}$$

where $\widetilde{u}_{j,c_j} := u_{j,c_j} - \bar{u}_j$ is the centered factor for concept $j$ for the value $c_j$, and $\bar{u}_j := \frac{1}{2}(u_{j,1} + u_{j,0})$ is the average of the two concept factors.

But the centered factors must sum to zero, so $\widetilde{u}_{j,0} = -\widetilde{u}_{j,1}$. Therefore for any $i \neq j$

$$(u_{i,1} - u_{i,0})^\intercal(u_{j,1} - u_{j,0}) = 0$$
$$\Rightarrow \quad (\widetilde{u}_{i,1} - \widetilde{u}_{i,0})^\intercal(\widetilde{u}_{j,1} - \widetilde{u}_{j,0}) = 0$$
$$\Rightarrow \quad \widetilde{u}_{i,1}^\intercal \widetilde{u}_{j,1} = 0. \tag{58}$$

By evaluating the classification rule at any $z_c$ for any concept $i$ we get

$$w_i^\intercal z_c + b_i = w_i^\intercal \left(\sum_{j=1}^{k}\left(\widetilde{u}_{j,c_j} + \bar{u}_j\right)\right) + b_i \tag{59}$$

$$= w_i^\intercal \sum_{j=1}^{k} \widetilde{u}_{j,c_j} + w_i^\intercal \sum_{j=1}^{k} \bar{u}_j + b_i, \tag{60}$$

but note that only the first term of (60) is affected by the concept $i$, and the following terms can be absorbed into the bias $b_i{}^2$. Thus, the classification is correct only if $\text{sign}\left(w_i^\intercal \sum_{j=1}^{k} \widetilde{u}_{j,c_j}\right) = 2c_i - 1$. By (56) it follows that

$$\text{sign}\left(w_i^\intercal \sum_{j=1}^{k} \widetilde{u}_{j,c_j}\right) = \text{sign}\left((2\widetilde{u}_{i,1})^\intercal \widetilde{u}_{i,c_i}\right) = \text{sign}\left(\widetilde{u}_{i,1}^\intercal \widetilde{u}_{i,c_i}\right) = 2c_i - 1. \tag{61}$$

And therefore the classification is correct.

**(1)** $|T| = 2^{k-1} + 1$ **case.** The main idea of the proof is that for any concept $i \in [k]$ and a pair of "counterfactual" points $z_c, z_{c(i \to \bar{c}_i)}$ it holds that the distance between the two points is the shortest possible distance between any two points in the convex hulls of the two classes and is equal to $\|u_{i,1} - u_{i,0}\|^2$. This follows the same argument as specified in the full dataset case (and in (56) in particular). All that is needed to be shown then, is the availability of such a pair of points in any training set with $|T| = 2^{k-1} + 1$.

This can be argued by contradiction. Assume that for some concept $i \in [k]$ there are no two points $(c, c')$ with $c_j = c'_j$ for all $j \neq i$ and $c_i \neq c'_i$. There are $2^{k-1}$ such pairs of points in total. Thus a dataset would have to have at most $2^k - 2^{k-1} = 2^{k-1}$ points, which contradicts the assumption that $|T| = 2^{k-1} + 1$. Therefore, such a pair of points must exist. Since this argument is independent of the particular choice of the concept $i$, it holds for any concept. Therefore the SVM solution will always be able to find such a pair of points that minimize the distance between the two classes (as per (54)) and will transfer as well as yield the same weight vector as the full dataset case.

**(2)** $\mathcal{D}^c$ **case with** $|T| = 1 + k$**.** By construction for any concept $i$ there exists "counterfactual" point from the center $c$ to $c(i \to \bar{c}_i)$. By following the same argument detailed in the full dataset case as well as the previous case, it follows that the SVM solution will transfer as well as yield the same weight vector as the full dataset case. In both cases stability of the solution follows due to recovering the same weight and bias vectors. $\square$

---

$^2$Alternatively, assume that $z_c$ is zero-centered and use the same argument.

## C.2. General case: linearly factored embeddings and sufficiency of recovering the factors

We provide a complementary analysis on the sufficient conditions for generalizing compositionally. Here, we detail the key results for recovering the factors $u$ from representations that already possess linear factorization.

We first note the minimal dataset setting using the notion of a cross dataset, defined below.

**Lemma 5** (Uniqueness up to concept-wise shifts). Let the concept space be $\mathcal{C} = \mathcal{C}_1 \times \cdots \times \mathcal{C}_k$. Assume two factor families $\{u_{i,j}\}_{i \in [k],\, j \in \mathcal{C}_i}$ and $\{v_{i,j}\}_{i \in [k],\, j \in \mathcal{C}_i}$ satisfy, for every $c \in \mathcal{C}$,

$$z_c = \sum_{i=1}^{k} u_{i,c_i} = \sum_{i=1}^{k} v_{i,c_i}.$$

Then there exist vectors $s_1, \ldots, s_k \in \mathbb{R}^d$ with

$$\sum_{i=1}^{k} s_i = 0$$

such that

$$v_{i,j} = u_{i,j} + s_i, \qquad \forall i \in [k],\ j \in \mathcal{C}_i.$$

Hence the factors are identifiable only up to concept-wise shifts.

*Proof.* For each concept $i \in [k]$ and value $j \in \mathcal{C}_i$, define the vector difference

$$\delta_{i,j} := v_{i,j} - u_{i,j}.$$

We first show that, within each concept $i$, this difference is independent of the value. Fix $i \in [k]$ and pick any two values $p, q \in \mathcal{C}_i$. Take any $c \in \mathcal{C}$ with $c_i = p$. Using the counterfactual tuple $c(i \to q)$ and subtracting the two factorizations gives

$$z_c - z_{c(i \to q)} = \sum_{\ell=1}^{k} u_{\ell,c_\ell} - \sum_{\ell=1}^{k} u_{\ell,(c(i \to q))_\ell}$$

$$= \sum_{\ell=1}^{k} v_{\ell,c_\ell} - \sum_{\ell=1}^{k} v_{\ell,(c(i \to q))_\ell}.$$

All terms except concept $i$ cancel on both sides, so

$$u_{i,p} - u_{i,q} = v_{i,p} - v_{i,q}, \qquad \Rightarrow \qquad \delta_{i,p} = \delta_{i,q}.$$

Hence, for each concept $i$, there is a single vector $s_i \in \mathbb{R}^d$ such that

$$\delta_{i,j} = s_i, \qquad \forall j \in \mathcal{C}_i.$$

Therefore $v_{i,j} = u_{i,j} + s_i$ for all $i, j$.

To obtain the zero-sum constraint, evaluate at any $c \in \mathcal{C}$:

$$0 = \sum_{i=1}^{k} \left( v_{i,c_i} - u_{i,c_i} \right) = \sum_{i=1}^{k} \delta_{i,c_i} = \sum_{i=1}^{k} s_i.$$

Conversely, if $s_1, \ldots, s_k \in \mathbb{R}^d$ satisfy $\sum_{i=1}^{k} s_i = 0$ and we set $v_{i,j} := u_{i,j} + s_i$, then for every $c \in \mathcal{C}$,

$$\sum_{i=1}^{k} v_{i,c_i} = \sum_{i=1}^{k} u_{i,c_i} + \sum_{i=1}^{k} s_i = \sum_{i=1}^{k} u_{i,c_i},$$

so the reconstructed factors generate the same embeddings. $\qquad\square$

We illustrate this lemma graphically in Figure 12.

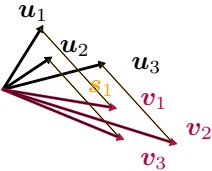

*Figure 12.* **Illustration of the shift ambiguity in the factorization equations.** The black factors $\{u_i\}$ and pink factors $\{v_i\}$ produce the same pairwise differences; the orange arrows show concept-wise shift vectors $s_i$ such that $v_i = u_i + s_i$.

A neat consequence is that centered factors are uniquely determined: any recovered factorization, once centered per concept, matches the true centered factors.

**Corollary 1** (Uniqueness of centered factors). Assume the setting of Lemma 5. For each concept $i$, let

$$\mu_i := \frac{1}{|\mathcal{C}_i|} \sum_{j \in \mathcal{C}_i} u_{i,j},$$

$$\nu_i := \frac{1}{|\mathcal{C}_i|} \sum_{j \in \mathcal{C}_i} v_{i,j},$$

and define the centered factors $u_{i,j}^{\circ} := u_{i,j} - \mu_i$ and $v_{i,j}^{\circ} := v_{i,j} - \nu_i$. Then $u_{i,j}^{\circ} = v_{i,j}^{\circ}$ for all $i \in [k]$ and $j \in \mathcal{C}_i$. Equivalently, for every $c \in \mathcal{C}$,

$$\sum_{i=1}^{k} u_{i,c_i}^{\circ} = \sum_{i=1}^{k} v_{i,c_i}^{\circ},$$

so the centered factorization is unique.

*Proof.* By Lemma 5, there exist $s_1, \ldots, s_k$ with $\sum_i s_i = 0$ such that $v_{i,j} = u_{i,j} + s_i$ for all $i, j$. Averaging over $j \in \mathcal{C}_i$ yields $\nu_i = \mu_i + s_i$. Thus,

$$
\begin{aligned}
v_{i,j}^{\circ} &= v_{i,j} - \nu_i \\
&= (u_{i,j} + s_i) - (\mu_i + s_i) \\
&= u_{i,j} - \mu_i \\
&= u_{i,j}^{\circ},
\end{aligned}
$$

as claimed. $\square$

First, we consider the general case where concept directions are not necessarily linearly independent. Suppose the inputs $z_c$ are linearly separable for any $i \in [k], j \in [n]$. If we can recover all factors $v_{i,j}$, we can reconstruct $z_c = \sum_{i=1}^{k} u_{i,c_i}$ and then fit linear probes for the concept values.

By Lemma 5 (and the centered-factor corollary above), recovery is unique up to concept-wise shifts, so the key requirement is rank of the one-hot design matrix induced by observed tuples.

**Proposition 5** (Maximum possible rank of the design matrix). Let $A \in \{0,1\}^{n^k \times kn}$ be the design matrix whose $kn$ columns are $\{a_{i,r} : i = 1, \ldots, k, \ r = 1, \ldots, n\}$, arranged in $k$ blocks of size $n$, with all $n^k$ treatment combinations as rows and each row having exactly one 1 in each block. Then,

$$\mathrm{rank}(A) = 1 + k(n - 1).$$

*Proof.* We first show the upper bound by spanning columns. Define $\mathbf{1} \in \mathbb{R}^{n^k}$ as the all-ones vector and, for each block $i$ and each $r = 2, \ldots, n$,

$$d_{i,r} := a_{i,r} - a_{i,1}.$$

Let

$$\mathcal{B} := \{\mathbf{1}\} \cup \{d_{i,r} : 1 \leq i \leq k, \ 2 \leq r \leq n\}, \quad \text{so} \quad |\mathcal{B}| = 1 + k(n - 1).$$

For every block $i$, $\sum_{r=1}^{n} a_{i,r} = \mathbf{1}$, hence

$$\sum_{r=2}^{n} d_{i,r} = \mathbf{1} - n a_{i,1} \implies a_{i,1} = \frac{1}{n}\left(\mathbf{1} - \sum_{r=2}^{n} d_{i,r}\right), \quad a_{i,r} = a_{i,1} + d_{i,r} \ (r \geq 2).$$

Thus every original column $\boldsymbol{a}_{i,r}$ lies in span $\mathcal{B}$, so $\mathrm{rank}(\boldsymbol{A}) \leq 1 + k(n-1)$.

For the matching lower bound, note that $\boldsymbol{A}^\mathsf{T}$ is exactly the on-off matrix from Proposition 6 with $\alpha = 1, \beta = 0$ (rows indexed by $(i, r)$ and columns by tuples $\boldsymbol{c}$). The rank argument in that proof applies verbatim and gives $\mathrm{rank}(\boldsymbol{A}^\mathsf{T}) = 1 + k(n-1)$. Hence

$$\mathrm{rank}(\boldsymbol{A}) = \mathrm{rank}(\boldsymbol{A}^\mathsf{T}) = 1 + k(n-1).$$

$\square$

When $\mathrm{rank}(\boldsymbol{A}) = 1 + k(n-1)$, the linear system $\boldsymbol{Z} = \boldsymbol{AU}$ determines the centered factors uniquely (up to the shift ambiguity already characterized above). Therefore one can reconstruct the grid embeddings and, under linear separability, fit linear readouts that recover concept values on all combinations (Definition 3).

# D. Packing and minimum dimension (proof of Proposition 2)

In this section, we elaborate on the geometric capacity question introduced in the main text: how large must the embedding dimension $d$ be for linear probes to realize all concept combinations in $\mathcal{C}$.

We use hyperplane-arrangement counting bounds to formalize this. Intuitively, each concept contributes $n$ decision boundaries, and to classify all combinations these boundaries must carve enough regions in $\mathbb{R}^d$ to accommodate all $n^k$ combinations. We first recall the relevant region-count results, then apply them to prove the lower bound $d \geq k$.

For $d \geq 1$, we work with affine hyperplanes (the setting used by linear probes with bias):

$$H_{\boldsymbol{w},b} = \{\, \boldsymbol{x} \in \mathbb{R}^d : \boldsymbol{w}^\mathsf{T}\boldsymbol{x} + b = 0 \,\}, \qquad \boldsymbol{w} \neq 0,\ b \in \mathbb{R}.$$

An *arrangement* $\mathcal{H} = \{H_1, \ldots, H_m\}$ is a family of hyperplanes. It is in *general position* when no $d+1$ hyperplanes intersect at a common point, which maximizes the number of connected regions in $\mathbb{R}^d$ (Zaslavsky, 1975; Ziegler, 1995).

This allows us to count the number of connected regions in $\mathbb{R}^d$ that an arrangement of $m$ hyperplanes carves out, a classical result that we state below.

**Theorem 1** (Zaslavsky's region bounds in general position (Zaslavsky, 1975))**.** Let $\mathcal{H}$ be an arrangement of $m$ hyperplanes in $\mathbb{R}^d$ that is in general position. Then, the number of connected regions $R(\mathcal{H})$ is given by:

(a) *Affine (with a bias) case.* If the hyperplanes may carry arbitrary offsets $b_i$ (so $\mathcal{H}$ is not required to be central), then

$$R(\mathcal{H}) = R_{\mathrm{aff}}(m, d) := \sum_{r=0}^{d} \binom{m}{r}.$$

(b) *Central case.* If every hyperplane passes through the origin,

$$R(\mathcal{H}) = R_{\mathrm{lin}}(m, d) := 2\sum_{r=0}^{d-1} \binom{m-1}{r}.$$

Note that for $d < k$, one has

$$R_{\mathrm{aff}}(k, d) = \sum_{r=0}^{d} \binom{k}{r} < 2^k,$$

which is the key inequality we will need. We now exploit Theorem 1 to prove the lower bound on probe dimension; first for the binary case, then for general $n$.

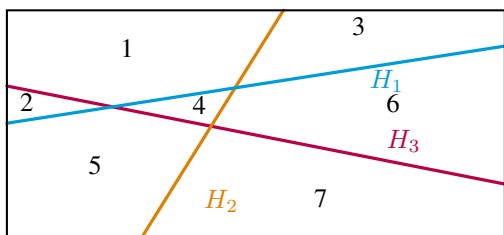
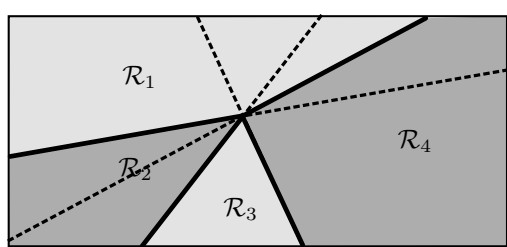

3 concepts, 2 values per concept, $2D$ space.     Reduction to two values

*Figure 13.* **Illustration of the probe dimension lower bound.** Schematic of probe hyperplane arrangements and induced regions in embedding space. **(left)** For $k = 3$ in $d = 2$, three affine hyperplanes partition the plane into at most 7 regions, fewer than the $2^3 = 8$ binary concept combinations; therefore $d \geq 3$. **(right)** For $n > 2$, restricting each concept to any two values induces a binary subproblem. Hence any model that realizes all $n^k$ combinations must also realize these binary restrictions, so the binary lower bound applies to the non-binary case as well.

**Proposition 2** (Minimum dimension for linear probes)**.** For $k$ concepts, each with $n$ values, suppose there exist linear probes that correctly classify each concept value for all $n^k$ combinations from embeddings $f(\boldsymbol{x}) \in \mathbb{R}^d$. Then necessarily $d \geq k$.

*Proof sketch.* Reduce to the binary case by fixing two values per concept and restricting to the resulting $2^k$ combinations. Each concept induces one affine separating hyperplane. To realize all binary labelings, the arrangement must carve at least $2^k$ regions. By Theorem 1, when $d < k$ we have $\sum_{r=0}^{d} \binom{k}{r} < 2^k$, so $2^k$ regions are impossible. Hence $d \geq k$. Tightness follows by a $d = k$ construction (coordinates per concept with suitable affine offsets). See Fig. 13.

*Proof.* First, we "reduce" the problem to the binary case. For each concept $i \in [k]$, take two distinct $a_i, b_i \in [n]$ and restrict the label space to

$$\mathcal{C}^{(2)} := \prod_{i=1}^{k} \{a_i, b_i\}, \qquad |\mathcal{C}^{(2)}| = 2^k.$$

Since every label combination in $[n]^k$ is classified correctly, all tuples in $\mathcal{C}^{(2)}$ are classified as well. For each concept $i$, we can define the induced binary score on only the considered values

$$h_i^{(a_i, b_i)}(\boldsymbol{z}) := h_{i,a_i}(\boldsymbol{z}) - h_{i,b_i}(\boldsymbol{z}) = (\boldsymbol{w}_{i,a_i} - \boldsymbol{w}_{i,b_i})^\mathsf{T} \boldsymbol{z} + (b_{i,a_i} - b_{i,b_i}). \tag{62}$$

Let $\boldsymbol{w}_i := \boldsymbol{w}_{i,a_i} - \boldsymbol{w}_{i,b_i}$ and $\beta_i := b_{i,a_i} - b_{i,b_i}$. Each induced binary classifier defines an affine hyperplane

$$\mathcal{H}_i := \{\boldsymbol{z} \in \mathbb{R}^d \mid \boldsymbol{w}_i^\mathsf{T} \boldsymbol{z} + \beta_i = 0\}. \tag{63}$$

Because multiclass predictions are correct on $\mathcal{C}^{(2)}$, each concept $i$ is correctly separated between values $a_i$ and $b_i$ on this restricted set. Hence the $k$ affine hyperplanes $\mathcal{H}_1, \dots, \mathcal{H}_k$ must separate $\mathbb{R}^d$ into at least $2^k$ distinct regions.

But the number of regions formed by $k$ affine hyperplanes in $\mathbb{R}^d$ is at most

$$\sum_{r=0}^{d} \binom{k}{r} < 2^k \quad \text{whenever } d < k \quad \text{(by Theorem 1)}. \tag{64}$$

Thus, we must have $d \geq k$.

Construction is simple. Let $d = k$, and assume (for $\boldsymbol{e}_i \in \mathbb{R}^d$ with $\boldsymbol{e}_{ij} = \delta_{ij}$)

$$\boldsymbol{z_c} := \sum_{i}^{k} \boldsymbol{e}_i c_i, \tag{65}$$

i.e. each embedding is a sum of standard basis unit vectors for each concept scaled by the value of that concept.

Then we can define probe vectors as

$$\boldsymbol{w}_{i,j} := 2j\,\boldsymbol{e}_i \quad \text{and} \quad b_{i,j} := -j^2. \tag{66}$$

Then

$$h_{i,j}(\boldsymbol{z_c}) = 2j\boldsymbol{e}_i^\mathsf{T} \boldsymbol{z_c} - j^2 = 2c_i - j^2 = c_i^2 - (j - c_i)^2. \tag{67}$$

Therefore, for each $i$, $\arg\max_{j \in [n]} h_{i,j}(\boldsymbol{z_q}) = q_i$, so all $n^k$ combinations are correctly classified in $d = k$. $\qquad \square$

For intuition of the constructed case, see Appendix E.1.

# E. Examples of linearly compositional models and their geometries

We give two toy geometries that make probe behavior explicit under strong linear-factorization assumptions. They can be understood as lying on two extremes of the space of linear representations: a case of "perfect LRH" geometry, where all the factors of concept values are co-linear (Appendix E.1), and a case where factors are implied to be orthogonal to any other factor, leading to a maximum possible dimensionality of linearly factorized spaces (Appendix E.2). These are *not* claims of full compositional generalization (they do not model the learning algorithm or stability), but provide intuition for probe geometry and dimensionality, especially in the cases where the classification behavior is intuitively desirable (Appendix E.2).

## E.1. Case 1: Ideal "LRH" concept classifier

To gain intuition into the geometry of features and linear probes, we consider a commonly-assumed setting of the Linear Representation Hypothesis (LRH) (Elhage et al., 2022; Park et al., 2023; Rajendran et al., 2024), and consider the problem of finding probes capable of classifying the representations. Instead of a joint optimization, we assume the representations are already given and follow a strict linear-factorization structure under LRH.

Specifically, we make the following assumptions: (1) The representation for any input with concept tuple $\boldsymbol{c} = (c_1, \ldots, c_k) \in [n]^k$ is

$$\boldsymbol{z_c} = \sum_{i=1}^k \alpha_{i,c_i} \boldsymbol{d}_i \quad \boldsymbol{d}_i \in \mathbb{R}^d. \tag{68}$$

(2) The concept direction vectors $\{\boldsymbol{d}_i\}_{i=1}^k \subset \mathbb{R}^d$ are fixed and orthogonal, i.e., $\boldsymbol{d}_i^\mathsf{T} \boldsymbol{d}_\ell = 0$ for $i \neq \ell$ (hence linearly independent and $d \geq k$)[3].

(3) For each concept $i$, its $n$ values correspond to a known ordered set of scalar coefficients $\{\alpha_{i,j}\}_{j=1}^n$, with distinct values $j$ for different $i$. Under these assumptions, orthogonality of the directions implies the representations form a regular grid in $\boldsymbol{z}$ inside $V$ (each coordinate axis is one concept).

We consider the problem of finding the linear probes $\{\boldsymbol{w}_{i,j}\}$ that classify the representations:

$$\min_{\{\boldsymbol{w}_{i,j}, b_{i,j}\}} \quad \sum_{\boldsymbol{c}} \sum_{i=1}^k \left( -\log \frac{\exp\left(\boldsymbol{w}_{i,c_i}^\mathsf{T} \boldsymbol{z_c} + b_{i,c_i}\right)}{\sum_{j=1}^n \exp\left(\boldsymbol{w}_{i,j}^\mathsf{T} \boldsymbol{z_c} + b_{i,j}\right)} \right). \tag{69}$$

We consider the nearest-neighbor classifier. Concretely, we define the approximated concept-value prototypes by averaging representations over all tuples with concept $i$ fixed to value $j$:

$$\widetilde{\boldsymbol{d}}_{i,j} := \frac{1}{n^{k-1}} \sum_{\boldsymbol{c} \in [n]^k : c_i = j} \boldsymbol{z_c} = \alpha_{i,j} \boldsymbol{d}_i + \sum_{\ell \neq i} \bar{\alpha}_\ell \boldsymbol{d}_\ell := \alpha_{i,j} \boldsymbol{d}_i + \boldsymbol{m}_i, \tag{70}$$

where $\bar{\alpha}_\ell := \frac{1}{n} \sum_{r=1}^n \alpha_{\ell,r}$ and $\boldsymbol{m}_i := \sum_{\ell \neq i} \bar{\alpha}_\ell \boldsymbol{d}_\ell$. Using these prototypes, we set the corresponding affine probes as[4]

$$\widetilde{\boldsymbol{w}}_{i,j} = 2\widetilde{\boldsymbol{d}}_{i,j}, \qquad \widetilde{b}_{i,j} = -\|\widetilde{\boldsymbol{d}}_{i,j}\|_2^2. \tag{71}$$

We can verify that such a classifier works. The score for the $j$-th probe of concept $i$ on an input $\boldsymbol{z_c}$ (where the true value for concept $i$ is $c_i$) is:

$$\begin{aligned}
h_{i,j}(\boldsymbol{z_c}) &:= \widetilde{\boldsymbol{w}}_{i,j}^\mathsf{T} \boldsymbol{z_c} + \widetilde{b}_{i,j} \\
&= 2\widetilde{\boldsymbol{d}}_{i,j}^\mathsf{T} \boldsymbol{z_c} - \|\widetilde{\boldsymbol{d}}_{i,j}\|_2^2 \\
&= 2(\alpha_{i,j} \boldsymbol{d}_i + \boldsymbol{m}_i)^\mathsf{T} \boldsymbol{z_c} - \|\alpha_{i,j} \boldsymbol{d}_i + \boldsymbol{m}_i\|_2^2 \\
&= \left(2\alpha_{i,j} \boldsymbol{d}_i^\mathsf{T}(\boldsymbol{z_c} - \boldsymbol{m}_i) - \alpha_{i,j}^2 \|\boldsymbol{d}_i\|_2^2\right) + \left(2\boldsymbol{m}_i^\mathsf{T} \boldsymbol{z_c} - \|\boldsymbol{m}_i\|_2^2\right).
\end{aligned}$$

---

[3]We consider this for simplicity; in general one may consider the directions to be linearly independent instead

[4]We find these weights by considering the power diagram (Aurenhammer, 1987) over the points $\widetilde{\boldsymbol{d}}_{i,j}$, the weights are biases are then remaps from the power diagram to the joint argmax classifier.

For every fixed concept $i$ and a datapoint $\boldsymbol{z_c}$, maximizing over $j$ is therefore

$$\arg\max_{j\in[n]} h_{i,j}(\boldsymbol{z_c}) = \arg\max_{j\in[n]} \left(2\alpha_{i,j}\boldsymbol{d}_i^{\mathsf{T}}(\boldsymbol{z_c}-\boldsymbol{m}_i) - \alpha_{i,j}^2\|\boldsymbol{d}_i\|_2^2\right)$$

$$= \arg\max_{j\in[n]} \left(2\alpha_{i,j}\boldsymbol{d}_i^{\mathsf{T}}\left(\sum_{\ell=1}^{k}\alpha_{\ell,c_\ell}\boldsymbol{d}_\ell - \boldsymbol{m}_i\right) - \alpha_{i,j}^2\|\boldsymbol{d}_i\|_2^2\right)$$

$$= \arg\max_{j\in[n]} \left(2\alpha_{i,j}\boldsymbol{d}_i^{\mathsf{T}}\left(\alpha_{i,c_i}\boldsymbol{d}_i + \sum_{\ell\neq i}\alpha_{\ell,c_\ell}\boldsymbol{d}_\ell - \boldsymbol{m}_i\right) - \alpha_{i,j}^2\|\boldsymbol{d}_i\|_2^2\right)$$

$$= \arg\max_{j\in[n]} \left(2\alpha_{i,j}\alpha_{i,c_i}\|\boldsymbol{d}_i\|_2^2 - \alpha_{i,j}^2\|\boldsymbol{d}_i\|_2^2\right)$$

$$= \arg\max_{j\in[n]} \left(-(\alpha_{i,j}-\alpha_{i,c_i})^2\|\boldsymbol{d}_i\|_2^2\right)$$

$$= \arg\min_{j\in[n]} (\alpha_{i,j}-\alpha_{i,c_i})^2$$

$$= c_i.$$

where we dropped the additive term $2\boldsymbol{m}_i^{\mathsf{T}}\boldsymbol{z_c} - \|\boldsymbol{m}_i\|_2^2$, since it does not depend on $j$, and used orthogonality: $\boldsymbol{d}_i^{\mathsf{T}}\boldsymbol{d}_\ell = 0$ for $\ell \neq i$ and $\boldsymbol{d}_i^{\mathsf{T}}\boldsymbol{m}_i = 0$. The last step follows since $\|\boldsymbol{d}_i\|_2^2 > 0$ is constant and $\{\alpha_{i,j}\}_{j=1}^{n}$ are distinct.

As shown above, such a classifier is easy to construct and correctly classifies all the points in the concept space. If orthogonality did not hold, the nearest neighbor classifier is not guaranteed to be correct. We illustrate the geometry of the probes together with the decision regions in Fig. 14 in two cases, where $k = 2, n = 30$ and the deviation from orthogonal features varies slightly. We note that the weight vectors, indicated as arrows in the plot, are near-parallel.

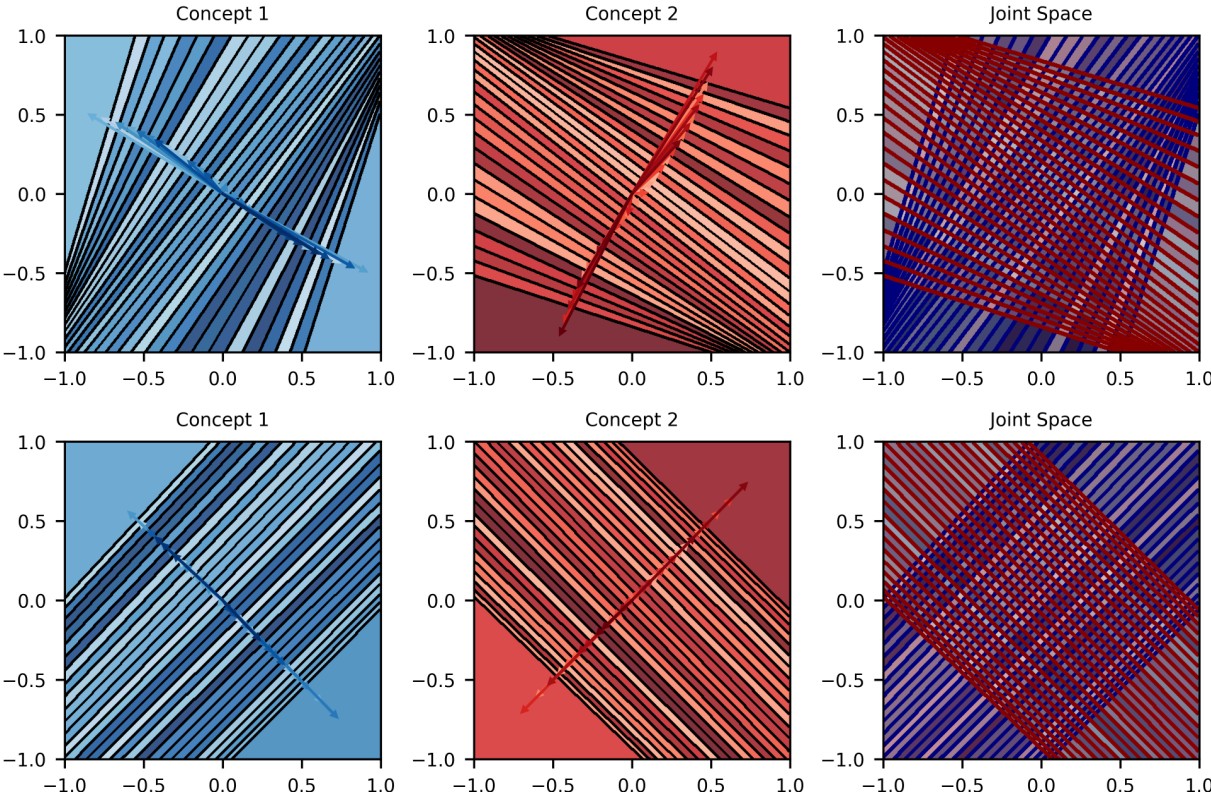

*Figure 14.* **Probe geometry in the LRH toy setup** ($k = 2, n = 30$). Each row corresponds to a different choice of concept directions (top: mildly non-orthogonal; bottom: closer to orthogonal). Left and middle columns show concept-wise decision regions for concept 1 (blue) and concept 2 (red), with arrows indicating probe directions. The right column overlays both concept families in the joint space; intersections of blue and red stripes form the cells associated with tuples $(c_1, c_2)$.

## E.2. Case 2: Ideal "on-off" concept classifier

Here we consider a setting that intuitively could be described as exhibiting behavior that is desirable for compositional generalization. In particular, we consider a CLIP-style linearly compositional model with the constraint that the probe scores $h_{i,j}$ should be constant among the matches, and constant among the mis-matches.

In CLIP-like models, this is usually viewed through cosine similarity. In the normalized view ($\|z\|_2 = 1$ and $\|w_{i,j}\|_2 = 1$), decision regions on the sphere are spherical caps rather than Euclidean half-spaces. For a cosine-similarity classifier, the decision region for class $(i, j)$ is

$$\mathcal{R}_{i,j} := \left\{ z \in \mathbb{S}^{d-1} : w_{i,j}^\mathsf{T} z > w_{i,k}^\mathsf{T} z \; \forall k \neq j \right\}. \tag{72}$$

We refer to this setting as "on-off" concept classifier, as there are only two possible scores for each probe: $\alpha$ if the concept matches, and $\beta$ if it does not. That is, exist constants $\alpha > \beta$ in $[-1, 1]$ such that for all $i, j$ and all tuples $c$,

$$w_{i,j}^\mathsf{T} z_c = \begin{cases} \alpha & \text{if } j = c_i \\ \beta & \text{if } j \neq c_i. \end{cases} \tag{73}$$

We illustrate this setting in Fig. 15.

The key results in this section are two-fold: (1) a model exhibiting this behavior requires at least $1 + k(n - 1)$ dimensions, meaning that under a large number of concepts and values per concept, even current systems will not be able to reliably distinguish between correct and incorrect matches between a sample and its corresponding concept values (Proposition 6); (2) assuming that the model is able to distinguish between correct and incorrect matches, one can linearly approximate the representations in a way that preserves the "on-off" pattern (Proposition 7).

## "On-off concept classifier"

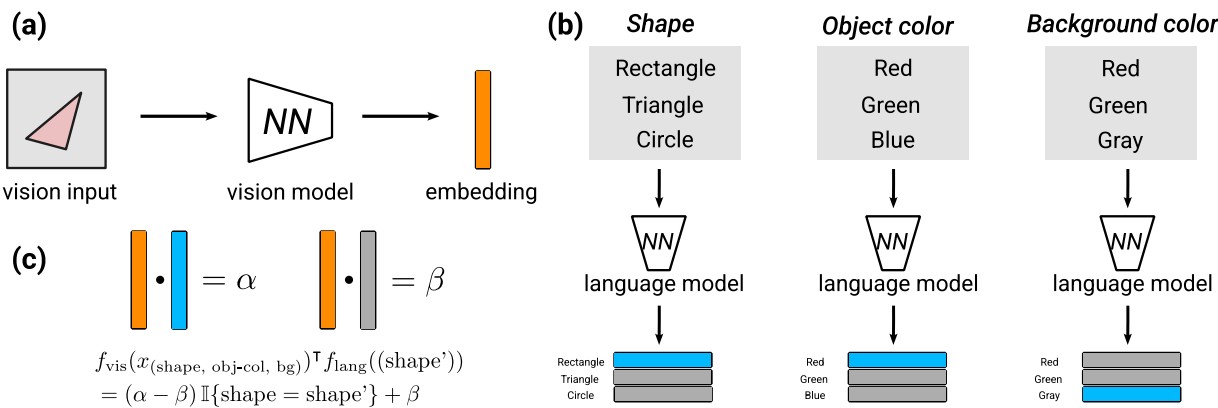

*Figure 15.* **Illustration of the "on-off concept classifier" mechanism. (a)** A vision input is processed by a neural network to produce an embedding. **(b)** Each concept (e.g., shape, object color, background color) is probed independently using a set of language model probes, one per possible value. **(k)** The probe for a given concept yields a high score $\alpha$ if the concept matches and a lower score $\beta$ otherwise, as formalized in the logit equation at bottom.

**Proposition 6** (Minimal dimensionality from fixed dot-products)**.** Assume $k \geq 2$ and $n \geq 2$. For each concept $i \in [k]$ and value $j \in [n]$, let the probe weight be $w_{i,j} \in \mathbb{R}^d$, such that for any tuple $c = (c_1, \ldots, c_k) \in [n]^k$ there exists a corresponding $z_c \in \mathbb{R}^d$, and there exist $\alpha, \beta \in \mathbb{R}$ with $\alpha \neq \beta$ such that the "on-off" pattern holds (and $\alpha \neq -\beta(n - 1)$):

$$w_{i,j}^\mathsf{T} z_c = \begin{cases} \alpha, & j = c_i, \\ \beta, & j \neq c_i, \end{cases} \qquad \text{for all } i, j, c, \tag{74}$$

Then the ambient dimension $d$ must satisfy

$$d \geq 1 + k(n - 1). \tag{75}$$

Moreover, this bound is tight: for any $(\alpha, \beta)$ with $\alpha \neq \beta$, there exist explicit probe/representation families that realize (74) in dimension $d = 1 + k(n - 1)$.

*Proof.* We write the constraints in matrix form. First, we stack probe vectors as rows:

$$\boldsymbol{P} = \begin{bmatrix} \boldsymbol{w}_{1,1}^{\mathsf{T}} \\ \vdots \\ \boldsymbol{w}_{k,n}^{\mathsf{T}} \end{bmatrix} \in \mathbb{R}^{kn \times d}, \qquad (\text{row } (i-1)n + j = \boldsymbol{w}_{i,j}^{\mathsf{T}}). \tag{76}$$

Then stack representations as columns:

$$\boldsymbol{X} = \begin{bmatrix} \boldsymbol{z}_{\boldsymbol{c}_1} & \cdots & \boldsymbol{z}_{\boldsymbol{c}_{n^k}} \end{bmatrix} \in \mathbb{R}^{d \times n^k}. \tag{77}$$

The "on-off" constraints (74) become

$$\boldsymbol{Y} = \boldsymbol{P}\boldsymbol{X} \in \mathbb{R}^{kn \times n^k}, \tag{78}$$

where entries of $\boldsymbol{Y}$ are

$$y_{(i,j),\boldsymbol{c}} = \begin{cases} \alpha, & j = c_i, \\ \beta, & j \neq c_i. \end{cases} \tag{79}$$

and will sometimes denote rows of $\boldsymbol{Y}$ as $\boldsymbol{y}_{(i,j)}$ instead. To understand the rank of $\boldsymbol{Y}$, first note that it is at most $kn$. Additionally, note that the rows can be divided into $k$ blocks. As we will show, each such block contains up to one dependence that is achieved by summing up the rows within a block.

For each block $i \in [k]$ and any column $\boldsymbol{c} \in [n]^k$, we can sum the $n$ entries in that block as

$$\sum_{j=1}^{n} y_{(i,j),\boldsymbol{c}} = \sum_{j=1}^{n} \big( \alpha \mathbb{I}_{c_i = j} + \beta \mathbb{I}_{c_i \neq j} \big) = \alpha + \beta(n-1). \tag{80}$$

Importantly, this sum is the same regardless of $\boldsymbol{c}$ and chosen $i$. Therefore, each block sums up to the same vector, and is therefore redundant. This implies that the rank can be at most $nk - (k-1) = 1 + k(n-1)$. Next, we show that within each block, each row is independent.

For that, we will show that for any concept "block" $i \in [k]$ every row $(i,j)$ corresponding to the value $j$ of the $i$-th concept is linearly independent of all the rows in blocks $i' \neq i$. We denote the span of the other blocks' rows as

$$\mathcal{S} := \operatorname{span}(\{\boldsymbol{y}_{(i',j')}\}_{i' \neq i, j' \in [n]}). \tag{81}$$

For that, we take any two columns $\boldsymbol{c}, \boldsymbol{c}' \in [n]^k$ that are identical up to the $i$-th concept and satisfy $c_m = c'_m$ for $m \neq i$ and $c_i \neq c'_i$, and note that all the rows outside the $i$-th block are identical: $y_{(i',j'),\boldsymbol{c}} = y_{(i',j'),\boldsymbol{c}'}$ for all $i' \neq i$ and all $j' \in [n]$, and only differ in the $i$-th block rows. Because of that, any linear combination from $\mathcal{S}$ with some coefficients will have matching columns:

$$\Big( \sum \lambda_n \boldsymbol{s}_n \Big)_{\boldsymbol{c}} = \Big( \sum \lambda_n \boldsymbol{s}_n \Big)_{\boldsymbol{c}'}, \tag{82}$$

but both $\boldsymbol{c}$ and $\boldsymbol{c}'$ differ in exactly the $i$-th concept, and therefore their column values in the $i$-th block must be different:

$$y_{(i,c_i),\boldsymbol{c}} \neq y_{(i,c_i),\boldsymbol{c}'}, \tag{83}$$

and therefore no vector in the span can be equal to $\boldsymbol{y}_{i,c_i}$. All that is needed to show now is that taking any linear combination of the rows in the $i$-th block cannot be expressed by any $\boldsymbol{c} \in \mathcal{S}$.

We do this by applying the same reasoning but instead of taking 2 rows we take $n$ rows. Concretely, for the $i$-th concept we take all $n$ rows: $(i,1), \ldots, (i,n)$ and consider $\boldsymbol{c}_1, \ldots, \boldsymbol{c}_n$ which are all identical up to the $i$-th concept. Again, for any $i' \neq i$ and any row index $j \in [n]$, the values are constant across these columns: $y_{(i',j),\boldsymbol{c}_m} = y_{(i',j),\boldsymbol{c}_{m'}}$ for all $m, m' \in [n]$, and only the $i$-th block rows change across $\boldsymbol{c}_1, \ldots, \boldsymbol{c}_n$. We can thus consider the span of the $i$-th block among the $n$ rows by taking $\boldsymbol{\lambda} \in \mathbb{R}^n$ and considering only the $n$ columns corresponding to $\boldsymbol{c}_1, \ldots, \boldsymbol{c}_n$, writing the linear combination as

$$\boldsymbol{r} := \sum_{j=1}^{n} \lambda_j \boldsymbol{y}_{(i,j)} \in \mathbb{R}^{n^k}. \tag{84}$$

We now evaluate $\boldsymbol{r}$ on the $n$ columns $\boldsymbol{c}_1, \ldots, \boldsymbol{c}_n$. By construction, in column $\boldsymbol{c}_m$ the only row in block $i$ that takes value $\alpha$ is $(i,m)$, while all other $(i,j)$ take value $\beta$. Therefore the $m$-th selected entry is

$$r_m = \lambda_m \alpha + \sum_{j \neq m} \lambda_j \beta = \lambda_m \alpha + (S - \lambda_m)\beta, \tag{85}$$

where $S := \sum_{j=1}^{n} \lambda_j$ is the sum of the coefficients. Finally, for $\boldsymbol{r}$ to be within $\mathcal{S}$ it has to hold that for any columns $m, m'$ the entries are equal:

$$\lambda_m \alpha + (S - \lambda_m)\beta = \lambda_{m'}\alpha + (S - \lambda_{m'})\beta$$
$$\Rightarrow \quad (\lambda_m - \lambda_{m'})(\alpha - \beta) = 0. \tag{86}$$

But by assumption $\alpha \neq \beta$, and therefore $\lambda_m = \lambda_{m'}$ for all $m, m' \in [n]$. Clearly, if $\lambda_m \neq 0$, then this results in the constant direction produced by any concept block, which is the only direction that lies in $\mathcal{S}$. Therefore each concept block produces $n-1$ linearly independent vectors.

Combined with the redundancy within a concept block, this implies that $\mathrm{rank}(\boldsymbol{Y}) = 1 + k(n-1)$.

Finally, since $\boldsymbol{Y} = \boldsymbol{P}\boldsymbol{X}$,

$$1 + k(n-1) = \mathrm{rank}(\boldsymbol{Y}) \leq \mathrm{rank}(\boldsymbol{P}) \leq d. \tag{87}$$

This proves (75) and shows that such a model requires at least $1 + k(n-1)$ dimensions.

Tightness follows from an explicit construction by placing probes and representations in, for example, orthogonal directions. $\square$

Below, we provide a numerical example to illustrate the form of the logit matrix $Y$ for the case of two concepts, three values each.

**Example 1** (Two concepts, three values each: $c = 2$, $n = 3$)**.** Set $(\alpha, \beta) = (1, 0.2)$. The row indices are $(i, j) \in \{1, 2\} \times \{1, 2, 3\}$, the column indices are the $3^2 = 9$ tuples $(v_1, v_2) \in \{1, 2, 3\}^2$:

$$
Y = 
\begin{array}{c}
\\
(1,1) \\
(1,2) \\
(1,3) \\
(2,1) \\
(2,2) \\
(2,3)
\end{array}
\begin{array}{c}
\begin{array}{ccccccccc}
11 & 12 & 13 & 21 & 22 & 23 & 31 & 32 & 33
\end{array} \\
\left(\begin{array}{ccccccccc}
1 & 1 & 1 & 0.2 & 0.2 & 0.2 & 0.2 & 0.2 & 0.2 \\
0.2 & 0.2 & 0.2 & 1 & 1 & 1 & 0.2 & 0.2 & 0.2 \\
0.2 & 0.2 & 0.2 & 0.2 & 0.2 & 0.2 & 1 & 1 & 1 \\
1 & 0.2 & 0.2 & 1 & 0.2 & 0.2 & 1 & 0.2 & 0.2 \\
0.2 & 1 & 0.2 & 0.2 & 1 & 0.2 & 0.2 & 1 & 0.2 \\
0.2 & 0.2 & 1 & 0.2 & 0.2 & 1 & 0.2 & 0.2 & 1
\end{array}\right)
\end{array}
\tag{88}
$$

Note that each block corresponding to either the first or the second concept sum up to the same vector. Within each block there are 2 linearly independent vectors. Hence, $\mathrm{rank}(Y) = 5 = 1 + c(n-1)$.

Under such a design, linear factorization holds immediately – at least up to projection onto the span of the probe vectors; or exactly if the dimensionality of the embeddings is exactly $1 + k(n-1)$. To make the setup explicit, we can also impose a fixed relation between $\alpha$ and $\beta$. We avoid the condition of the global mean being zero, as this implies that $\alpha + (n-1)\beta = 0$, i.e., $\alpha = -(n-1)\beta$ (similar to Proposition 6). Related normalization choices are discussed in (Lee et al., 2024).

**Proposition 7** (Additive factorization from the on-off pattern)**.** Assume $k \geq 2$ and $n \geq 2$. For each $i \in [k]$ and $j \in [n]$, let $\boldsymbol{w}_{i,j} \in \mathbb{R}^d$ be a probe vector. For each tuple $\boldsymbol{c} = (c_1, \ldots, c_k) \in [n]^k$, let $\boldsymbol{z_c} \in \mathbb{R}^d$ be its corresponding representation. Assume there exist $\alpha > \beta$, with $\alpha, \beta \in \mathbb{R}$, such that

$$
\boldsymbol{w}_{i,j}^{\mathsf{T}} \boldsymbol{z_c} = \begin{cases} \alpha & \text{if } j = c_i, \\ \beta & \text{if } j \neq c_i, \end{cases} \qquad \forall i, j, \boldsymbol{c}.
\tag{89}
$$

Then there exist vectors $\bar{\boldsymbol{z}} \in \mathbb{R}^d$, $\boldsymbol{u}_{i,j} \in \mathbb{R}^d$, and reconstructed representations $\widetilde{\boldsymbol{z}}_{\boldsymbol{c}} \in \mathbb{R}^d$ of the additive form

$$
\widetilde{\boldsymbol{z}}_{\boldsymbol{c}} = \bar{\boldsymbol{z}} + \sum_{i=1}^{k} \boldsymbol{u}_{i,c_i}, \qquad \forall \boldsymbol{c} \in [n]^k,
\tag{90}
$$

such that for every $i, j, \boldsymbol{c}$,

$$
\boldsymbol{w}_{i,j}^{\mathsf{T}} \widetilde{\boldsymbol{z}}_{\boldsymbol{c}} = \boldsymbol{w}_{i,j}^{\mathsf{T}} \boldsymbol{z_c} = \begin{cases} \alpha & \text{if } j = c_i, \\ \beta & \text{if } j \neq c_i. \end{cases}
\tag{91}
$$

So $\widetilde{\boldsymbol{z}}_{\boldsymbol{c}}$ and $\boldsymbol{z_c}$ are indistinguishable to the probes.

*Proof.* We again show that computing the factors as averages satisfies the condition. For that, we define the global mean $\bar{\boldsymbol{z}}$, the concept factors $\boldsymbol{a}_{i,j}$, and the centered factors $\boldsymbol{u}_{i,j}$:

$$
\bar{\boldsymbol{z}} := \frac{1}{n^k} \sum_{\boldsymbol{c}' \in [n]^k} \boldsymbol{z}_{\boldsymbol{c}'}, \qquad \boldsymbol{a}_{i,j} := \frac{1}{n^{k-1}} \sum_{\boldsymbol{c}': c_i' = j} \boldsymbol{z}_{\boldsymbol{c}'}, \qquad \boldsymbol{u}_{i,j} := \boldsymbol{a}_{i,j} - \bar{\boldsymbol{z}}, \qquad \widetilde{\boldsymbol{z}}_{\boldsymbol{c}} := \bar{\boldsymbol{z}} + \sum_{i=1}^{k} \boldsymbol{u}_{i,c_i}.
$$

We will confirm this choice works through a simple calculation. First, for any $(i, j)$ the dot product with the average representation is

$$\boldsymbol{w}_{i,j}^{\mathsf{T}} \bar{\boldsymbol{z}} = \frac{1}{n^k}\left(n^{k-1}\alpha + (n^k - n^{k-1})\beta\right) = \frac{\alpha + (n-1)\beta}{n} =: \delta, \tag{92}$$

Second, for any $i', r$, we expand the dot product with the average non-centered factor $\boldsymbol{a}_{i,j}$:

$$\boldsymbol{w}_{i',r}^{\mathsf{T}} \boldsymbol{a}_{i,j} = \frac{1}{n^{k-1}} \sum_{\boldsymbol{c}':c_i'=j} \boldsymbol{w}_{i',r}^{\mathsf{T}} \boldsymbol{z}_{\boldsymbol{c}'}. \tag{93}$$

and decompose it into two cases:

**Case 1: $i' = i$ (probe and conditioning on the same concept).** Here the condition $c_i' = j$ fixes whether the target concept is probed:

- If $r = j$, every term in the sum equals $\alpha$, so $\boldsymbol{w}_{i',r}^{\mathsf{T}} \boldsymbol{a}_{i,j} = \alpha$.
- If $r \neq j$, every term in the sum equals $\beta$, so $\boldsymbol{w}_{i',r}^{\mathsf{T}} \boldsymbol{a}_{i,j} = \beta$.

**Case 2: $i' \neq i$ (probe and conditioning on different concepts).** Now $c_i' = j$ does not constrain $c_{i'}'$, so:

- exactly $n^{k-2}$ tuples satisfy $c_{i'}' = r$ and contribute $\alpha$,
- exactly $(n-1)n^{k-2}$ tuples satisfy $c_{i'}' \neq r$ and contribute $\beta$.

Therefore

$$\boldsymbol{w}_{i',r}^{\mathsf{T}} \boldsymbol{a}_{i,j} = \frac{n^{k-2}\alpha + (n-1)n^{k-2}\beta}{n^{k-1}} = \delta.$$

Putting it all together:

$$\boldsymbol{w}_{i',r}^{\mathsf{T}} \boldsymbol{a}_{i,j} = \begin{cases} \alpha, & i' = i, \ r = j, \\ \beta, & i' = i, \ r \neq j, \\ \delta, & i' \neq i. \end{cases}$$

By linearity, subtracting the global mean term gives

$$\boldsymbol{w}_{i',r}^{\mathsf{T}} \boldsymbol{u}_{i,j} = \boldsymbol{w}_{i',r}^{\mathsf{T}} \boldsymbol{a}_{i,j} - \boldsymbol{w}_{i',r}^{\mathsf{T}} \bar{\boldsymbol{z}} = \begin{cases} \alpha - \delta, & i' = i, \ r = j, \\ \beta - \delta, & i' = i, \ r \neq j, \\ 0, & i' \neq i. \end{cases} \tag{94}$$

We now evaluate the reconstructed representation:

$$\begin{aligned} \boldsymbol{w}_{i,j}^{\mathsf{T}} \widetilde{\boldsymbol{z}}_{\boldsymbol{c}} &= \boldsymbol{w}_{i,j}^{\mathsf{T}} \bar{\boldsymbol{z}} + \sum_{t=1}^{k} \boldsymbol{w}_{i,j}^{\mathsf{T}} \boldsymbol{u}_{t,c_t} \\ &= \delta + \boldsymbol{w}_{i,j}^{\mathsf{T}} \boldsymbol{u}_{i,c_i} + \sum_{t \neq i} \underbrace{\boldsymbol{w}_{i,j}^{\mathsf{T}} \boldsymbol{u}_{t,c_t}}_{=0 \text{ by } (94)} \\ &= \delta + \left(\boldsymbol{w}_{i,j}^{\mathsf{T}} \boldsymbol{a}_{i,c_i} - \delta\right) = \boldsymbol{w}_{i,j}^{\mathsf{T}} \boldsymbol{a}_{i,c_i} \\ &= \begin{cases} \alpha, & j = c_i, \\ \beta, & j \neq c_i. \end{cases} \end{aligned}$$

Hence $\widetilde{\boldsymbol{z}}_{\boldsymbol{c}}$ reproduces exactly the same on-off probe responses as $\boldsymbol{z}_{\boldsymbol{c}}$ for every concept and value, which proves (91). $\qquad\square$

## F. Discussion on stability

Here we discuss the role of Stability and what can go wrong without it. Without Stability (Desideratum 3), the only requirement on the readout is that it correctly classifies the observed grid. This places no constraint on how the decision regions are shaped: they can vary arbitrarily across training supports, as long as every grid point "lands" in the correct region. In practice, this means that decision boundaries may pass arbitrarily close to data points, that some concept-value regions may be infinitesimally thin, and that retraining on a different valid support can produce a completely different partition of the space. In short, no robustness holds. As we show below, this pathology does not disappear with scale: even as $n \to \infty$, the decision regions can remain arbitrarily complex, which is never a desirable property. Linear separability alone therefore does not force linear factorization, nor does it force projected $R^2$ to be close to 1 (see also Appendices G.2 and G.4).

Stability prevents these pathologies by requiring that the predicted probability vector over concept values agrees at every input, regardless of which valid support was used for training. In the linear-softmax setting, this implies that probe weights are the same across supports (or, up to a shift in non-binary case), pinning down the decision boundaries and forcing the representational structure established in our main results.

**Counterexamples to linear factorization even as $n \to \infty$.** Suppose we only assume linear separability on the full grid: for concept tuples $\boldsymbol{q} = (q_1, \ldots, q_k) \in [n]^k$ with embeddings $\boldsymbol{z_q} \in \mathbb{R}^d$, there exist probes $\{(\boldsymbol{w}_{i,j}, b_{i,j})\}_{j=1}^n$ such that

$$\arg \max_{j \in [n]} \left( \boldsymbol{w}_{i,j}^\mathsf{T} \boldsymbol{z_q} + b_{i,j} \right) = q_i, \qquad \forall i \in [k], \ \forall \boldsymbol{q} \in [n]^k. \tag{95}$$

This guarantees perfect classification but does not enforce Stability across supports.

Does this imply additive structure? In general, no. Even in the simple case of $d = 2$, $k = 2$, and even as $n \to \infty$, one can construct point clouds $\{\boldsymbol{z}_{q_1, q_2}\} \subset \mathbb{R}^2$ with perfect linear separability that do *not* admit a decomposition

$$\boldsymbol{z}_{q_1, q_2} = \boldsymbol{u}_{1, q_1} + \boldsymbol{u}_{2, q_2}. \tag{96}$$

Linear separability is therefore compatible with non-factorized geometry. Fig. 16 illustrates this failure mode, which is consistent with the low-$R^2$ counterexamples in Appendix G.4.

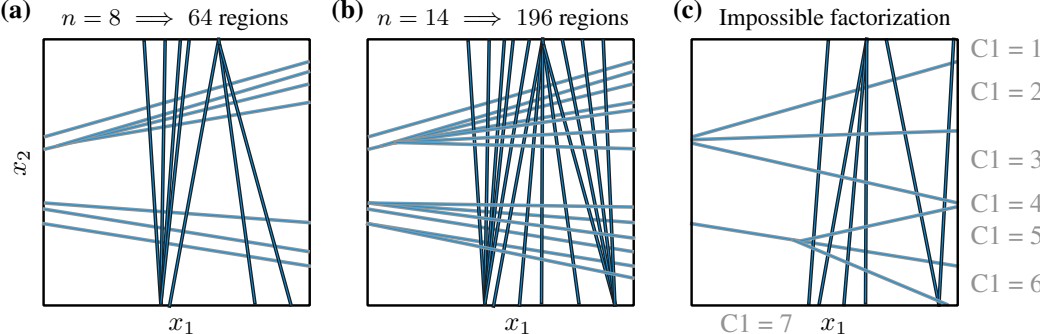

**(a)** $n = 8 \implies 64$ regions   **(b)** $n = 14 \implies 196$ regions   **(c)** Impossible factorization

*Figure 16.* **Linear separability without linear factorization.** Two families of affine decision boundaries in $\mathbb{R}^2$ (black for concept 1, gray for concept 2) divide the plane into regions, one per pair of concept values. **(a,b)**: with $n = 8$ and $n = 14$ levels per concept, the arrangement yields $n^2$ regions (64 and 196). By inserting additional nearly-parallel boundaries, existing regions can be split into arbitrarily thin pieces while maintaining perfect linear separability. **(c)**: No linear factorization can be achieved: whichever factors we pick, the separability of some datapoints is violated.

Concretely, in Fig. 16, panels **(a)** and **(b)** show two non-parallel families of decision boundaries whose intersections produce $n^2$ convex cells, one per pair $(q_1, q_2)$. Since perfect classification only requires each grid point to fall in the correct cell, the geometry of the cells themselves is unconstrained. By adding $\varepsilon$-perturbed boundaries, one can subdivide regions so that some concept values occupy arbitrarily small areas as $n$ grows, while perfect separability is preserved. Without Stability, there is nothing to prevent such degenerate configurations, and linear readout constraints alone do not pin down factorized structure. Handling such cases, e.g., by imposing minimum-area constraints on decision regions, is possible in principle but becomes technical; our framework avoids these pathologies by design through the Stability desideratum.

More broadly, any model aiming for robust compositional generalization will need to prevent such degenerate configurations, whether through exact Stability or approximate forms of it.

# G. Additional information

In this section we expand on linear factorization, make a note on the non-triviality of linear factorization, and expand on the reasoning of using whitening in measuring linear factorization. In Appendix G.1 we summarize the overall procedure of measuring linear factorization. In Appendix G.3 we provide an intuition of linear factorization through a simple example. In Appendix G.4 we show that linear factorization is not a trivial property of linearly compositional models, and illustrate a few cases where the representation cannot be decomposed into a sum of per-concept components even under perfect classification.

## G.1. Testing linear factorization

Large pre-trained models may encode information beyond the specific concepts in our dataset. To isolate the conceptual structure, we train per-concept linear probes. For each concept $i \in [k]$ and value $j$, we learn a linear probe $\boldsymbol{w}_{i,j}$, form the probe matrix $\boldsymbol{W} \in \mathbb{R}^{m \times d}$, where $m$ is the number of values across all concepts, and project embeddings onto the joint probe span. We do this by first computing the projection matrix $P_{\boldsymbol{W}}$ and then projecting the embeddings onto the joint probe span.

We report *Projected $R^2$* after projecting embeddings onto the probe span. To prevent trivial high scores from dominant directions, we whiten the embeddings by applying PCA and normalizing to unit covariance. We compute metrics on $P_{\boldsymbol{W}} \boldsymbol{z}$ after PCA-whitening, applying the same transform to data and reconstructions. We elaborate on this below.

## G.2. Whitening in measuring linear factorization

We need to be cautious when assessing the degree of linearity in the representations, otherwise, we may mistake high $R^2$ scores for linear factorization when in fact the representation is not linearly factored. For example, if certain concept values dominate the variance in the representation, the $R^2$ may be inflated. To address this, in the main experiments in Section 5.1 we whiten the representations by applying PCA and normalizing to unit covariance. This ensures that a few dominant directions do not dominate the variance in the representation. If the representations are already linearly factored, this will not affect the $R^2$ score.

We illustrate this through three examples in a hypothetical two-dimensional representation space with two concepts in Fig. 17. In the first case (**(a)**) the representation is already linearly factored: each embedding is written as a sum of two concept components without noise. This yields an $R^2$ score of 1; whitening does not change the score.

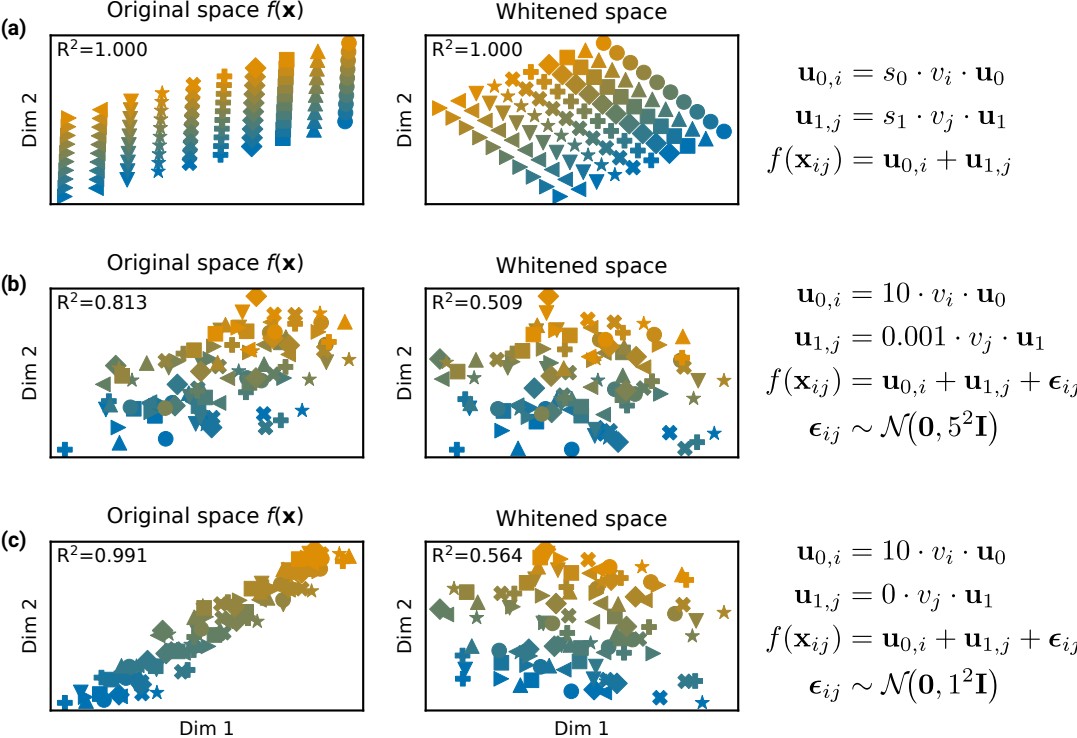

*Figure 17.* **Whitening in measuring linear factorization**. The representation is not linearly factored, but the $R^2$ score is high due to the dominance of the dominant direction.

In the second case (**(b)**) the representation is partly linear, but the noise $\boldsymbol{\epsilon}_{ij}$, independent of the concept values, dominates the overall

variance. Since the scale of the noise is generally lower than the scale of the first concept component, the $R^2$ score is high at $0.813$. Whitening, however, removes the dominant direction, and the $R^2$ score drops to $0.509$.

Lastly, in the third case (**(k)**) the representation does not express any information about the second concept, yet the $R^2$ score is still high at $0.991$. Again, whitening reveals the underlying issue and changes the score to $0.564$ due to the noise in the embeddings.

### G.3. Intuition of linear factorization

We measure the extent of linearity present in the embeddings through the $R^2$ score. Intuitively, the score quantifies how well the representation can be decomposed into a sum of per-concept components. Recall from Definition 1 that we assume a presence of $k$ concepts, each of which can take any of the $n$ values. A value of $R^2 = 1$ indicates that the representation can be perfectly decomposed into a sum of per-concept components.

We illustrate a few examples to give intuition. We consider a two-dimensional representation space with two concepts ($k = 2$). In the first case, we consider a case of 24 values per concept ($n = 24$). In the second case, we consider a case of 6 values per concept ($n = 6$). In both cases the reported $R^2$ are w.r.t. the whitened space.

The first case (Fig. 18, **(a)**) exhibits perfect linearity in the embeddings with $R^2 = 1$. In this case, the $n^2 = 24^2 = 576$ can be perfectly generated using only $2 \cdot 24 = 48$ vectors in $\mathbb{R}^2$. The second and third columns of the plot show the approximations of the underlying factors $\boldsymbol{u}_{0,i}, \boldsymbol{u}_{1,j}, i, j \in [n]$. As expected, using these approximate factors allows us to perfectly reconstruct the representation, shown in the fourth column.

The second case (Fig. 18, **(b)**) exhibits lower degree of linearity with $R^2 = 0.53$. As such, we cannot perfectly reconstruct the representation using only the approximate factors, as shown in the last column of the plot.

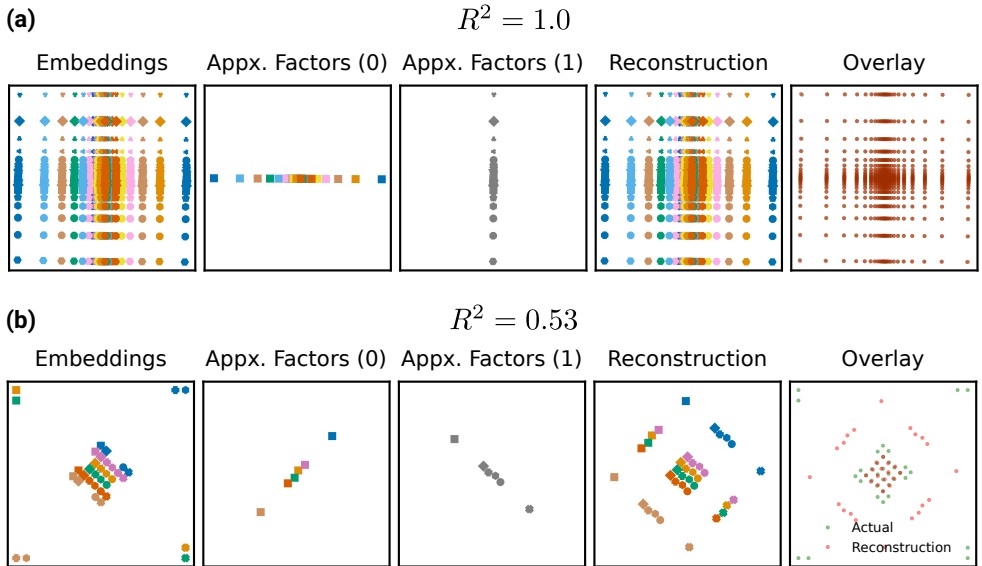

*Figure 18.* **Intuition of linear factorization**. In **(a)** the representations can be perfectly reconstructed by a set of per-concept components, while in **(b)** they are insufficient to reconstruct the representation. Refer to the text for more details.

### G.4. Non-triviality of linear factorization of linearly compositional models

Recall that linearly compositional models (though not necessarily generalizable ones), as defined in Definition 3, admit a set of probes that can perfectly classify all inputs in the grid $\mathcal{C}$. Proposition 1 shows that linearly compositional models must exhibit linear factorization. This naturally raises the converse question: does the mere existence of a set of perfect linear classifiers imply linear factorization? We answer in the negative.

The intuition is as follows. As per Desideratum 1, linearly compositional models need to divide the representation space into all possible combinations of concept values, $n^k$ of them. Each region within the $n^k$ partitions must contain the corresponding combination of concept values. Under linear factorization, the degrees of freedom of the embeddings within each cell are low, yielding an $R^2$ score of 1. However, even if linear factorization initially holds, the embeddings can generally be perturbed to violate the linear factorization constraint while still being contained within the correct cell.

To illustrate this point, we consider two general cases: (i) the number of concepts is equal to the dimension of the embeddings ($k = d$), and (ii) the number of concepts is less than the dimension of the embeddings ($k < d$). As detailed in Section 4.2, case (i) is tight (the dimension

cannot be further reduced), while case (ii) is not. In both cases we assume two concepts and an embedding space that admits two linear probes, one for each concept. Additionally, in both cases we illustrate separately the argmax regions where a certain concept value is predicted ($\mathcal{R}_{i,j}, i \in [2], j \in [n]$), and the region where a certain combination of concept values is predicted ($\mathcal{R}_{0,j} \cap \mathcal{R}_{1,k}, j, k \in [n]$, as per Desideratum 1).

The first concept values' regions in the embedding space are shown in blue, while the second concept values' regions are shown in orange.

**Case (i):** $k = d$. In Fig. 19, **(a)**, **(b)** we show two cases that exhibit perfect linear classification. In **(a)** a few outliers violate the linearity of the representation, which is also reflected in the $R^2 = 0.53 < 1$. In **(b)** the argmax regions are highly irregular, and the majority of the embeddings are almost intersecting the decision boundaries, resembling an extremely brittle embedding space susceptible to adversarial attacks, though the classification accuracy is still 100%.

**Case (ii):** $k < d$. In Fig. 19, **(k)**, **(d)**, **(e)** we show three cases that exhibit perfect linear classification, but with linearity scores ranging from $R^2 = 0.32$ to $R^2 = 0.83$. Because of the higher degrees of freedom, the embeddings enjoy even more space to be perturbed while still exhibiting perfect linear classification.

Overall, these points illustrate that linear factorization is not a trivial property of linearly compositional models, even when perfect classification holds.

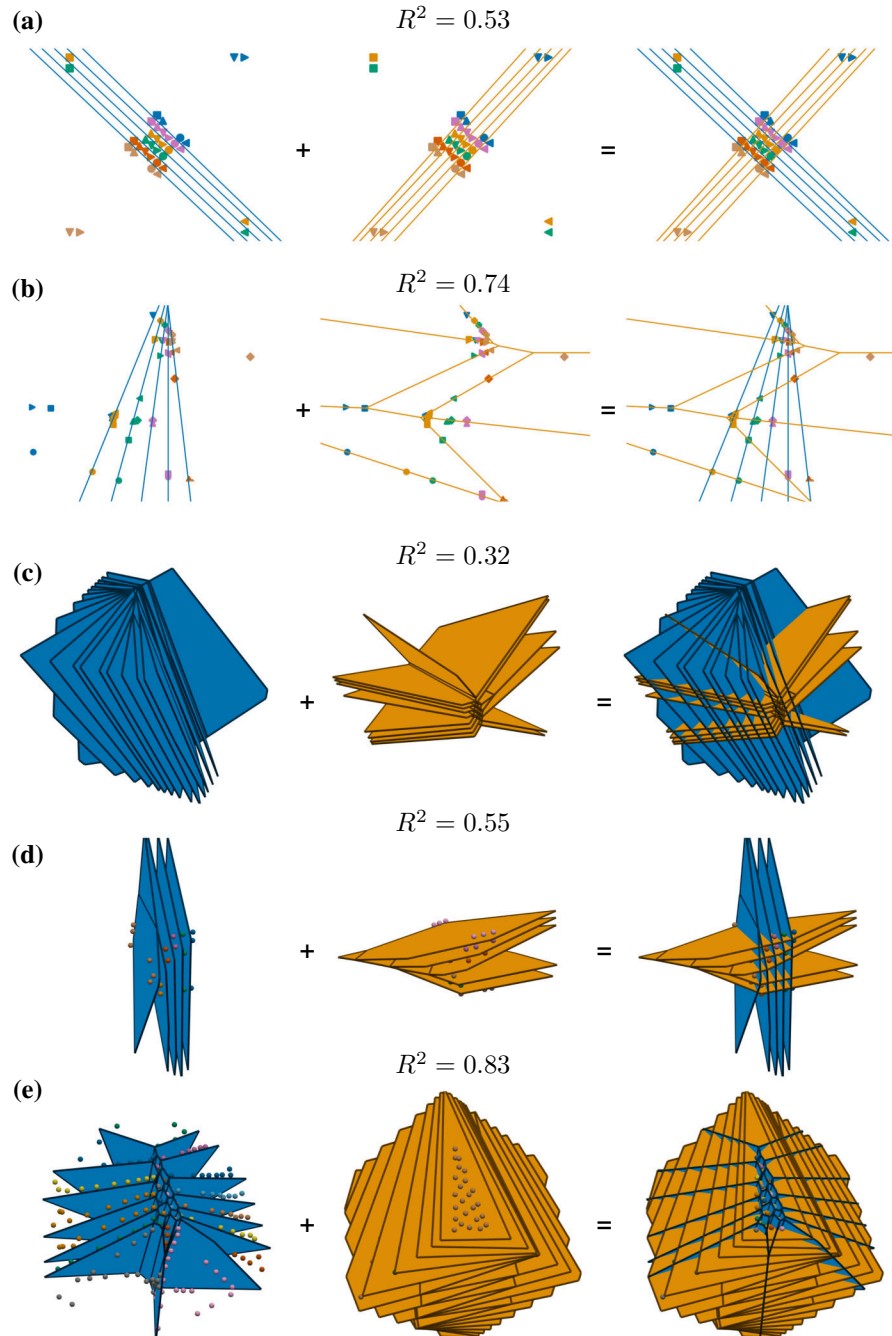

*Figure 19.* **Counterexamples of linear factorization under perfect classification**. The two concepts are linearly separable, but the representation cannot be decomposed into a sum of two concept components. Each subfigure shows the embedding space overlaid with three columns of argmax regions: the first column shows $\mathcal{R}_{0,i}, i \in [n]$ (shown in blue), the regions where the first concept values are predicted; the second column shows $\mathcal{R}_{1,j}, j \in [n]$ (shown in orange), the regions where the second concept values are predicted; and the third column shows $\mathcal{R}_{0,i} \cap \mathcal{R}_{1,j}, i, j \in [n]$, the joint argmax regions where specific combinations of concept values are predicted. **(a), (b)** show embeddings for two concepts (color and shape) in $\mathbb{R}^2$ ($k = d = 2$). **(d), (e)** show embedding points colored by the first concept value, all for two concepts in $\mathbb{R}^3$ ($k = 2, d = 3$). See text for details.

*Table 1.* **Datasets used in our main experiments.** Here $k$ is the number of concepts, $n_i$ is the number of values for concept $i$, and $N$ is the number of samples after the preprocessing described in the Notes column.

| Dataset | $k$ | Values per concept ($n_i$) | $N$ | Notes (main-experiment preprocessing) |
|---|---|---|---|---|
| PUG-Animal | 4 | `background(64)`, `character(68)`, `scale(3)`, `texture(3)` | 39,168 | We fix `camera-yaw=0`, drop the default character texture, and remove the `Goldfish` character. |
| dSprites | 6 | `color(10)`, `shape(3)`, `scale(6)`, `orient(10)`, `posX(10)`, `posY(10)` | 180,000 | We keep orientations in $[0°, 90°)$ (10 bins), down-sample `posX`/`posY` to 10 bins each, and add 10 colors via on-the-fly coloring. |
| MPI3D | 7 | `obj-color(6)`, `obj-shape(6)`, `obj-size(2)`, `cam-height(3)`, `bg-color(3)`, `horiz(10)`, `vert(10)` | 64,800 | We use the real-world complex variant (original factor sizes `[6,6,2,3,3,40,40]`) and down-sample the horizontal/vertical axes to 10 bins each. |

## G.5. Models and probing

We evaluate a broad model set spanning contrastive vision-language and self-supervised vision encoders: OpenAI CLIP (Radford et al., 2021), OpenCLIP, MetaCLIP (Xu et al., 2023), and MetaCLIP2 (Chuang et al., 2025) checkpoints from the LAION ecosystem (Schuhmann et al., 2021a), SigLIP and SigLIP 2 (Zhai et al., 2023; Tschannen et al., 2025), and DINOv1–v3 (Caron et al., 2021; Siméoni et al., 2025). We evaluate on three compositional datasets: `PUG-Animal` (Bordes et al., 2023), `dSprites` (Matthey et al., 2017), and `MPI3D` (Gondal et al., 2019), plus ImageNet-AO (Abbasi et al., 2024) (Appendix H.4.2).

# H. Additional experimental results

In this section we provide additional experimental results discussed in the main text. We used (Rubinstein & Uselis, 2025) for conducting the experiments.

## H.1. Linearity and compositional generalization

We provide an extended view of the relationship between projected linearity ($R^2$) and compositional generalization under different train/test splits. We follow the same pipeline as in Section 5.2: for each dataset/model, we train one linear probe per concept and evaluate average compositional accuracy on held-out concept combinations. The full split-ablation results are shown in Fig. 21: across held-out fractions, higher projected $R^2$ is still associated with higher held-out accuracy, while the trend flattens in near-ceiling regimes (especially on dSprites and MPI3D).

For each split level, we set a test fraction $\rho_{\text{test}} \in \{0.95, 0.90, 0.50, 0.30, 0.10\}$ and train on the remaining $1 - \rho_{\text{test}}$ fraction. For every model/split setting, probes are trained with Adam for 1000 epochs using cosine decay to zero, and we select the best result over learning rates $\{10^{-3}, 10^{-2}\}$.

*Table 2.* **Vision models used in our experiments.**

| Family | Model identifier | Pretraining tag | Params (M) | Hidden |
|---|---|---|---|---|
| `openaiclip` | | | | |
| clip | ViT-B/32 | openai | 151.3M | 512 |
| clip | ViT-L/14 | openai | 427.6M | 768 |
| *openclip* | | | | |
| `metaclip2` | | | | |
| openclip | ViT-H/14 (MetaCLIP2 Worldwide, 378px) | metaclip2_worldwide | 1859.4M | 1024 |
| openclip | ViT-H/14 (MetaCLIP2 Worldwide, QuickGELU) | metaclip2_worldwide | 1858.8M | 1024 |
| openclip | ViT-G/14 (bigG, MetaCLIP2 Worldwide) | metaclip2_worldwide | 3630.4M | 1280 |
| openclip | ViT-G/14 (bigG, MetaCLIP2 Worldwide, 378px) | metaclip2_worldwide | 3631.2M | 1280 |
| `siglip2` | | | | |
| openclip | SigLIP2-B/16 (224) | webli | 375.2M | 768 |
| openclip | SigLIP2-B/16 (256) | webli | 375.2M | 768 |
| openclip | SigLIP2-L/16 (256) | webli | 881.5M | 1024 |
| openclip | SigLIP2-L/16 (384) | webli | 881.9M | 1024 |
| openclip | SigLIP2-SO400M/16 (256) | webli | 1135.7M | 1152 |
| `siglip` | | | | |
| openclip | SigLIP-B/16 (224) | webli | 203.2M | 768 |
| openclip | SigLIP-B/16 (256) | webli | 203.2M | 768 |
| openclip | SigLIP-L/16 (256) | webli | 652.2M | 1024 |
| `openclip` | | | | |
| openclip | ViT-B/16 | laion400m_e32 | 149.6M | 512 |
| openclip | ViT-B/32 | laion400m_e32 | 151.3M | 512 |
| openclip | ViT-L/14 | laion2b_s32b_b82k | 427.6M | 768 |
| *dino* | | | | |
| `dinov3` | | | | |
| dino | DINOv3 ViT-S/16 | timm_default | 21.6M | 384 |
| dino | DINOv3 ViT-B/16 | timm_default | 85.6M | 768 |
| dino | DINOv3 ViT-L/16 | timm_default | 303.1M | 1024 |
| `dinov2` | | | | |
| dino | DINOv2 ViT-S/14 | timm_default | 22.1M | 384 |
| dino | DINOv2 ViT-B/14 | timm_default | 86.6M | 768 |
| dino | DINOv2 ViT-L/14 | timm_default | 304.4M | 1024 |
| `dinov1` | | | | |
| dino | DINO ViT-S/16, 224px | timm_default | 21.7M | 384 |
| dino | DINO ViT-B/16, 224px | timm_default | 85.8M | 768 |
| *timm* | | | | |
| `metaclip` | | | | |
| timm | ViT-B/16 CLIP (MetaCLIP 2.5B, 224px) | timm_default | 86.2M | 768 |
| timm | ViT-B/32 CLIP (MetaCLIP 2.5B, 224px) | timm_default | 87.8M | 768 |
| timm | ViT-L/14 CLIP (MetaCLIP 2.5B, 224px) | timm_default | 304.0M | 1024 |

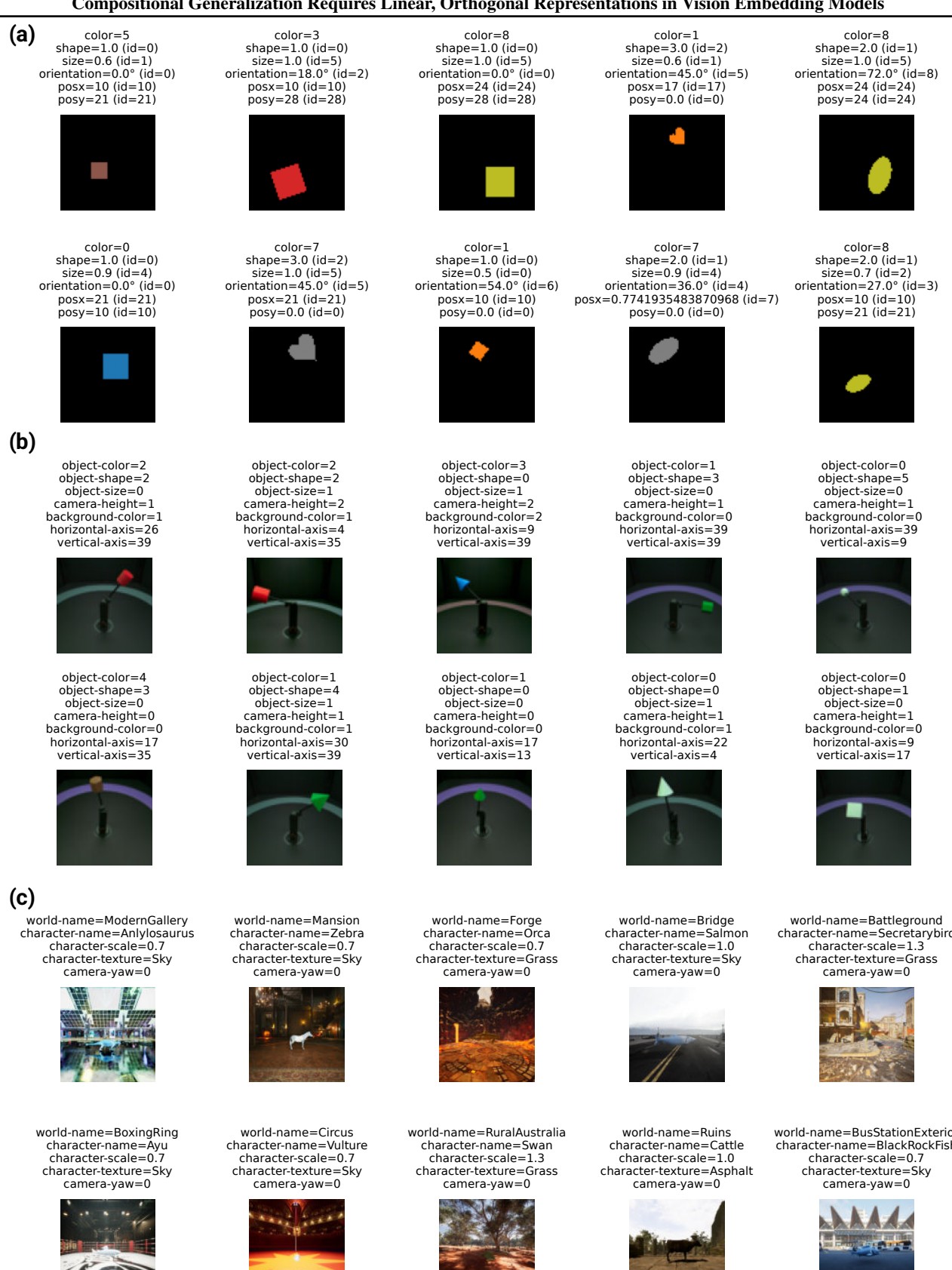

*Figure 20.* **Example samples from the main datasets used in our experiments. (a)** dSprites, **(b)** MPI3D, **(c)** PUG-Animal.

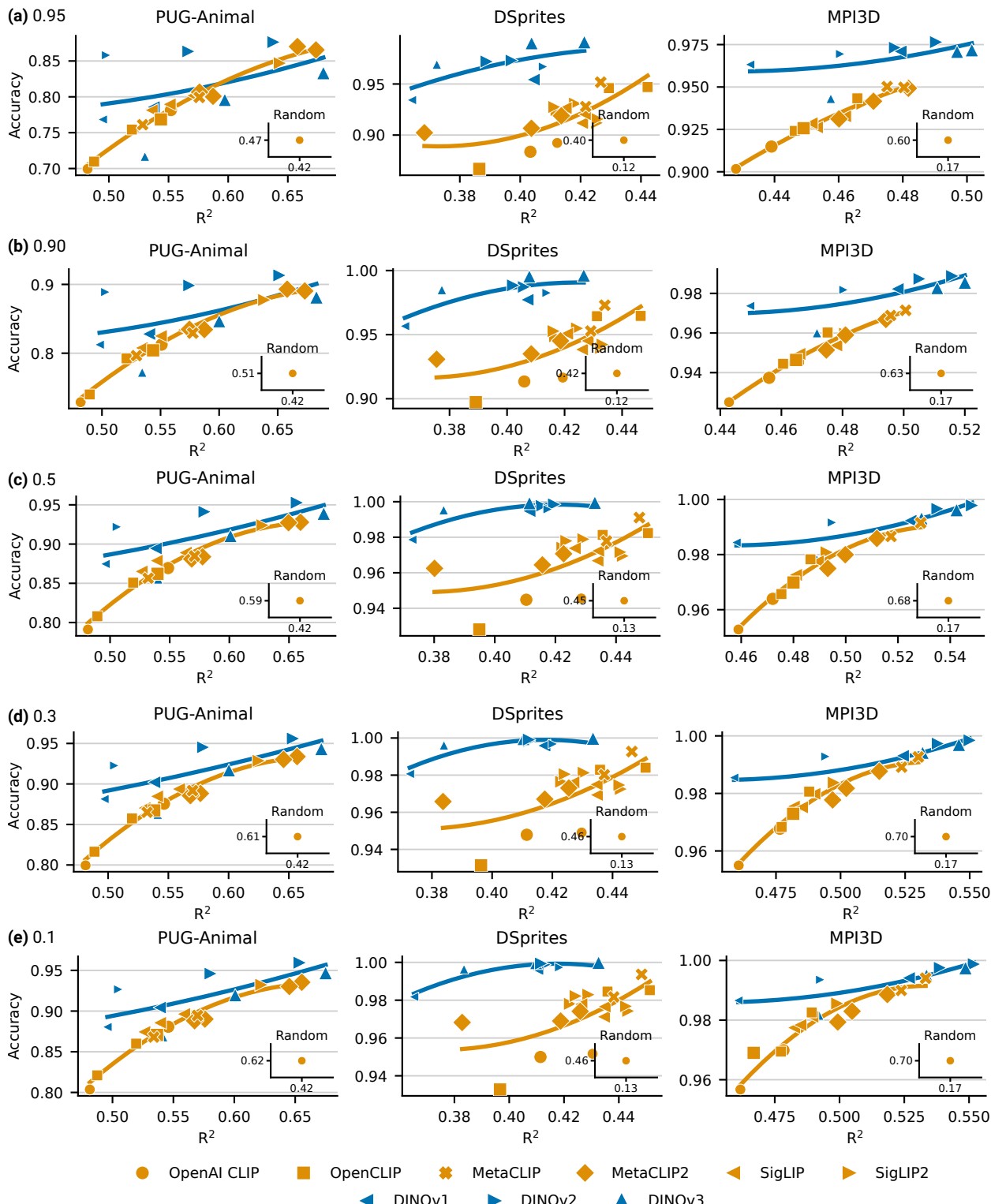

*Figure 21.* **Projected $R^2$ vs compositional accuracy across split regimes.** Columns correspond to datasets (PUG-Animal, dSprites, MPI3D). Rows **(a)–(e)** correspond to different held-out test fractions $\rho_{\text{test}}$ (shown next to each row). The x-axis is projected $R^2$ and the y-axis is compositional accuracy on held-out combinations. Each marker is a model checkpoint (legend); the inset in each panel reports the randomly initialized OpenCLIP baseline for the same dataset/split. Across split regimes, higher projected $R^2$ is consistently associated with higher held-out compositional accuracy.

## H.2. Orthogonality of factors

**Setup.** For each dataset/model, we extract image embeddings $z_c$ and recover concept factors $\{u_{i,v}\}$ as in Section 5.1. For orthogonality, we compute all quantities in the original embedding space (without projection by $P_W$). For each concept $i$, we define centered and normalized factor directions:

$$\bar{u}_i := \frac{1}{|\mathcal{C}_i|} \sum_{v \in \mathcal{C}_i} u_{i,v}, \tag{97}$$

$$d_{i,v} := u_{i,v} - \bar{u}_i, \qquad \tilde{d}_{i,v} := \frac{d_{i,v}}{\|d_{i,v}\|}. \tag{98}$$

**Metrics.** We measure orthogonality via absolute cosine between direction vectors (lower $|\cos| \Rightarrow$ greater orthogonality). For any concepts $i \neq j$, we define

$$\mathrm{Orth}(i,j) := \frac{1}{|\mathcal{C}_i||\mathcal{C}_j|} \sum_{a \in \mathcal{C}_i} \sum_{b \in \mathcal{C}_j} |\tilde{d}_{i,a}^{\mathsf{T}} \tilde{d}_{j,b}|,$$

$$\mathrm{Orth}(i,i) := \frac{1}{|\mathcal{C}_i|(|\mathcal{C}_i| - 1)} \sum_{\substack{a,b \in \mathcal{C}_i \\ a \neq b}} |\tilde{d}_{i,a}^{\mathsf{T}} \tilde{d}_{i,b}|.$$

We report $\mathrm{Orth}(i,i)$ as *within-concept direction similarity* and $\mathrm{Orth}(i,j)$ (for $i \neq j$) as *across-concept direction similarity*. In this parameterization, stronger cross-concept orthogonality corresponds to lower $\mathrm{Orth}(i,j)$.

**Results.** In Fig. 22, we provide the full per-model/per-dataset orthogonality matrices (including a randomly-initialized baseline) across the three datasets used in the main text.

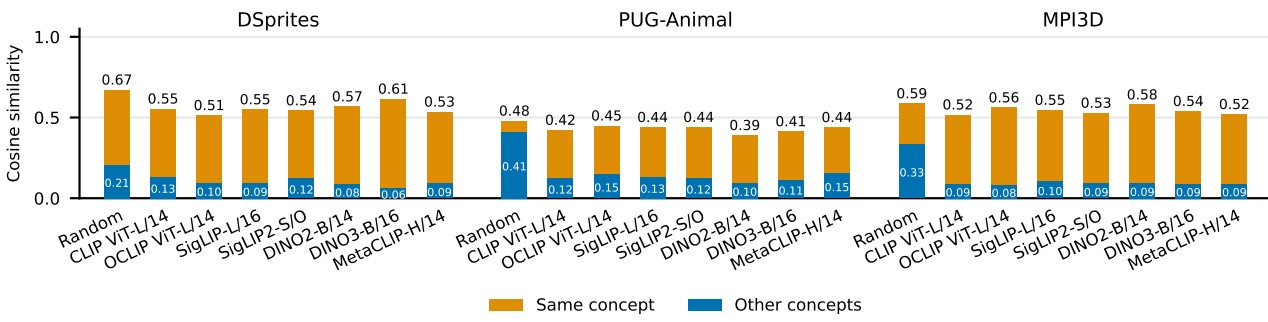

*Figure 22.* **Full orthogonality matrices across models and datasets.** Each entry reports mean absolute cosine between centered factor directions. Diagonal blocks correspond to within-concept similarity; off-diagonal blocks correspond to across-concept similarity. Lower off-diagonal values indicate stronger cross-concept orthogonality.

We summarize the same trend in aggregate form in Fig. 23, comparing within-concept and across-concept values directly.

*Figure 23.* **Aggregated orthogonality summary.** Bars compare within-concept direction similarity and across-concept similarity; across-concept values are consistently lower, matching the orthogonality pattern predicted by the theory.

## H.3. Dimensionality of factors

Our theory predicts that generalizing linear compositional models require linear factorization of embeddings into per-concept components. When many concepts must coexist in a fixed embedding dimension, each concept's subspace should be low-rank to enable efficient packing (Section 5.1). Here, we investigate to which extent concept factors in pretrained models are low-dimensional.

**Metrics and setup.** We study factor geometry (with factor recovery following Section 5.1). For each concept $i \in [k]$ with value set $\mathcal{C}_i$ ($n_i = |\mathcal{C}_i|$), we aggregate the per-concept factors $\boldsymbol{u}_{i,j}$ for $j \in \mathcal{C}_i$ into a matrix $\boldsymbol{U}_i \in \mathbb{R}^{n_i \times d}$. We then analyze (1) the dimensionality of each concept and (2) how this dimensionality compares across models. To do so, we examine the spectrum of $\boldsymbol{U}_i$ (PCA on its rows) and report the number of principal components required to explain 95% of the variance across values $j$.

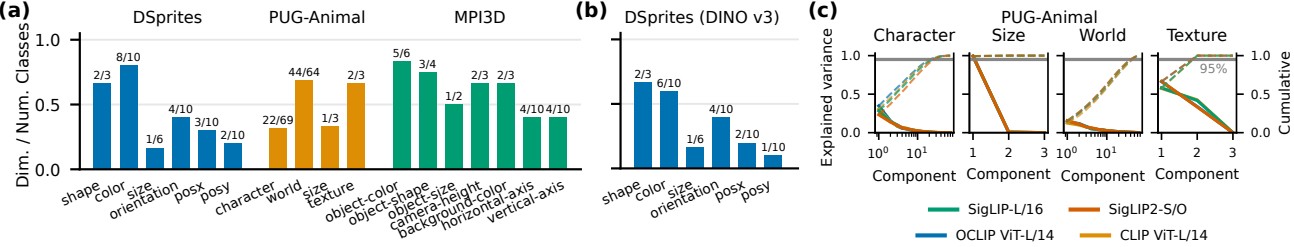

*Figure 24.* **Dimensionality of factors. (a)** Normalized effective rank across datasets and concepts under OpenCLIP ViT-L/14; text above each bar reports "effective dimension / number of values" for that concept. **(b)** Same analysis as in (a), shown for DINOv3 on dSprites; the recovered factors are typically lower-rank (variance concentrates in fewer PCs). **(c)** Cumulative explained variance of recovered PUG-Animal factors across model families (CLIP, OpenCLIP, SigLIP, SigLIP2); for each concept, the curves are nearly overlapping, indicating strong cross-model similarity in factor geometry.

**Results.** Fig. 24 shows that most ordinal factors lie in low-dimensional subspaces relative to their cardinality (e.g., dSprites size $1/6$, MPI3D vertical-axis $4/10$). Panel (b) shows the same tendency in a DINOv3-specific view on dSprites: factor variance typically concentrates in a small number of principal components. Panel (c) shows that this structure is highly consistent across architectures: for each PUG-Animal concept, explained-variance curves from different model families are nearly indistinguishable. This cross-model similarity is consistent with the Platonic Representation Hypothesis (Huh et al., 2024) and with recent evidence that independently trained multimodal contrastive models can often be aligned by near-orthogonal transforms (Gupta et al., 2026). Across datasets and models, $\geq 95\%$ of variance is typically captured by one or two PCs, indicating that spectra align closely by concept. Discrete concepts show higher rank, potentially due to being composed of more atomic attributes. Overall, semantic factors are low-rank and geometrically similar across models, while discrete concepts are not strictly low-rank.

We also visualize dSprites factors (orientation, size, $y$-position) in Fig. 25. Each subspace is effectively $<3D$ ($\geq 95\%$ variance in $\leq 2$ PCs). Size and $y$-position trace near-1D path, while orientation forms a smooth 2D curve with small curvature, matching the effective dimensions in Fig. 24.

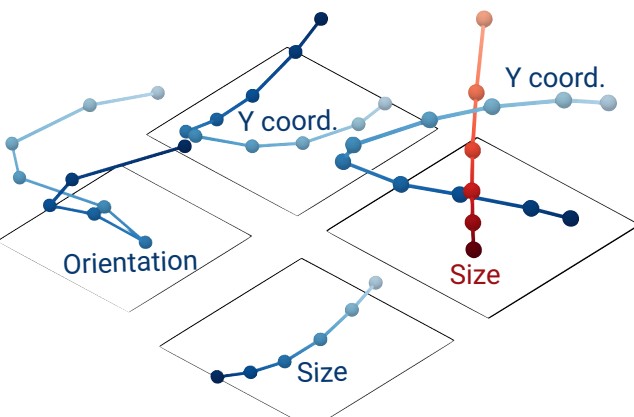

*Figure 25.* **Geometry of *factors* $\{\boldsymbol{u}_{i,j}\}$ in OpenCLIP ViT-L/14.** The factors are often low dimensional and near co-linear within a concept. Across concepts, the factors are near-orthogonal.

### H.3.1. FULL PER-MODEL RESULTS

For each model and dataset, we estimate per-concept effective dimensionality as the minimum number of PCA components needed to explain 95% variance in recovered factors, and report it relative to concept cardinality. The complete per-model plots are shown in Fig. 26.

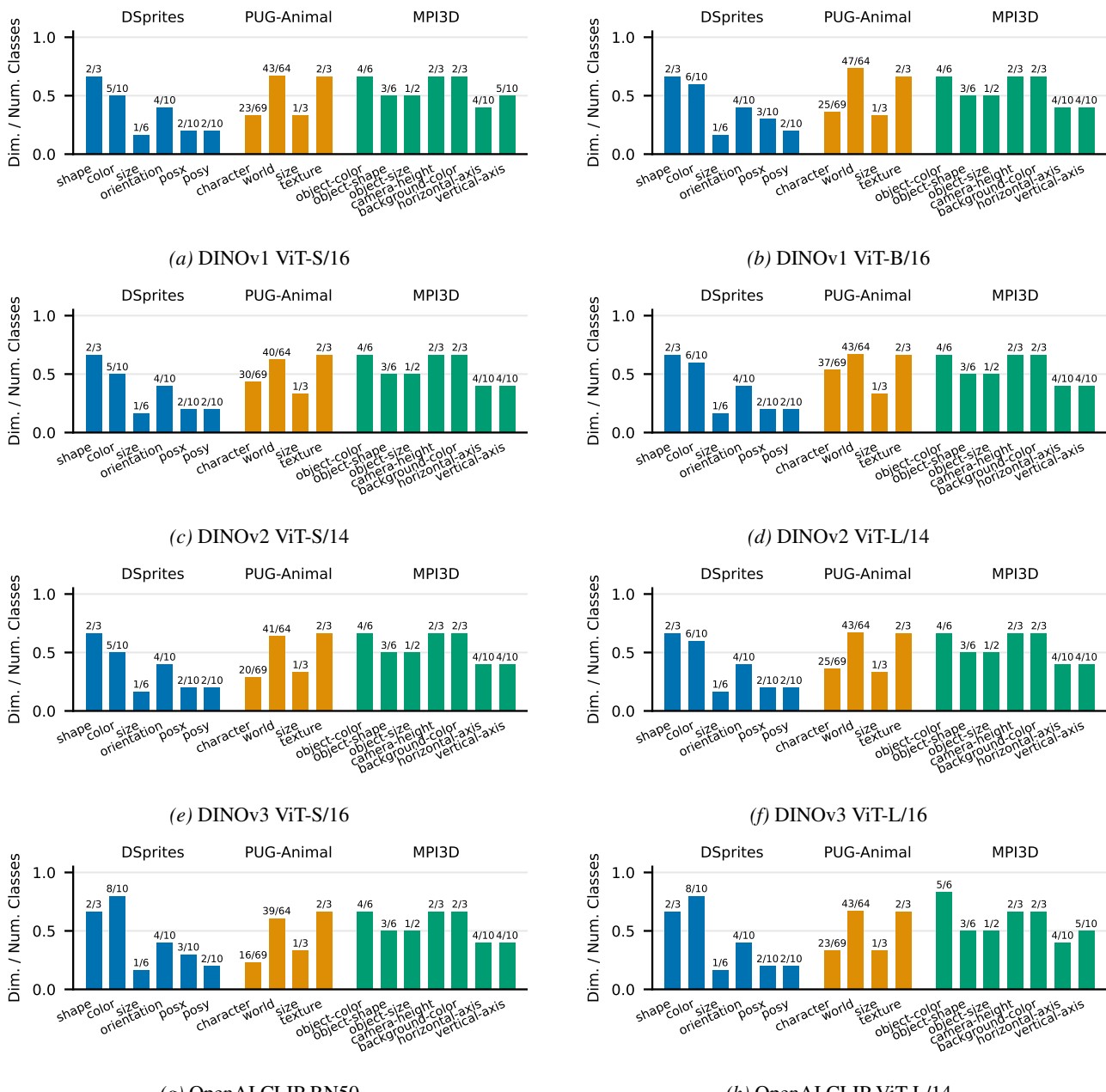

*Figure 26.* Dimensionality results computed as the number of SVD factors required to reach 95% explained variance, per dataset, across representative vision(-language) backbones.

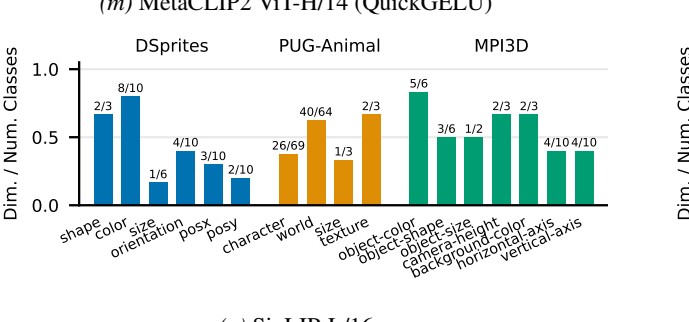

*(i)* OpenCLIP ViT-B/16 (LAION-400M)

*(k)* MetaCLIP 2.5B ViT-B/32

*(m)* MetaCLIP2 ViT-H/14 (QuickGELU)

*(o)* SigLIP-L/16

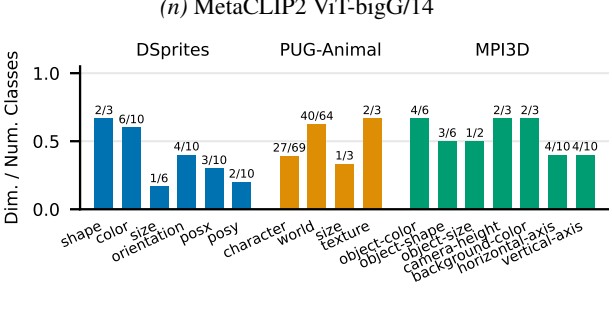

*(j)* OpenCLIP ViT-L/14 (LAION-2B)

*(l)* MetaCLIP 2.5B ViT-L/14

*(n)* MetaCLIP2 ViT-bigG/14

*(p)* SigLIP2 SO400M/16 (256)

*Figure 26.* Dimensionality results (continued from Fig. 26).

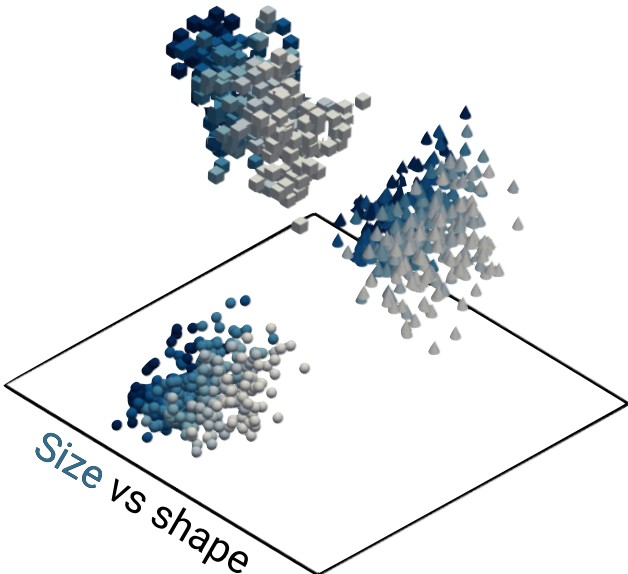

*Figure 27.* **Geometry of *datapoints* in OpenCLIP ViT-L/14.** We show the span of the joint features of OpenCLIP ViT-L/14.

## H.4. Experiments using text encoders as probes (zero-shot results)

In the main text (Section 5.1), we analyzed the factors of the models by training linear probes on the image embeddings using gradient descent with cross-entropy. This was done for two reasons: (1) to handle concepts that are difficult to express as text prompts (e.g., visually complex backgrounds or continuous attributes like size or orientation), and (2) to avoid potential misalignment between the text and vision modalities, where the text encoder must accommodate many visual categories, potentially leading to suboptimal performance for certain domains. Here, we ask what happens when we do not take into account these problems and instead rely on the linear probes that the text encoder already produces.

In this section, we provide analogous analyses to those in the main text, but using the text encoder as probes instead of external linear probes for two datasets: PUG-Animal and ImageNet-AO. We use these datasets for two reasons: (1) their concepts and values map naturally to text prompts, and (2) the datasets were released after the CLIP models and exhibit many unnatural concept combinations unlikely to have appeared in text captions during pre-training, and not present in the visual training data.

### H.4.1. EXPERIMENTS ON PUG-ANIMAL

**Setup.** Four concepts are exposed: character, background, scale, and texture. For each character we parse the character name into a set of words and use prompts of the form "A picture of a <character>". For each background, we use prompts of the form "A picture of a <background>" (detailed in Tab. 3).

We map numeric scale values and texture labels to descriptive prompt templates for evaluating the models. Specifically, for scale, we use:

- `0.7` → "A picture of a small object"

- `1.0` → "A picture of a medium-sized object"

- `1.3` → "A picture of a large object"

For textures, we use the following mappings:

- "Sky" → "A picture of an object in sky texture"

- "Grass" → "A picture of an object in grass texture"

- "Asphalt" → "A picture of an object in asphalt texture"

These prompt templates are used to generate the corresponding text embeddings for each concept, matching exactly with the setup of the experiments in the main text.

*Table 3.* **Mapping from class names to clean prompt names for PUG-Animal experiments.**

| Original Name | Prompt Name |
| --- | --- |
| Desert | a desert |
| Tableland | a tableland |
| EuropeanStreet | a European street |
| OceanFloor | the ocean floor |
| Racetrack | a racetrack |
| Ruins | ancient ruins |
| TrainStation | a train station |
| BusStationInterior | the interior of a bus station |
| BusStationExterior | the exterior of a bus station |
| IndoorStairs | indoor stairs |
| Circus | a circus |
| BoxingRing | a boxing ring |
| Mansion | a mansion |
| ShoppingMall | a shopping mall |
| ConferenceRoom | a conference room |
| VillageOutskirt | a village outskirt |
| VillageSquare | a village square |
| Courtyard | a courtyard |
| Forge | a forge |
| Library | a library |
| Museum | a museum |
| Gallery | an art gallery |
| Opera | an opera house |
| Restaurant | a restaurant |
| RuralAustralia | rural Australia |
| AustraliaRoad | a road in Australia |
| ShadyRoad | a shady road |
| SaltFlats | salt flats |
| Castle | a castle |
| Temple | a temple |
| Snow | a snowy landscape |
| Grass | a grassy field |
| DryGrass | a dry grassland |
| Forest | a forest |

Concretely, for each concept value $j \in [n]$, we pass the prompt template through the text encoder $g$ to obtain a ($\ell_2$-normalized) probe vector $\boldsymbol{w}_{i,j} = g(p_{i,j}) \in \mathcal{Z}$, as detailed in Section 3.4.

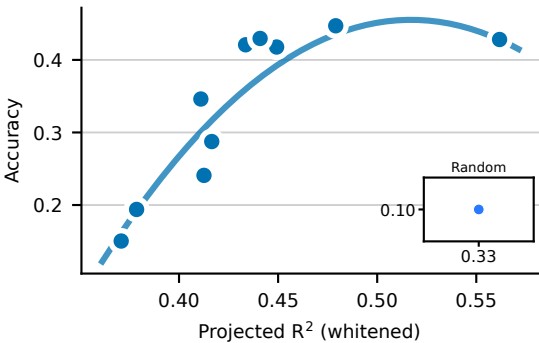

*Figure 28.* **Projected $R^2$ vs accuracy on PUG-Animal across models.** Higher projected $R^2$ coincides with higher accuracy on the full dataset. The probes are extracted from the text encoder.

**Linearity of factors and generalization.** We show the projected $R^2$ and average accuracy on all concept combinations on PUG-Animal across models in Fig. 28 when using the text encoder as probes. Models exhibiting higher linearity of representations generally exhibit higher accuracy on the full dataset. This coincides with the observations in the main text (Section 5.1); random baseline achieves low projected $R^2$ and accuracy.

**Orthogonality of the factors.** For each of the concepts, we compute the linear factors as detailed in the main text (Section 5.1) with the text encoder as probes. We compute the within- and across-concept orthogonality as detailed in Appendix H.2 and illustrate the results in Fig. 29 for each of the models.

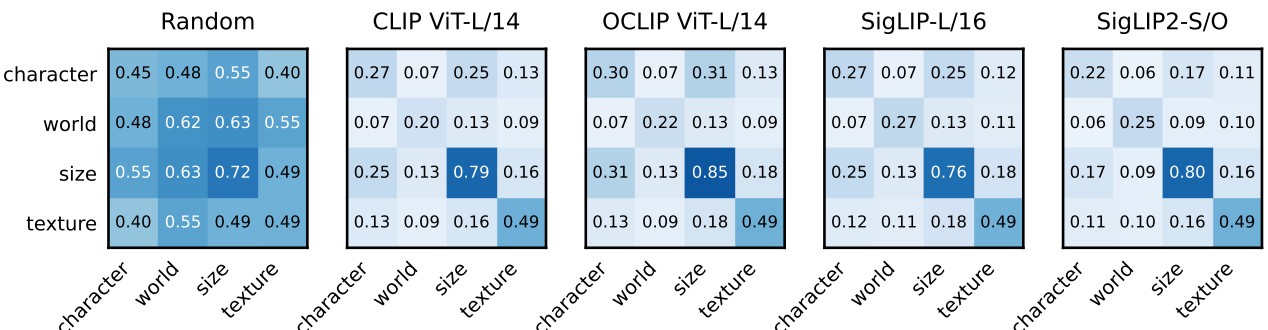

*Figure 29.* **Orthogonality of the factors on PUG-Animal.** Heatmaps show pairwise cosine similarity between factors for the four PUG-Animal concepts (character, world, size, texture) across multiple models. The factors are more orthogonal across concepts (off-diagonal) than within concepts (diagonal). The random baseline does not generally show this pattern.

For all evaluated models, we observe the same orthogonality pattern: the factors are more orthogonal across concepts (off-diagonal) than within concepts (diagonal). The average cosine similarity for the random baseline is higher (around 0.5) both within and across concepts.

We also note the qualitative similarity between the factors to the case when probes were trained on $90\%$ of the concept combinations (Fig. 22, second row).

**Qualitative examples.** We illustrate some of the highest- and lowest-scoring samples in terms of $R^2$ for the SigLIP2 model in Fig. 30. We note that high-scoring samples generally depict clean scenes where the character and its size and texture are easier to discern compared to the lower-scoring samples.

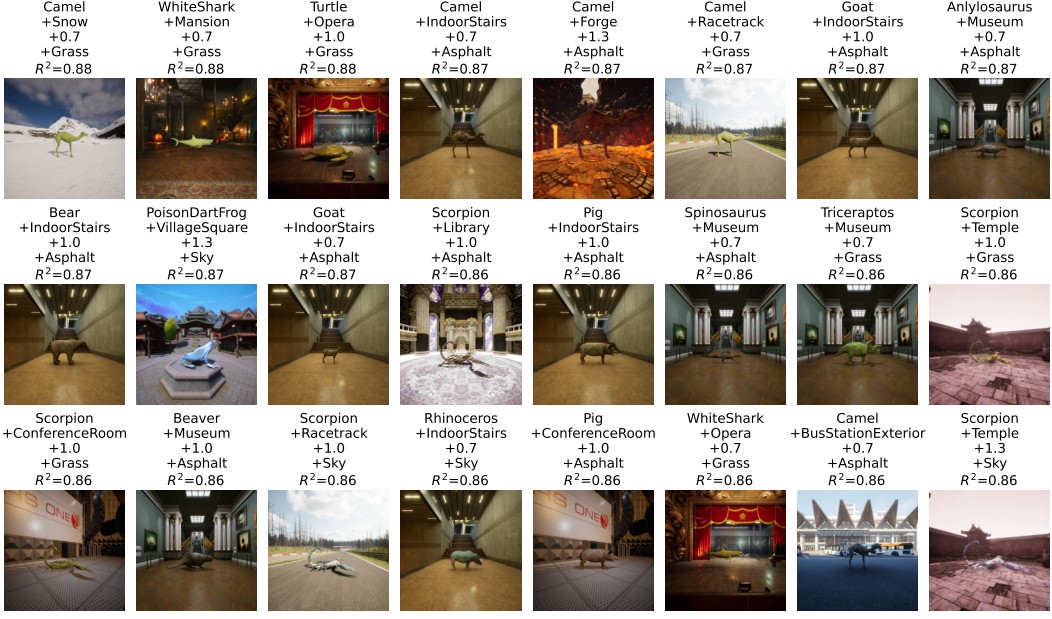

*(a)* Top-scoring samples in terms of $R^2$.

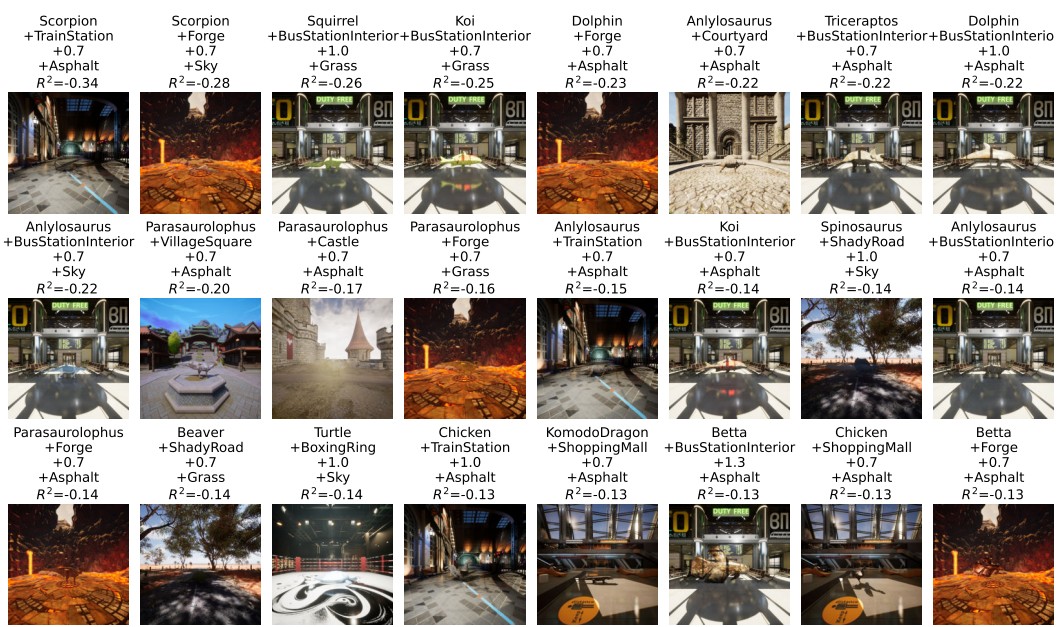

*(b)* Bottom-scoring samples in terms of $R^2$.

*Figure 30.* **Qualitative examples of the top- and lowest-scoring samples in PUG-Animal for the SigLIP2 model.** Each sample shows its character name, world name, size value (0.7 corresponds to "small", 1.0 corresponds to "medium", 1.3 corresponds to "large"), texture name, and its $R^2$ score.

### H.4.2. EXPERIMENTS ON IMAGENET-AO

We additionally perform experiments on a coarse-captioned dataset ImageNet-AO Abbasi et al. (2024), where each image sample has an associated caption composed of an adjective and a noun.

The experiments here are slightly dissimilar from the main experiments in Section 5.1, for a few reasons: (1) scarcity of per-combination data, (2) inability to train linear probes, (3) noisy/ambiguous data, and (4) coarse categories. Regardless, our framework still applies.

**Dataset description.** The dataset contains images described by an adjective and a noun. There are around 80 unique adjectives and over 600 unique nouns. To make the analysis balanced, we work with the dataset restricted to the most common 80 nouns and adjectives. Each potential combination of adjective and noun may have between 0 and 6 images. The dataset is thus sparse, and many of the potential combinations are not observed in the dataset. This results in a total of 3243 datapoints. We illustrate the sparsity and the pairs we work with in Fig. 31.

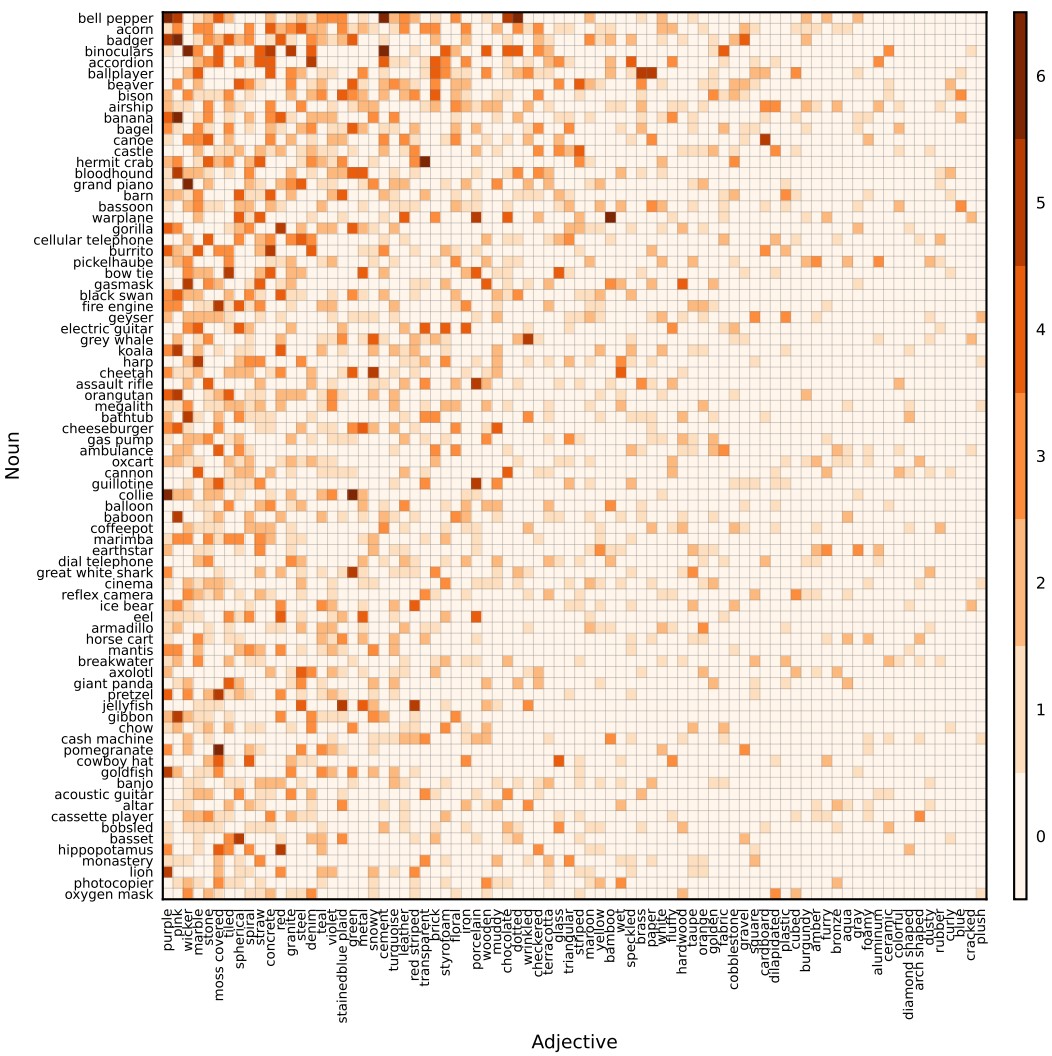

*Figure 31.* **Adjective-noun count matrix for ImageNet-AO (Abbasi et al., 2024) of the top 80 adjectives and nouns.** The adjective-noun pairs are sparse, and many of them are not observed in the dataset.

**General setup.** Due to limited availability of the data samples, we *do not* train linear probes. Because each sample is associated with a (noun, adjective) combination, we instead use the probes from the text encoder to assess the performance of the models (as detailed in the main text in Section 3.4). Concretely, we pass captions in the style of "A picture of <noun>" in the case of noun, and "A picture showing <adjective>" in the case of adjective, through the text encoder.

Because of imbalance and sparsity, we cannot rely on averaging to extract the factors as done in Section 5.1. Instead, we follow (Uselis et al., 2025) and solve a linear system of equations to recover the factors. Concretely, we construct a design matrix $\boldsymbol{A} \in \{0, 1\}^{3243 \times 80 \cdot 2}$ where each row corresponds to a sample, and each column corresponds to either the presence of a noun (if the column index $< 80$) or the presence of an adjective (if the column index $\geq 80$). The matrix was of full rank $2 \cdot 80 - 1$. Then, we solve the linear system

$$\boldsymbol{A} \begin{bmatrix} \boldsymbol{u}_{\text{noun}} \\ \boldsymbol{u}_{\text{adj}} \end{bmatrix} = \boldsymbol{X}$$ to recover the factors $\boldsymbol{u}_{\text{noun}} \in \mathbb{R}^{80 \times d}$ and $\boldsymbol{u}_{\text{adj}} \in \mathbb{R}^{80 \times d}$, where $d$ is the dimension of the representation space, and $\boldsymbol{X} \in \mathbb{R}^{3243 \times d}$ is the centered image embeddings. We show the whitened $R^2$ scores. The remaining procedure in the analysis follows Section 5.1.

**Linearity of factors and generalization.** We show the projected $R^2$ vs accuracy on ImageNet-AO across models in Fig. 32. As seen in

the main text (Section 5.1), higher projected $R^2$ coincides with higher accuracy on the full dataset. Importantly, the random baseline achieves substantially lower projected $R^2$ (less than 0.1) compared to the other models.

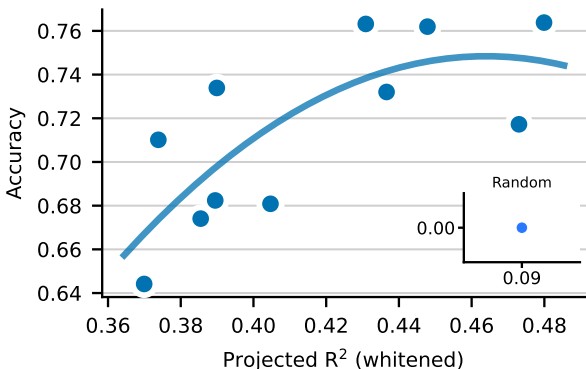

Figure 32. **Projected $R^2$ vs accuracy on ImageNet-AO across models.** Higher projected $R^2$ coincides with higher accuracy on the full dataset. Linear probes were not trained here, and the results are computed using the text encoder.

**Orthogonality of the factors.** To substantiate the claims of orthogonality of factors across concepts, we extract the factors for all the models as detailed in the setup above. Concretely, for each of the attribute factor $u_i, i \in [80]$ and noun factor $u_j, j \in [80]$, within- and across-concept orthogonality as detailed in Section 5.1.

We illustrate the results in Fig. 33. For all of the evaluated models the same pattern of orthogonality is observed: the factors are more orthogonal across concepts than they are within concepts. For example, for the CLIP ViT-L/14 model, the within-concept similarity on average is 0.10 between nouns, and 0.14 between adjectives, while the average cosine similarity across concepts is 0.07. The random baseline on average yields 0.49 cosine similarity both across and within concepts.

Interestingly, all of the non-random models exhibit surprising degree of similarity in terms of the cosine similarities. For example, CLIP ViT-L/14 and OpenCLIP ViT-L/14 on average exhibit almost the same cosine similarity within and across concepts, differing only in the noun-noun cosine similarity (0.10 vs 0.11, respectively). These results support the notions of universality between models as argued by the Platonic Representation Hypothesis (Huh et al., 2024), and empirically observed in Universal Sparse Autoencoders (Thasarathan et al., 2025).

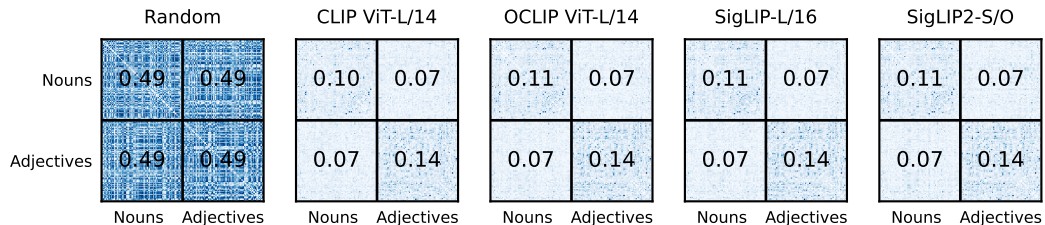

Figure 33. **Orthogonality of the factors on ImageNet-AO.** We show the cosine similarity of the factors for the SigLIP2 model on ImageNet-AO; we separate the first concept (nouns) from the second concept (adjectives) and show average similarity across each $2 \times 2$ block. The factors are more orthogonal across concepts than they are within concepts. The random baseline does not show this pattern.

**Qualitative examples.** To understand the results deeper, we show the qualitative examples of the top- and lowest-scoring samples in ImageNet-AO for the SigLIP2 model in Fig. 34. The top-scoring samples show high degree of projected $R^2$ scores (generally $> 0.75$), and correctly depict the adjective and noun of the sample. Even there, however, some samples are incorrectly predicted by the model, suggesting a potential lack of alignment between the image and text encoders[5].

The lowest-scoring samples show low degree of projected $R^2$ scores (generally $< 0.10$), and are often incorrectly predicted by the model. Few of the samples appear to be incorrectly labeled (e.g. first image depicting a orangutan as a gorilla), while some are correctly classified by the model but show a lack of factorization.

---

[5]This was less of an issue in the main experiments because the image embeddings were analyzed using linear probes.

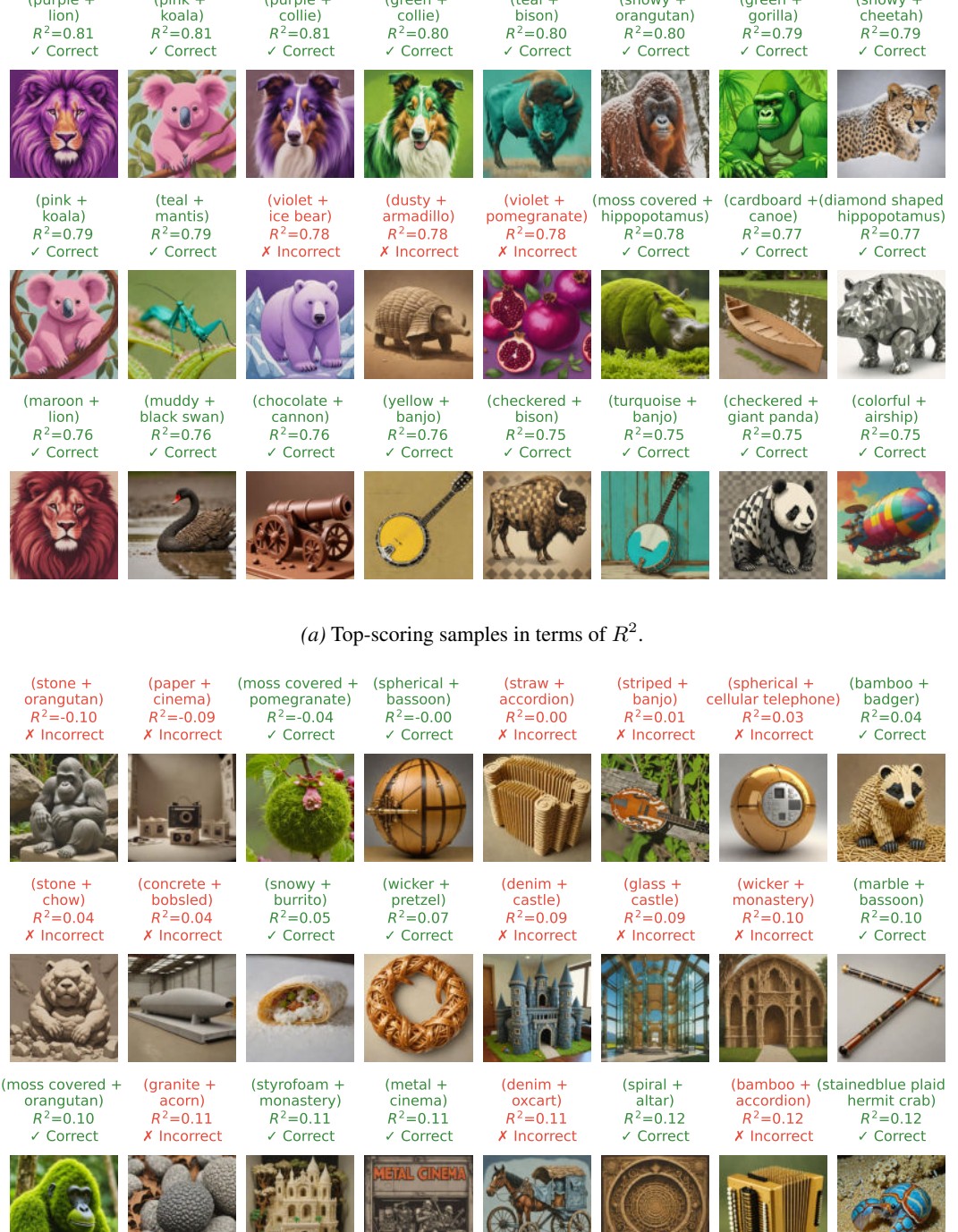

*(a)* Top-scoring samples in terms of $R^2$.

*(b)* Bottom-scoring samples in terms of $R^2$.

*Figure 34.* **Qualitative examples of the top- and lowest-scoring samples in ImageNet-AO for the SigLIP2 model.** Each sample shows its adjective and noun, its $R^2$ score, and whether it was correctly classified by the model. Note that both top- and lowest-scoring samples may be either correctly or incorrectly classified by the model.

## H.5. Robustness across train fractions, error bars, and residual structure

This subsection collects additional analyses that complement the main-text experiments: (i) the projected $R^2$ vs. compositional accuracy correlation across a wide range of train fractions, (ii) the behavior of projected $R^2$ as a function of train fraction itself, (iii) orthogonality metrics with error bars and on the full $32\times32$ dSprites grid, and (iv) a spectral analysis of the residuals after subtracting the recovered linear factorization.

### H.5.1. CORRELATION ACROSS TRAIN FRACTIONS ($|S|/N \in \{0.001, \dots, 0.9\}$)

We replicate the projected $R^2$ vs. compositional accuracy analysis of Section 5.2 across train fractions $|S|/N \in \{0.001, 0.003, 0.007, 0.01, 0.03, 0.07, 0.1, 0.3, 0.7, 0.9, 0.95, 0.99, 0.999\}$ over 3 seeds. Figs. 35 and 36 show that the positive correlation between projected $R^2$ and compositional accuracy persists across *all* train fractions, including the extreme low-data regime. The rightmost column in each row reports results on dSprites with the full $32\times32$ position grid.

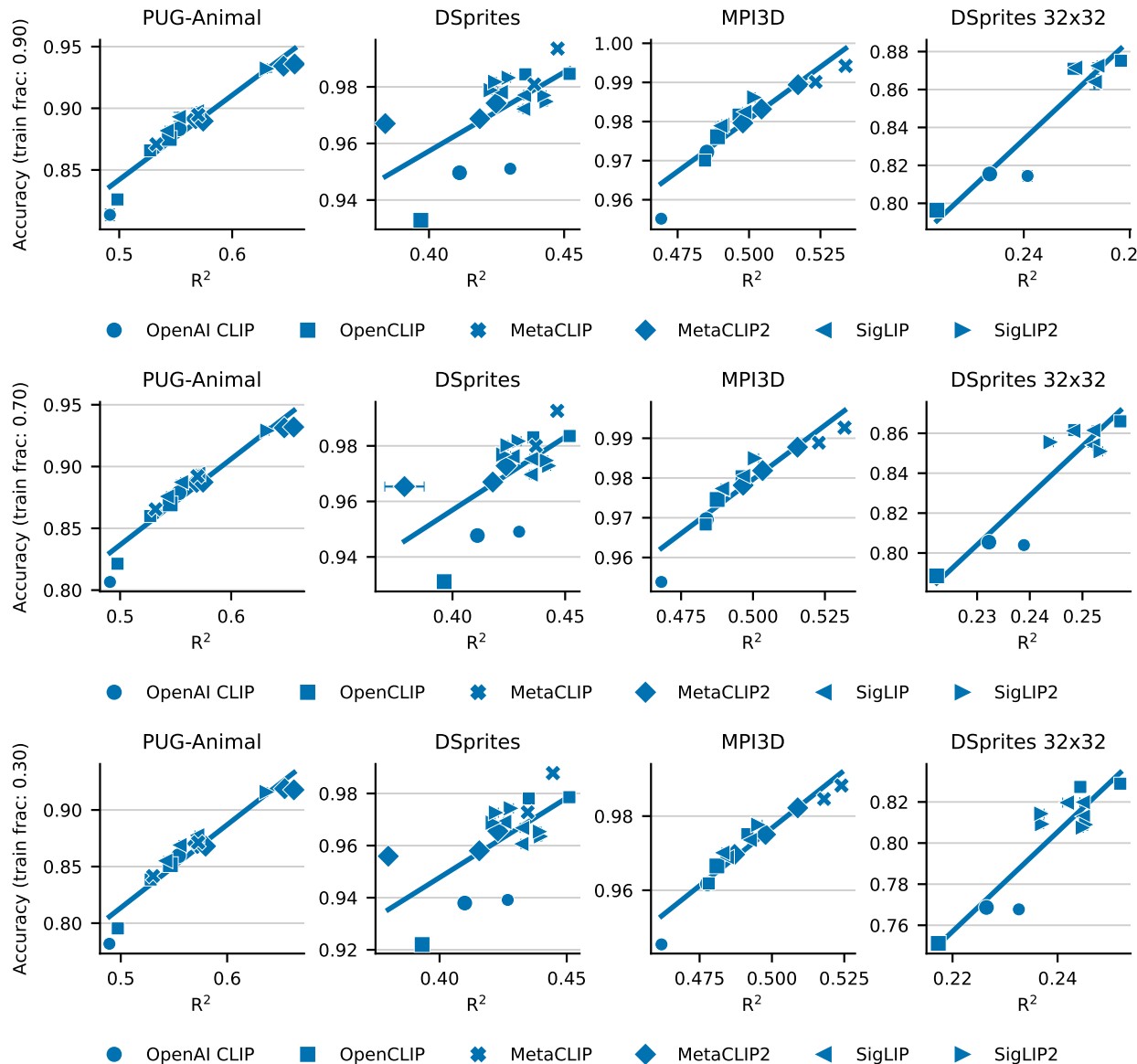

*Figure 35.* **Correlation between projected $R^2$ and compositional accuracy across train fractions (part 1/2).** Each row uses a different train fraction (shown on the $y$-axis). The positive correlation persists across all regimes. The rightmost column shows results on dSprites with the full $32\times32$ position grid.

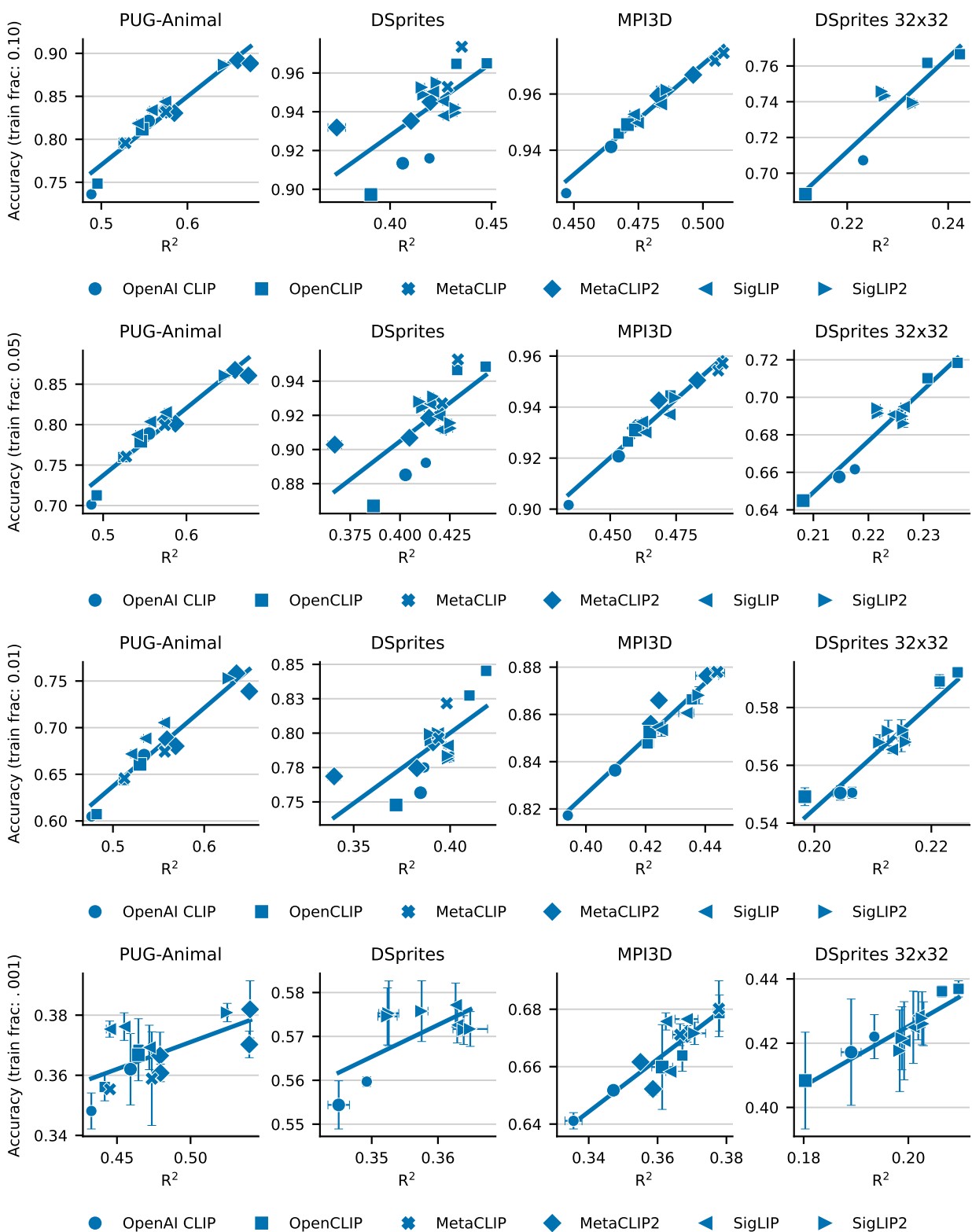

*Figure 36.* **Correlation between projected $R^2$ and compositional accuracy across train fractions (part 2/2).** Continuation of Fig. 35. The positive correlation persists throughout. The rightmost column shows results on dSprites with the full $32\times32$ position grid.

### H.5.2. PROJECTED $R^2$ AS A FUNCTION OF TRAIN FRACTION

Fig. 37 shows projected $R^2$ as a function of train fraction $|S|/N$ across models and datasets. The $R^2$ score increases with train fraction but plateaus, and the relative ranking across models is preserved across regimes. In particular, the metric is not trivially inflated at high $|S|$: the major jump in $R^2$ occurs only as $|S|$ approaches $N$, indicating that the values reported in the main text are not an artifact of the chosen split.

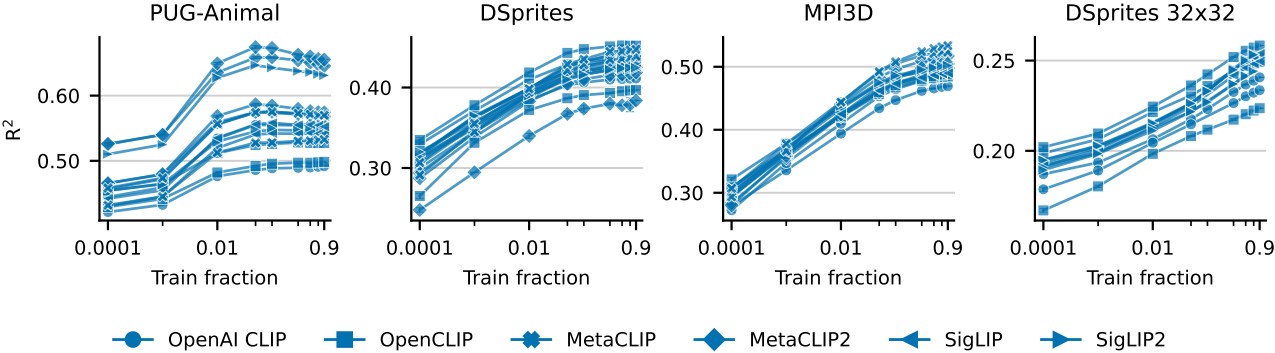

*Figure 37.* **Projected $R^2$ as a function of train fraction $|S|/N$.** $R^2$ increases with train fraction but plateaus; relative model ranking is preserved across regimes. The rightmost panel reports results on dSprites with the full $32 \times 32$ position grid.

### H.5.3. ORTHOGONALITY WITH ERROR BARS AND FULL DSPRITES GRID

Fig. 38 complements Fig. 10 with (i) standard deviations across concept pairs, providing uncertainty quantification for the within- vs. across-concept comparison, and (ii) a re-run on dSprites with the full $32 \times 32$ position grid (denoted dSprites32). The within-concept vs. cross-concept gap is consistent across all models and datasets, and the orthogonality pattern is preserved on the complete dSprites grid.

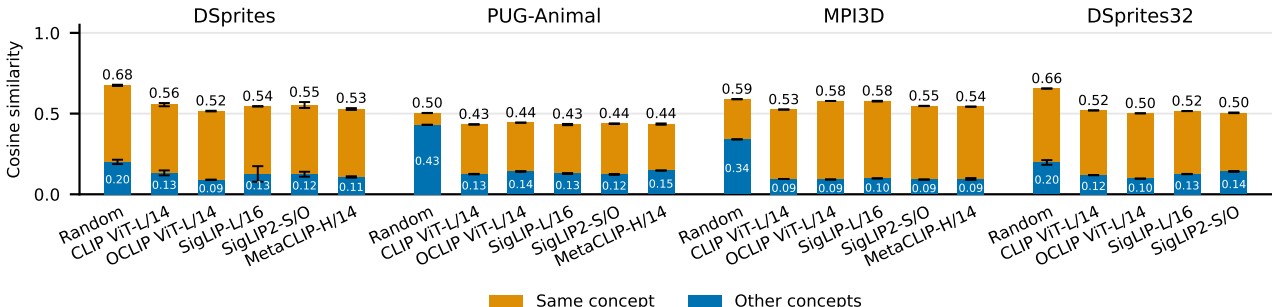

*Figure 38.* **Within-concept vs. across-concept cosine similarity with uncertainty.** Error bars are standard deviations across concept pairs. Pretrained models exhibit consistently higher within-concept similarity (orange) than across-concept similarity (blue), indicating partial orthogonality. The rightmost panel reports results on dSprites with the full $32 \times 32$ position grid, confirming that the orthogonality pattern holds on the complete dataset.

### H.5.4. SPECTRAL ANALYSIS OF RESIDUALS

To characterize the structure of the variance not explained by linear factorization, we compute the SVD of the residuals after subtracting the recovered factorization $\sum_i \boldsymbol{u}_{i,c_i}$ (in the projected and whitened space, consistent with how projected $R^2$ is computed). Fig. 39 shows that the residual variance is spread across many components with no dominant singular values, indicating that the unexplained variance is not concentrated in a low-dimensional structured subspace. This holds across both models (CLIP ViT-L/14 and SigLIP2-SO/400M) and all three datasets (dSprites, MPI3D, PUG-Animal).

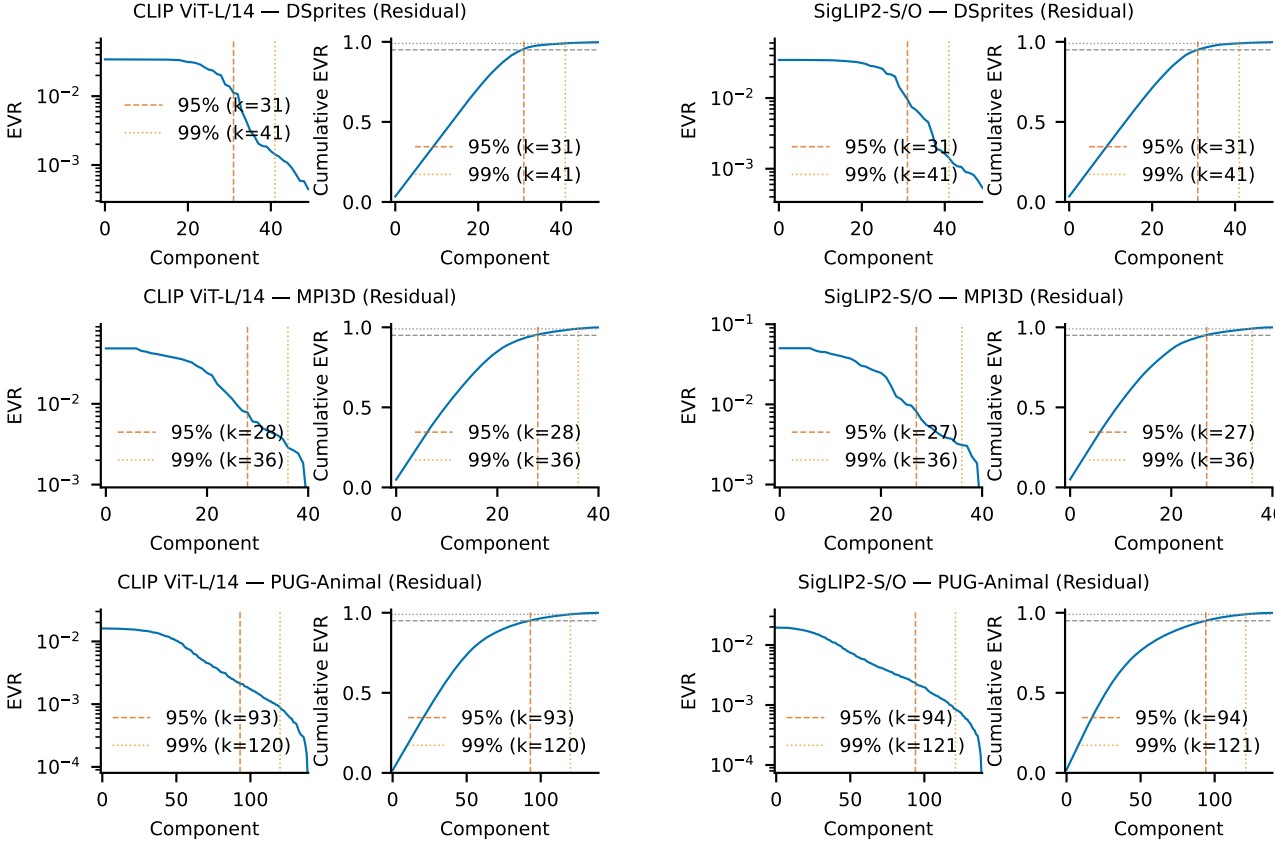

*Figure 39.* **Spectral analysis of residuals after subtracting the linear factorization** (projected + whitened space). Left column: CLIP ViT-L/14. Right column: SigLIP2-SO/400M. Rows: dSprites, MPI3D, PUG-Animal. The residual variance is spread across many components rather than concentrated in a few, indicating that the unexplained variance is not structured.

## H.6. Qualitative results

We visualize the recovered factor geometry across three models (DINOv3, OpenCLIP, MetaCLIP-G) and three datasets (dSprites, MPI3D, PUG-Animal). For dSprites (Fig. 41) and MPI3D (Fig. 42), we show both the per-concept factors and the pairwise projections (analogous to Fig. 7c,d in the main text), illustrating low-rank factor structure and near-orthogonality across concepts. For PUG-Animal (Fig. 40), we show only the per-concept factors projected onto their top 3 principal components, since most of its concepts (e.g., character, background) are discrete with many values, resulting in higher-rank factor sets where pairwise projections are less informative.

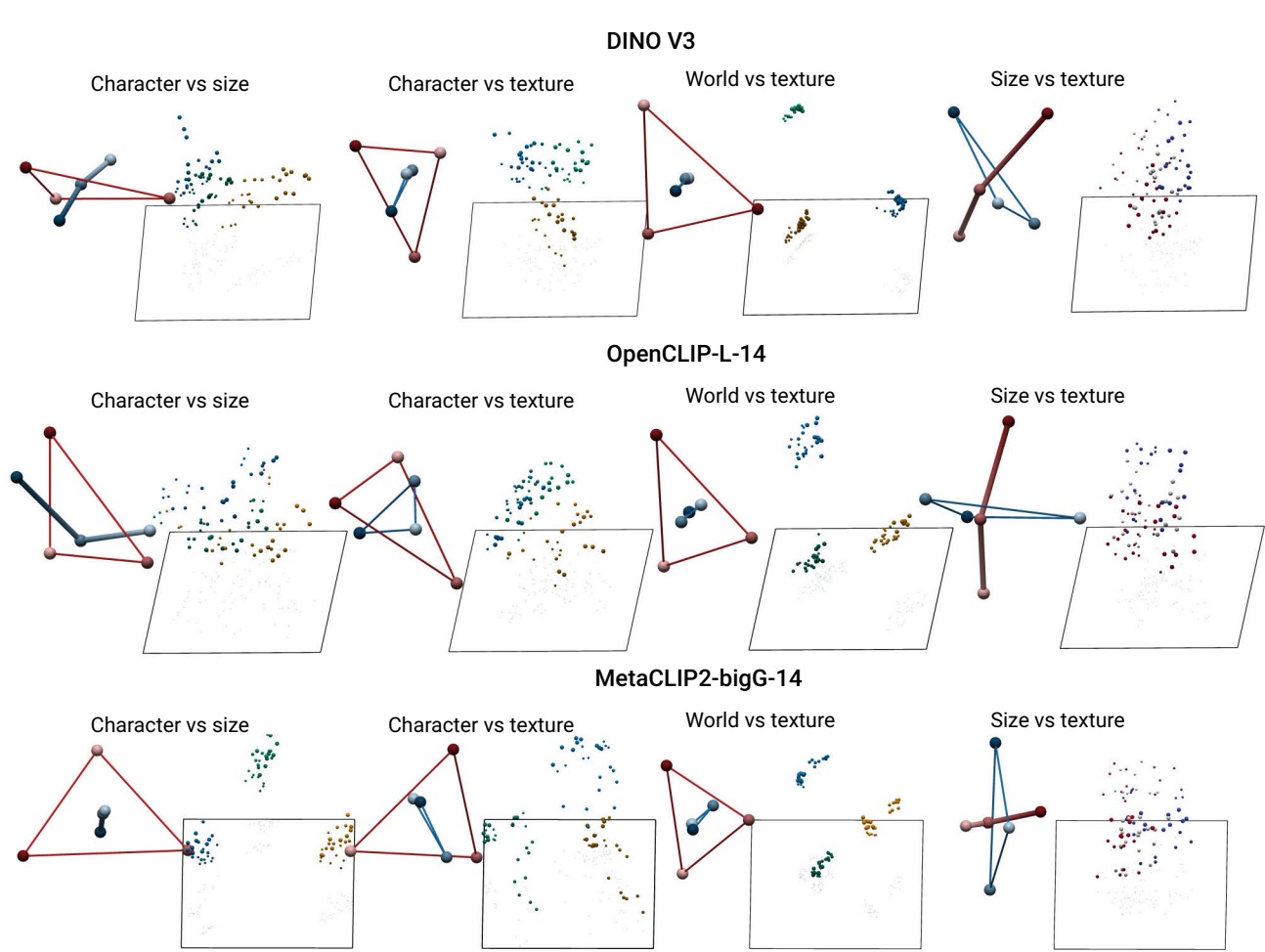

*Figure 40.* **Factor geometry on PUG-Animal across DINOv3, OpenCLIP, and MetaCLIP2-bigG.** Only per-concept factors (projected onto their top 3 PCs) are shown, since most PUG-Animal concepts (character, background) are discrete with many values, yielding higher-rank factor sets where pairwise projections are less informative.

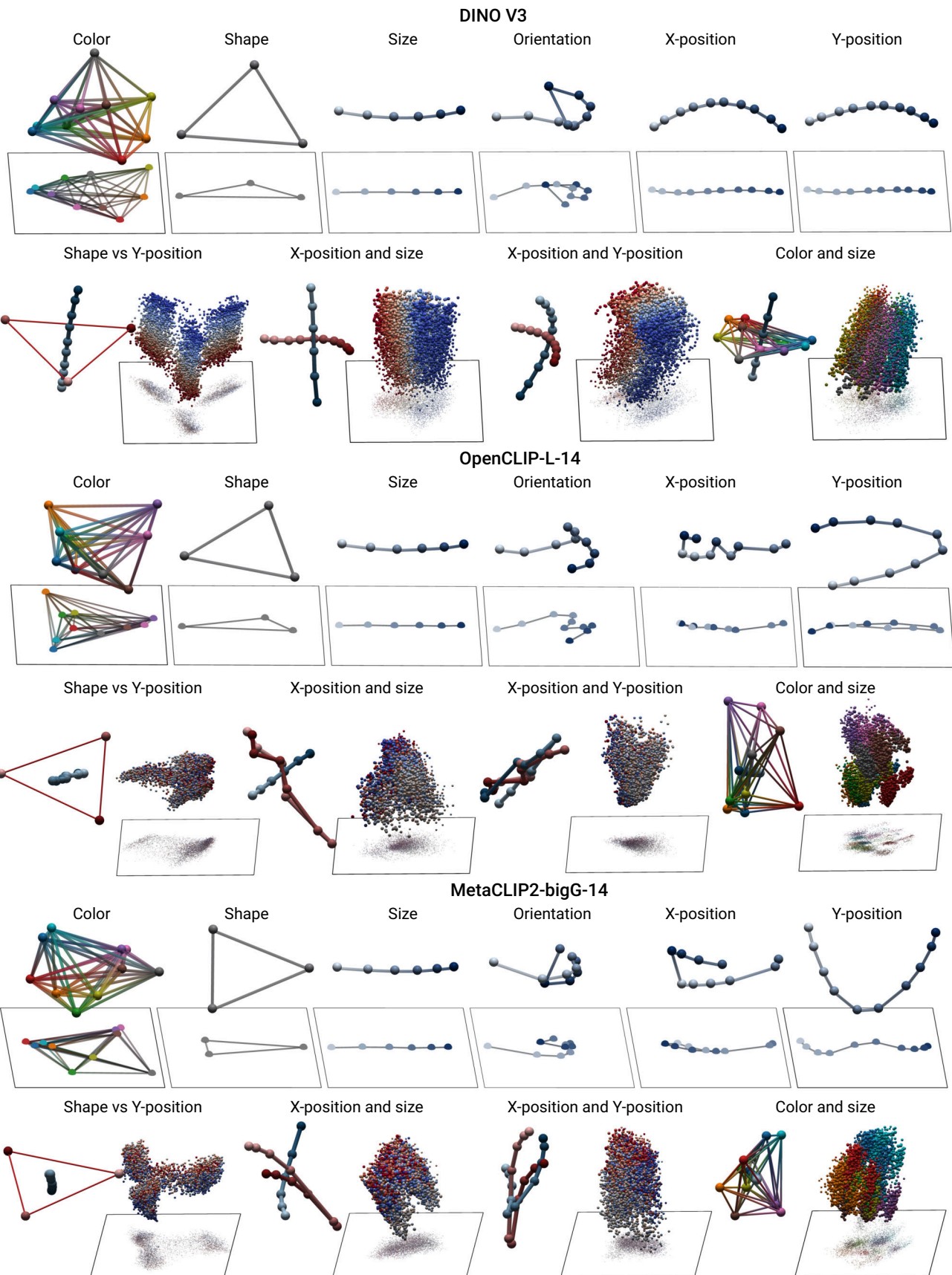

*Figure 41.* **Factor geometry on dSprites across DINOv3, OpenCLIP, and MetaCLIP2-bigG.** For each model: the first row shows the recovered per-concept factors (one 3D plot per concept); the second row shows the pairwise factor geometry for selected concept pairs; and the third row shows all datapoints projected onto the span of those concept pairs.

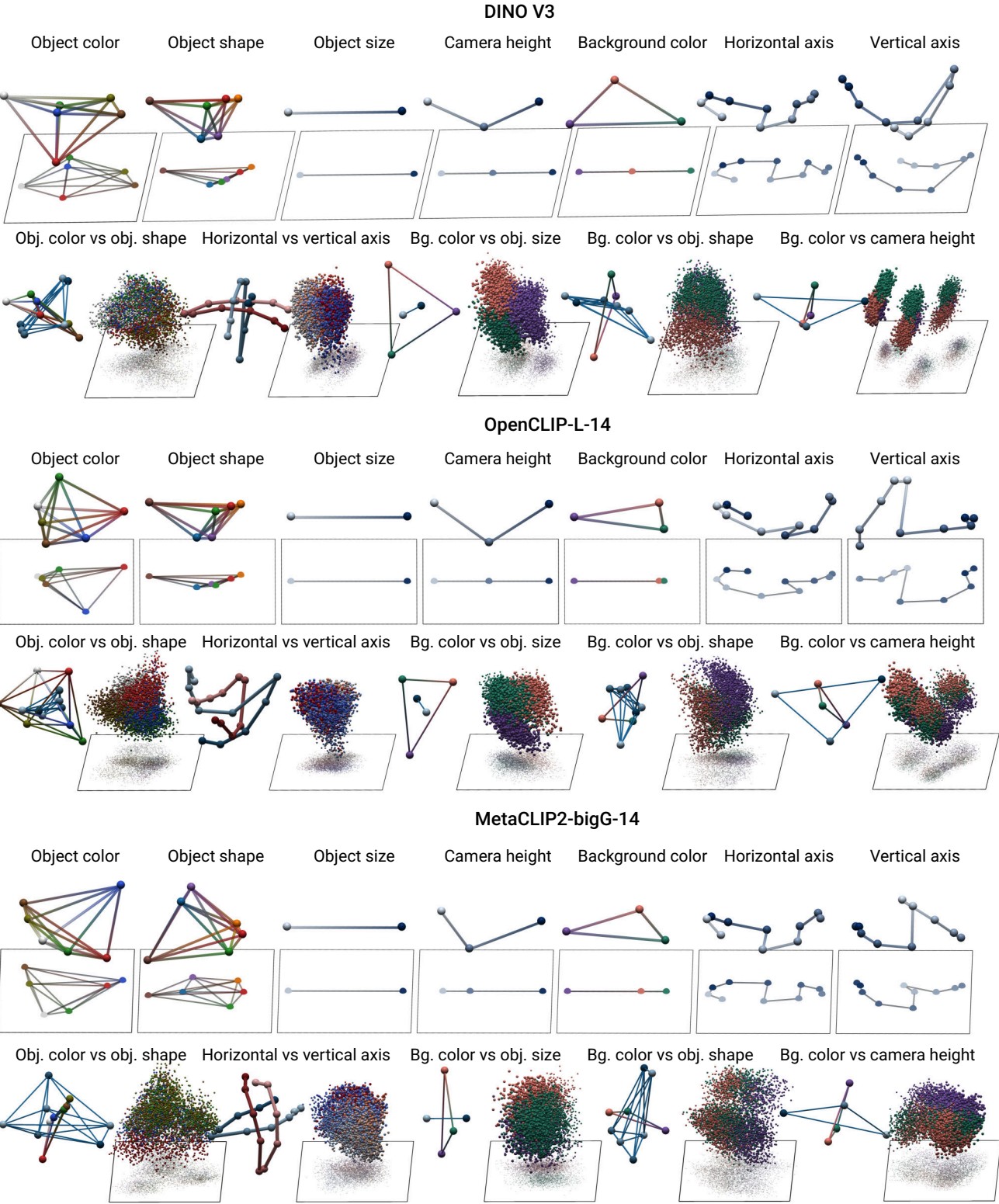

*Figure 42.* **Factor geometry on MPI3D across DINOv3, OpenCLIP, and MetaCLIP2-bigG.** Layout follows Fig. 41: for each model, per-concept factors (first row), pairwise factor geometry (second row), and pairwise datapoint projections (third row).

## H.7. Empirical evaluation on synthetic data

Results in Section 4 establish the geometry of compositionally generalizing models. A natural question is what geometry emerges in models trained with standard classification losses, without explicit pressure to generalize compositionally. In addition, the core theory is derived for binary concepts, whereas the experiments below also include multi-valued concepts. We study this empirically in this section.

Standard pretraining regimes (e.g., CLIP/SigLIP) do not explicitly optimize for Desideratum 2. This controlled setup therefore lets us isolate which geometric structure emerges by common objectives. The results in this subsection should be read as properties of compositional models trained with standard losses, and not as guarantees of compositional generalization.

We vary two representation geometries and two losses, independently, to cover common training settings and model families, such as CLIP and SigLIP models. We remain architecture-agnostic here, and we optimize embeddings corresponding to each concept combination directly (similarly to (Weller et al., 2025b)).

Because the number of combinations grows as $n^k$, we use at most 100,000 combinations per setting: if $n^k \leq 100,000$, we use all of the combinations; otherwise, we sample 100,000 combinations. As a result, reported $R^2$ and factor-orthogonality values in those regimes are approximate estimates and should be interpreted with caution. For each setting, we report three metrics, following Section 4: linear-factorization $R^2$, factor orthogonality, and dimension needed to support linearly compositional models (Definition 3).

We use concept spaces $\mathcal{C}_{k,n} := \mathcal{C}_1 \times \cdots \times \mathcal{C}_k \subset [n]^k$, with embeddings $\boldsymbol{z_c} \in \mathbb{R}^d$ for each $\boldsymbol{c} \in \mathcal{C}_{k,n}$ and scores $h_{i,j}(\boldsymbol{z}) := \tau \, \boldsymbol{w}_{i,j}^\mathsf{T} \boldsymbol{z} + b_{i,j}$. Each $\boldsymbol{c}$ is initialized randomly: $\boldsymbol{z_c} \sim \mathcal{N}(\boldsymbol{0}, \boldsymbol{I})$. Unless stated otherwise, optimization uses Adam for 50,000 epochs with initial learning rate 0.1 and cosine annealing ($T_{\max} = 50{,}000$, $\eta_{\min} = 0$). We fit the probes as well as the embeddings jointly.

We vary the following factors independently.

**(1) Representation geometry.** We consider (1) Euclidean ($\boldsymbol{z_c} \in \mathbb{R}^d$), where $\tau$ is absorbed into probe scale (so $\tau = 1$), and (2) spherical ($\boldsymbol{z_c} \in \mathcal{S}^{d-1}$ with $\|\boldsymbol{z_c}\| = \|\boldsymbol{w}_{i,j}\| = 1$) geometries, where CLIP/SigLIP-style normalization uses explicit temperature $\tau$.

**(2) Loss type.** We compare two per-concept losses (one object per scene). First, per-concept softmax cross-entropy (CE):

$$\ell_{\mathrm{CE}}(\boldsymbol{z_c}) = \sum_{i=1}^{k} \left( -\log \frac{\exp h_{i,c_i}(\boldsymbol{z_c})}{\sum_{v=1}^{n} \exp h_{i,v}(\boldsymbol{z_c})} \right). \tag{99}$$

Second, per-concept one-vs-rest binary cross-entropy with sigmoid outputs (BCE):

$$\ell_{\mathrm{BCE}}(\boldsymbol{z_c}) = \sum_{i=1}^{k} \frac{1}{n} \left[ -\log \sigma(h_{i,c_i}(\boldsymbol{z_c})) - \sum_{v \neq c_i} \log \sigma(-h_{i,v}(\boldsymbol{z_c})) \right]. \tag{100}$$

**(3) Concept space and dimension.** For each geometry/loss pair, we retrain with $k \in [10]$, $n \in \{2, 6, 12, 24, 48, 96\}$, and $d \in \{3, \ldots, 32\}$.

### H.7.1. RESULTS: LINEARITY.

We illustrate the linearity of the embeddings in the from-scratch setting in Fig. 43. Notably, the majority of the cases exhibit $R^2 \geq 0.7$.

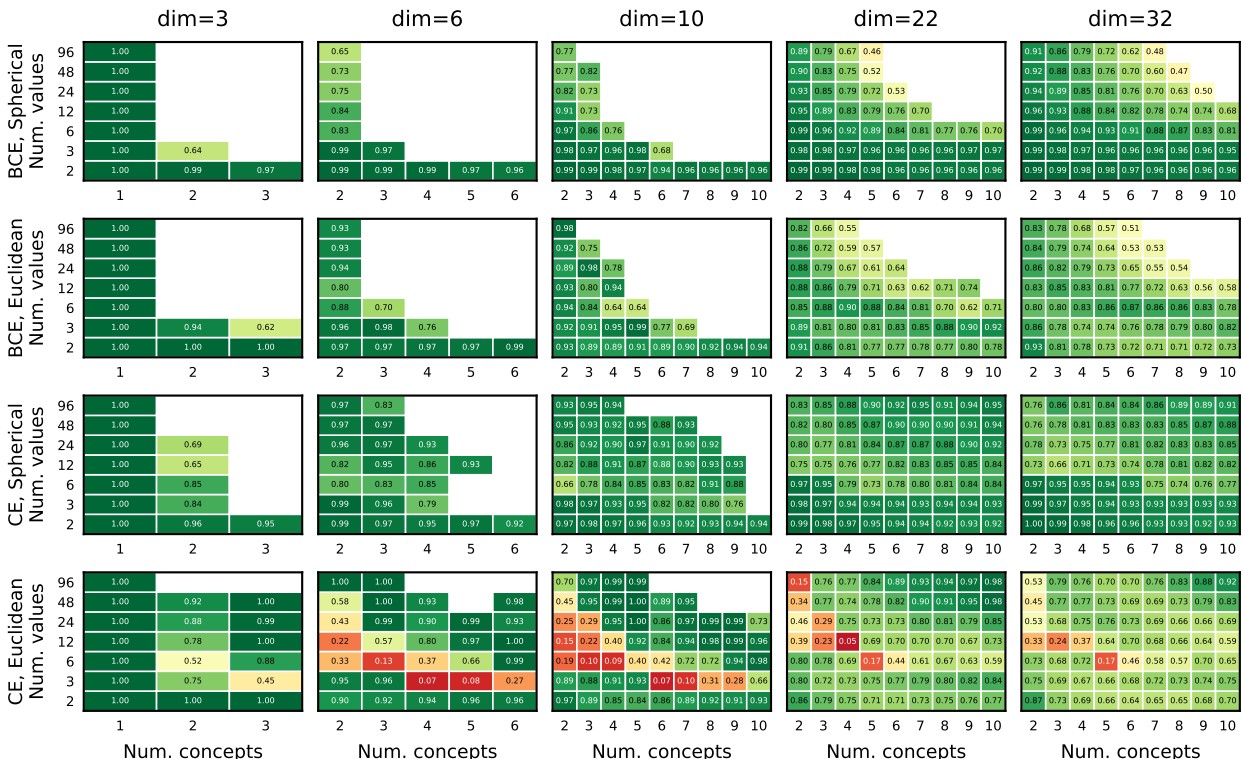

*Figure 43.* **Linear factorization $R^2$ results in from-scratch setting.** We show the linear-factorization $R^2$ when representation space varies and loss type varies. In the majority of the cases, $R^2 \geq 0.7$.

### H.7.2. RESULTS: ORTHOGONALITY.

The theory in Section 4.1 predicts orthogonality of *cross-concept* factor differences, while not requiring within-concept directions to be orthogonal. We measure this exactly as in Appendix H.2: factors are recovered by averaging (centered) embeddings per concept value, then we compute absolute-cosine summaries for within-concept similarity ($\mathrm{Orth}(i, i)$) and cross-concept orthogonality ($\mathrm{Orth}(i, j), i \neq j$). In the from-scratch experiments, we observe this pattern consistently across CE/BCE losses and Euclidean/spherical geometries: cross-concept cosine similarity is lower than within-concept cosine similarity, and it decreases as $k$ grows. This behavior is therefore aligned with the geometric story in the main theory, though still approximate.

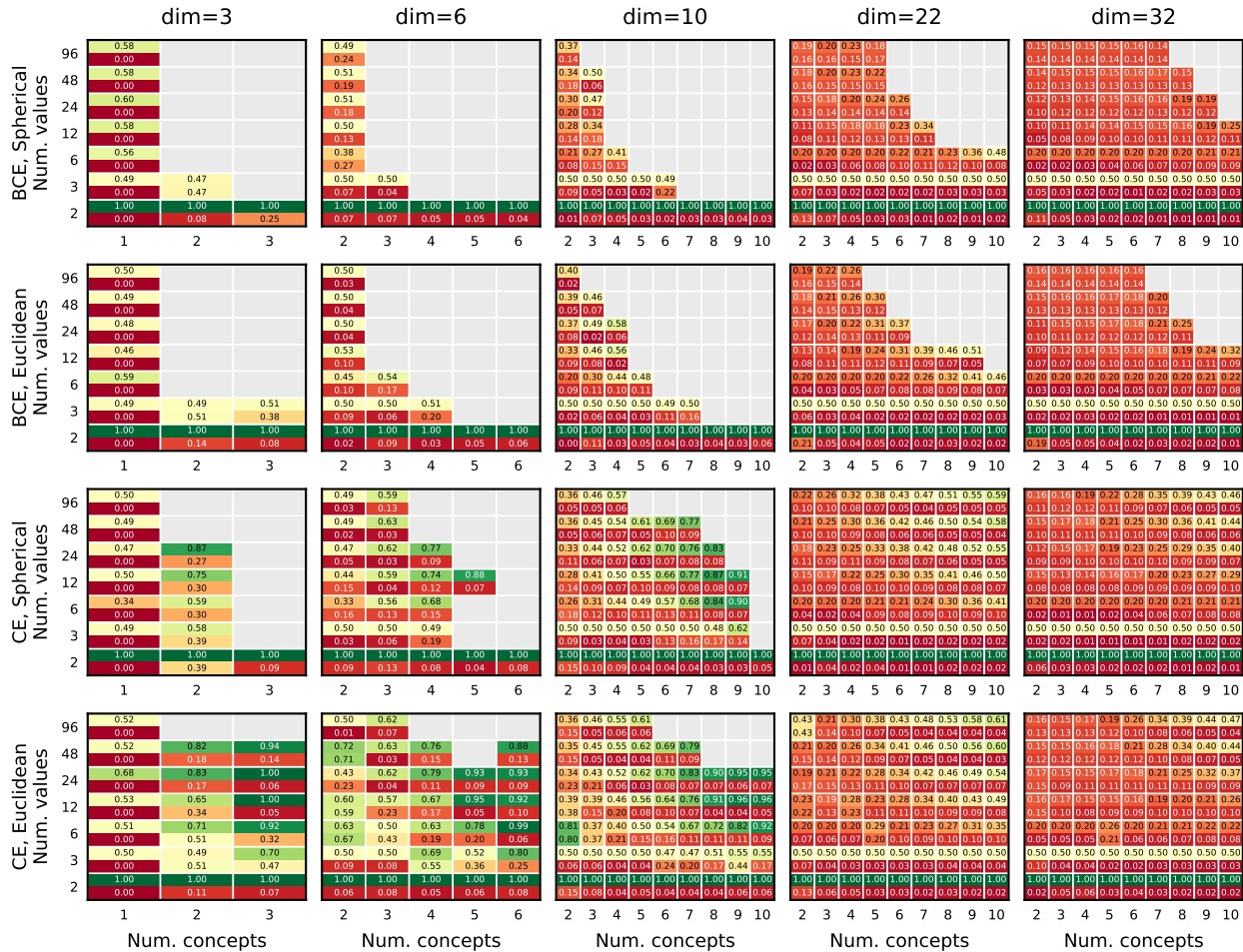

*Figure 44.* **Within-concept and cross-concept cosine similarity of factors in the from-scratch setting.** Metrics are computed as in Appendix H.2. For each $(k, n)$ setting, each cell reports two values: within-concept cosine similarity (top) and cross-concept cosine similarity (bottom). As the number of concepts increases, cross-concept cosine similarity decreases, indicating stronger cross-concept orthogonality.

### H.7.3. RESULTS: DIMENSIONALITY.

In Section 4.2, we proved the lower bound $d \geq k$ for linear readout, independent of the number of values $n$. That result is a capacity bound: it states what is necessary in principle, but does not guarantee that a particular training objective reaches the bound.

Here, we estimate the minimum dimension needed to reach $\geq 0.99$ per-concept accuracy across concept spaces varying in $k$ and $n$, under CE (CLIP-like training) and BCE (SigLIP-like) losses, and under Euclidean (dot-product) and spherical (cosine-normalized) geometries.

The trends in Fig. 45 are consistent with the theory. CE (CLIP-like) in Euclidean space is typically close to the theoretical minimum ($d \approx k$), whereas BCE (SigLIP-like) generally requires larger dimension (often around $2k$), perhaps intuitively: in BCE each class is trained as an independent one-vs-rest binary problem, so each positive set must be separated from all negatives by its own hyperplane; CE instead uses relative margins across classes and can thus separate more flexibly (Crammer & Singer, 2002; Bangachev et al., 2025), matching the inference rule with the optimization objective. Spherical geometry shifts required dimensionality upward by roughly one dimension relative to Euclidean settings. Overall, this supports the theoretical claim that concept count is the primary dimensional driver, while showing that objective affect the attainability of the bound.

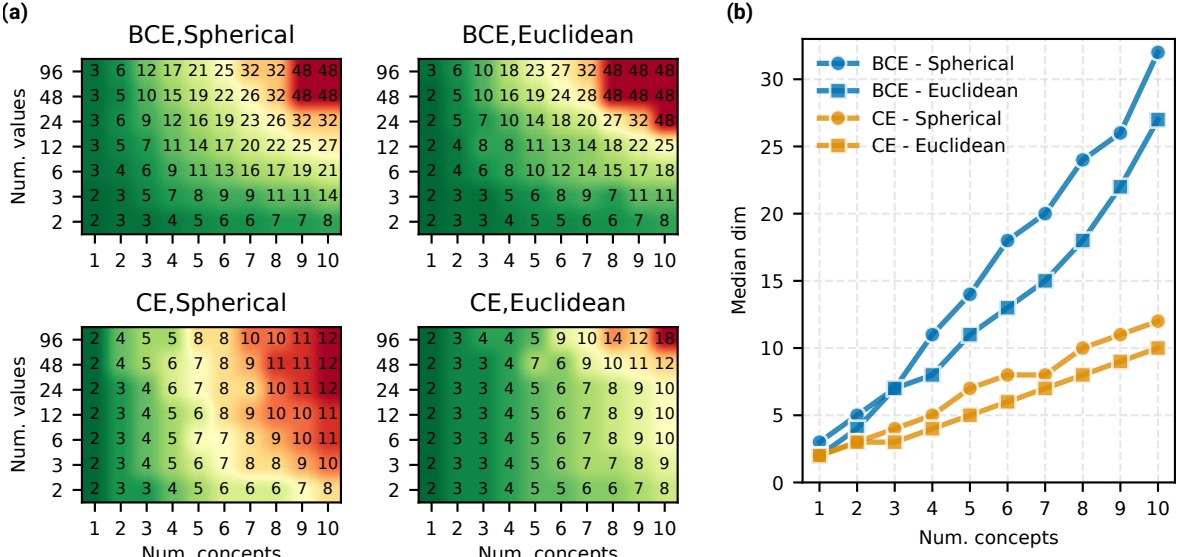

*Figure 45.* **Minimum embedding dimensionality across concept spaces, losses, and representation geometries.** We report the smallest dimension $d$ needed to reach $\geq 0.99$ per-concept classification accuracy for concept spaces with $k$ concepts and $n$ values per concept. **(a)** Required $d$ as both $k$ and $n$ vary. **(b)** Median required $d$ versus $k$: CE is typically near $d \approx k$, BCE needs roughly $2k$, and spherical variants require about one additional dimension compared to Euclidean.

