# OpenReview forum: "Compositional Generalization Requires Linear, Orthogonal Representations in Vision Embedding Models"
_ICML.cc/2026/Conference — ICML 2026 spotlight_

### Official Review · Reviewer_iHWt · 2026-03-03

**Soundness:** 3
**Presentation:** 4
**Significance:** 2
**Originality:** 2
**Overall Recommendation:** 5
**Confidence:** 3

**Summary:**

This paper investigates the structural properties required for embedding models to achieve compositional generalization. By formalizing three desiderata (divisibility, transferability, and stability), the authors prove that standard gradient descent with cross-entropy loss forces representations into an orthogonal, linearly factorized geometry. They support this with an empirical survey of CLIP and SigLIP models on compositional datasets, analyzing linear factorizability, orthogonality, and the dimension of factors.

**Compliance With Llm Reviewing Policy:**

Affirmed.

**Final Justification:**

The rebuttal clarified the points I had raised in my review, leading me to update my recommendation favourably.

**Key Questions For Authors:**

- How should we interpret the requirement of $|T| = 2^{k-1} + 1$ in Proposition 1, given the claim that foundation models learn from a small subset of the concept space?

- Can you clarify the statement that "the conclusions of Proposition 1 hold for $1 + c \le |T| \le 1 + 2^{c-1}$"?

- To evaluate the correlation between $R^2$ and compositional accuracy, the linear probes are trained on 90% of all concept combinations and tested on the remaining 10%. How does regime reflect the compositional generalization that foundation models face in practice?

- Given that the empirical $R^2$ scores for linear factorization hover between 0.40 and 0.60, nearly half of the variance remains unexplained by the linear components. How can we interpret linear factorization as a necessary condition, in light of this?

**Limitations:**

The authors have not discussed limitations. I would encourage them to include them in the main paper, addressing, for instance, whether the desiderata hold in practice, the $|T| = 2^{k-1} + 1$ requirement in Proposition 1, the fact that stability is crucial to ensure linear factorization, the empirical evidence showing only partial linear factorization, etc. (see my comments).

**Strengths And Weaknesses:**

**Presentation**

- The paper is very well written. The transition from the empirical problem of VLMs failing at compositional generalization to a foundational geometric question is compelling.

- Typo in line 341 ("datasaets").

- Typo in line 414 ("liear").

**Soundness**

- Within the strict requirements of the provided idealized assumptions (perfect zero-shot transfer, rigid posterior probabilities), the results, namely, the geometric proofs for linear factorization hold up. However, these seem to be an ideal case that seldom holds in practice. Plus, as shown in the appendix, if stability is relaxed, we lose the linear factorization altogether.

- The proof for linear factorization explicitly requires a training set size of $|T| = 2^{k-1} + 1$ (Proposition 1). This means the model must see strictly more than 50% of the entire combinatorial universe to guarantee the geometry. This directly contradicts the motivating premise that models learn from a "tiny, biased subset".

- Immediately after Proposition 1, the text claims the conclusions hold for a minimal-learning regime. This seems incompatible with the "random sampling" ($|T| = 2^{k-1} + 1$) assumption.

**Significance**

- Connecting the empirical phenomenon of the linear representation hypothesis to the max-margin bias of gradient descent provides a valuable contribution for the representation learning community.

- The experiments that aim to establish a relation between compositional generalization and linear factorization relied on probes trained on 90% of all concept combinations. This is a large number of training examples given that the goal is to test generalization.

- Evaluating foundation models trained on massive web data collections (CLIP, SigLIP) on clean, synthetic toy datasets (dSprites, MPI3D) introduces a severe domain mismatch for the benchmarks.

- The title explicitly claims "Necessary conditions". However, the empirical results show $R^2$ scores between $0.4$ and $0.6$, leaving nearly half the variance unexplained. If a model can achieve compositional generalization while missing 50% of the "necessary" linear structure, can we call the condition necessary?

- While there is a trend between $R^2$ and compositional accuracy, the variance is surprisingly narrow. For example, in dSprites, $R^2$ varies only in 0.40-0.46, while accuracy is heavily saturated between 0.93-0.98. This makes it difficult to definitively claim a strong causal link.

**Originality**

- The overall theoretical contribution and empirical results are insightful, despite the weakness mentioned above.

- However, I do not think the dimensionality result from Proposition 2 is particularly novel. Extracting $k$ independent linear features without interference requires at least $k$ orthogonal dimensions (we can represent $n$ different colors along an axis but to ensure disentanglement wrt to shape, the different shape values would need to be placed in an axis orthogonal to the color one).

---

> ### Author Rebuttal · Authors · 2026-03-30
>
> Thank you for the thoughtful review. We provide supplementary figures for the new experiments in the [rebuttal PDF](https://anonymous.4open.science/api/repo/anonrebuttal5314413-89BD/file/rebuttal.pdf?v=40554247).
>
> **1. How should we interpret the requirement of $|S|>N/2$ in Proposition 1, given the claim that foundation models learn from a small subset of the concept space?**
>
> Thank you for pointing it out. Our intention was to highlight that the proposition applies in two scenarios: when $|S|$ is close to $|T|$ (minimal-learning regime, $|T|=1+k$) and when $|S|$ is close to $|N|$ (large-sample regime, $|S|>N/2$). As such, the result should be understood as holding in both low-data and high-data regimes and applies universally. We will clarify this in the text.
>
> **2. Can you clarify the statement that 'the conclusions of Proposition 1 hold for $|T|=1+k$'?**
>
> Thank you for pointing this out. Proposition 1 covers two separate cases with different validity rules: (1) random sampling, where $|T| = 2^{k-1} + 1$ (just over half the grid), and (2) a minimal cross-dataset construction, where $|T| = 1 + k$ chosen points that each differ in exactly one concept. The $|T| = 1+k$ case does not use random sampling, it uses a structured validity rule. We agree the current text does not make this distinction clear enough and will explicitly separate the two cases and their respective validity rules in the revision.
>
> **3. To evaluate the correlation between $R^2$ and compositional accuracy, the linear probes are trained on 90% of all concept combinations and tested on the remaining 10%. How does regime reflect the compositional generalization that foundation models face in practice?**
>
> We agree that the 90/10 split is generous. We ran additional experiments across a wide range of train fractions (from 0.001 to 0.9) over 3 seeds and find that the positive correlation between $R^2$ and compositional accuracy persists across all regimes, including extreme low-data settings. See **Fig. 1–2 in the rebuttal PDF**.
>
> **4. The empirical $R^2$ scores hover between 0.40 and 0.60, nearly half of the variance remains unexplained. How can we interpret linear factorization as a necessary condition, in light of this?**
>
> The theory says if a model perfectly generalizes compositionally, its representations must be linearly factorized ($R^2 = 1.0$). Current models only partially generalize; their $R^2$ is also partial ($0.4$–$0.6$). The necessary condition is thus not violated, it is partially met, and the models partially succeed. Our correlation plots (**Fig. 1–2 in the rebuttal PDF**) show that models with higher $R^2$ also achieve better compositional accuracy, which is what we'd expect if linear factorization is necessary. The gap from $R^2 = 1.0$ may explain why current models still fail on compositional benchmarks.
>
>
> **5. Evaluating foundation models trained on massive web data collections (CLIP, SigLIP) on clean, synthetic toy datasets (dSprites, MPI3D) introduces a severe domain mismatch**
>
> We agree that the visual domain of dSprites and MPI3D differs from natural images. However, foundation models are expected to perform well on a wide range of tasks, including the ones we consider. Importantly, the tasks we use have natural corresponding language descriptions, variations of which are seen during pre-training. oreover, our results are consistent across both synthetic datasets and the photorealistic PUG-Animal, suggesting that domain mismatch does not change the conclusions.
>
> **6. I do not think the dimensionality result from Proposition 2 is particularly novel. Extracting $k$ independent linear features without interference requires at least $k$ orthogonal dimensions**
>
> We agree that given orthogonality, $d \geq k$ is intuitive. But Proposition 2 does not assume orthogonality. It shows $d \geq k$ follows from divisibility alone (correct linear probes for all $n^k$ combinations). So it is a stronger statement than the one requiring orthogonality. We will be sure to highlight this in the revision.
>
> **7. If stability is relaxed, we lose the linear factorization altogether**
>
> We agree that stability is a strong condition, but it is practically motivated: a model whose predictions change depending on the training subset is undesirable. We belive any practitioner would want this property. The fact that relaxing stability breaks linear factorization informs future work on approximately stable models: what happens under small distributional deviations across supports, and what geometric guarantees can still be maintained. However, our goal was understanding models that behave in an ideal way.
>
> **8. The authors have not discussed limitations.**
>
> Thank you for pointing this out. We will add an explicit limitations section in the revision.
>
> **9. Typos**
>
> Thank you for pointing these out!

---

> > ### Author Rebuttal · Reviewer_iHWt · 2026-04-03
> >
> > The rebuttal addressed the points I made. I will thus be raising my score.

---

### Official Review · Reviewer_KLRh · 2026-03-10

**Soundness:** 3
**Presentation:** 3
**Significance:** 3
**Originality:** 3
**Overall Recommendation:** 5
**Confidence:** 3

**Summary:**

The authors propose a formal theoretical framework outlining the necessary conditions for embedding models to achieve compositional generalization. They define three core desiderata, i.e., divisibility, transferability, and stability, and theoretically prove that models trained with linear heads using gradient descent and cross-entropy loss must exhibit linear factorization with near-orthogonal concept difference directions to satisfy these criteria. Furthermore, the authors establish a mathematical lower bound on the embedding dimension, demonstrating it must be at least the number of concepts ($d \ge k $) for a linear readout to succeed across all combinations. Empirically, the authors evaluate families of CLIP and SigLIP models on datasets including PUG-Animal, dSprites, and MPI3D. Their study shows that the degree of linear factorization (measured via projected $R^2$) positively correlates with compositional generalization performance, although current pre-trained models only partially achieve this ideal geometry

**Compliance With Llm Reviewing Policy:**

Affirmed.

**Final Justification:**

I am raising my final score to an Accept because the authors' rebuttal addressed my main concerns regarding statistical rigor and the unexplained variance gap. By providing error bars for the orthogonality metrics and conducting an SVD analysis on the residuals, the authors significantly improved the soundness and clarity of their empirical evaluations. In view of this and the empirical improvements against the paper's highly original and significant theoretical contributions, I find this to be a technically strong submission that merits acceptance.

**Key Questions For Authors:**

1. How sensitive is the observed linear factorization ($R^2$) to the number of concepts $k$ as it scales to highly complex, unconstrained real-world scenes?

2. There are no significance testing or error bars on the experimental results. Could you provide confidence intervals for the projected $R^2$ and orthogonality measurements in your empirical evaluations?

3. The empirical models show only partial linear factorization ($R^2 \approx 0.4 - 0.6$). By reporting metrics such as singular value gap and effective condition number, could you better characterize the structural geometry of the space that remains unexplained by the linear factors?

4. How does the theoretical framework perform when the concept labels or the data collection process are inherently noisy or non-isomorphic? Would the learned representations still predictably factorize?

**Limitations:**

No uncertainty/confidence/error bars on experimental results or significance testing.

**Strengths And Weaknesses:**

Strengths of the Paper:

1. The authors provide a clear description of the background knowledge and motivations needed to understand the proposed theoretical framework for compositional generalization.

2. The exposition of the theoretical claims is clear, establishing a rigorous link between practical training desiderata and the widely observed Linear Representation Hypothesis.

3. The theoretical bound demonstrating that the minimal dimensionality depends solely on the number of concepts ($k$) rather than the number of concept values ($n$) is an original and highly insightful contribution.

4. The authors performed a comprehensive empirical evaluation across multiple model architectures (CLIP, SigLIP) and datasets, demonstrating a strong correlation between their theoretical linear factorization metric ($R^2$) and actual compositional transfer.

Weaknesses of the Paper:

1. No uncertainty/confidence/error bars on experimental results, or significance testing for the $R^2$ and orthogonality metrics.

2. The paper could benefit from a deeper structural analysis of the embeddings. While the authors measure projected $R^2$ to assess linearity, the unexplained variance (the gap from $R^2=1.0$ to the observed $0.4-0.6$) is not fully characterized.

3. There is limited discussion on the computational complexity and scalability of the proposed framework in highly heterogeneous, noisy, or massive-scale concept spaces where concepts might not be perfectly orthogonal or well-defined.

---

> ### Author Rebuttal · Authors · 2026-03-30
>
> Thank you for the thoughtful review. We provide supplementary figures for the new experiments in the [rebuttal PDF](https://anonymous.4open.science/api/repo/anonrebuttal5314413-89BD/file/rebuttal.pdf?v=40554247).
>
> **1. No uncertainty/confidence/error bars on experimental results, or significance testing for the $R^2$ and orthogonality metrics.**
>
> Thanks for raising this. We recomputed the orthogonality metrics and show standard deviations across concept pairs (see **Fig. 4 in the rebuttal PDF**). The within-concept vs cross-concept gap is consistent across all models and datasets. We also reran the $R^2$ vs compositional accuracy experiments over 3 random seeds across all train fractions (**Fig. 1–3 in the rebuttal PDF**); the positive correlation is prevalent across seeds. We will add error bars to all remaining plots in the revision.
>
> **2. The unexplained variance (the gap from 1.0 to the observed $R^2$) is not fully characterized. Could you better characterize the structural geometry of the space that remains unexplained by the linear factors?**
>
> Thank you for the suggestion. We computed the SVD of the residuals after subtracting the linear factorization $\sum_i \mathbf{u}_{i,c_i}$ (in the projected + whitened space, consistent with how we compute $R^2$). See **Fig. 5 in the rebuttal PDF**. The residual variance appears to be spread across many components with no dominant singular values, suggesting the unexplained variance is not necessarily structured. This is consistent across models (CLIP ViT-L/14, SigLIP2) and datasets (dSprites, MPI3D, PUG-Animal).
>
> **3. How sensitive is the observed linear factorization ($R^2$) to the number of concepts $k$ as it scales to highly complex, unconstrained real-world scenes?**
>
> We think the key insight is that not all concepts apply to all images at the same time. Different images involve different subsets of concepts, and these subsets may not fully overlap. So the relevant $k$ is not the total number of concepts in the world, but the number of concepts active in a given context. This means the effective $k$ that needs to be packed into the representation at any one time is bounded and likely much smaller than it may seem. As an example, consider images of shapes versus images of animals: the relevant concepts for shapes (color, size, orientation) are different from those for animals (species, pose). A model does not need to represent all of these simultaneously for any single image.
>
> Our framework models exactly the concepts that are used to describe images in a given domain. For unconstrained scenes where the relevant concepts are not well-defined, compositional generalization itself is not well-defined either. Within a domain (e.g. dSprites or PUG-Animal), we study the concepts that naturally describe that domain, and the number of such concepts is moderate. We believe this is the right level of analysis: scaling to "all concepts everywhere" in practice would probably still be mapped to some meaningful (albeit probably large) set of concepts. We will include this discussion in the revision.
>
> **4. How does the theoretical framework perform when the concept labels or the data collection process are inherently noisy or non-isomorphic?**
>
> Our theory does not consider noisy labels. Our goal was to first establish what the ideal geometry looks like under clean data. Extending this to noisy settings (e.g. by modeling label error rates) is a natural next step. That said, the models we evaluate empirically (CLIP, SigLIP) are trained on noisy web data (e.g. LAION-2B), so our empirical findings already reflect realistic noise conditions.
>
> **5. There is limited discussion on the computational complexity and scalability of the proposed framework in highly heterogeneous, noisy, or massive-scale concept spaces.**
>
> We agree, we will include the part about noisy data observations in the limitations.

---

> > ### Author Rebuttal · Reviewer_KLRh · 2026-04-03
> >
> > Thank you, authors, for the responses, the thorough rebuttal, and the additional experiments. All my concerns have been resolved. I am happy to raise my final score to an Accept.

---

### Official Review · Reviewer_miQG · 2026-03-12

**Soundness:** 3
**Presentation:** 4
**Significance:** 4
**Originality:** 4
**Overall Recommendation:** 5
**Confidence:** 4

**Summary:**

This paper studies compositional generalization in embedding models from a theoretical lens, proposes three conditions for compositional generalization, and conducts extensive experiments to validate the conditions. The basic idea is that there is a product concept space from which training data are sampled and generated. An embedding model produces an embedding for each concept tuple, and then a readout yields the final classification. Compositional generalization is defined for an embedding model when the training data are only supported on a subset of the product concept space.
Based on this general setup, the authors propose three conditions / desiderata for compositional generalization. The main theoretical result shows that under some settings, compositional generalization implies linear factorization and orthogonal representation of concepts in the embedding space.
Then the authors conduct extensive experiments with pretrained models to show that  linear factorization and orthogonality approximately hold (though not perfectly) in real models, and they are correlated with compositional generalization.

**Compliance With Llm Reviewing Policy:**

Affirmed.

**Key Questions For Authors:**

1. The current theory is limited in the sense that the number of base concepts is no larger than the embedding dimension, since otherwise we don't have strict orthogonality. It would be interesting to allow near-orthogonality so that the embedding space can accommodate exponentially many base concepts. This would be valuable in the LLM literature for example, since empirical studies have strongly supported the near-orthogonality structure.

2. The authors haven't explained why the current embedding models succeed or fail in terms of the pretraining datasets. How does the selection of dataset impact the geometry and generalization? Is there spurious correlation in the base concepts or imbalances in frequency of concepts? It would be great to expand on the *distributional aspect* of training data.

**Limitations:**

I feel that the authors could have written this part better. Compositional generalization is a critical capability for AI models, so it's valuable to state which finding is likely to help practice and which finding is not thoroughly validated.

**Strengths And Weaknesses:**

Overall I think this paper provides a fresh and intriguing angle at a fundamental scientific question in deep learning. While there are several flaws, the originality of this paper clearly stands out (especially in the AI era), so I'd recommend "accept".

**Strengths**
- Most of the theoretical ideas look reasonable to me. While I don't have time to check all details, the general theoretical strategy makes sense to me. I also think that the authors have crafted experiments and evaluations rigorously.
- The presentation is great considering the technicality of the theoretical constructs. The figures, illustrations, theorem statements all look clear to me.
- The paper addresses a fundamental question, which is compositional generalization. What's interesting in the paper is that the theory builds the connection between generalization behavior with the model's embedding geometry (linear factorization, orthogonality). This might be of great value to the interpretability literature and shed insights to future architecture design.
- I also find the angle of this paper very novel. The theoretical constructs do seem to capture what we mean by compositionality.

**Weaknesses**
- The conditions are difficult to evaluate, and I find the Stability condition quite strong---we ask the embedding models to satisfy an invariance equality for all valid *training datasets*. I'd interpret the conditions more as conceptual guidance than as verifiable assumptions.
- It seems that the readout head is mostly restricted to linear readout. This means that we are look at the last-layer embeddings in a deep neural network. This significantly reduces the potential impact, but perhaps it can be relaxed in future work.
- The rigor of Proposition 1 could be improved. It feels sloppy to say the algorithm is "GD+CE". If the authors want to use the connection to max-margin classifier, they need to be clear about the conditions, like whether the model is overparametrized, the dimension vs. sample size, non-degeneracy, etc.

---

> ### Author Rebuttal · Authors · 2026-03-30
>
> Thank you for the thoughtful review.
>
> **1. The current theory is limited in the sense that the number of base concepts is no larger than the embedding dimension, since otherwise we don't have strict orthogonality. It would be interesting to allow near-orthogonality so that the embedding space can accommodate exponentially many base concepts.**
>
> Thank you for the interesting suggestion. We want to clarify a subtle point: in our framework, $k$ is the number of *concepts* (factors), not the number of *values*. Our Prop 2 shows $d \geq k$, and this bound is independent of $n$ (the number of values per concept). So we can have 10 concepts each with arbitrarily many values and only need $d \geq 10$. The theory already accommodates arbitrarily many values per concept.
>
> We think the "exponentially many base concepts" mentioned in the superposition literature are closer to feature directions or values, not independent compositional factors in our sense. A concept like "color" is one factor with many values, not many separate concepts. As such, only a single dimension could be used, in theory, to accommodate many different values.
>
> Also, orthogonality in our theory is a consequence of the desiderata. The $d \geq k$ bound follows from divisibility alone, without assuming orthogonality. That said, we believe that relaxing the desiderata, e.g. by considering approximate stability, would lead to relaxed orthogonality constraints as well.
>
>
> **2. The authors haven't explained why the current embedding models succeed or fail in terms of the pretraining datasets. How does the selection of dataset impact the geometry and generalization? Is there spurious correlation in the base concepts or imbalances in frequency of concepts?**
>
> Our work studies necessary conditions on the representation, not on what training data leads to them. The evaluation datasets we use (dSprites, MPI3D, PUG-Animal) are clean and balanced by construction, so spurious correlations and frequency imbalances are controlled for.
>
> Understanding how pretraining data composition affects the learned geometry is a separate and interesting direction. Recent works have started studying this [1, 2]; our goal was to characterize the endpoint: what representational geometry should models converge to as data and scale grow.
>
> [1] Kempf et al., "When and How Does CLIP Enable Domain and Compositional Generalization?", 2025
>
> [2] Mahajan et al., "Compositional Risk Minimization", 2024.
>
>
> **3. The rigor of Proposition 1 could be improved. It feels sloppy to say the algorithm is 'GD+CE'. If the authors want to use the connection to max-margin classifier, they need to be clear about the conditions.**
>
> We will explicitly connect the conditions of Soudry et al. (2018) to our setup: (1) linear separability holds because we assume the model is linearly compositional (Def 2), which by definition means a correct linear classifier exists; (2) we use the asymptotic convergence in direction result; (3) Will add a sentence or remark after Prop 1 making this explicit, and expand key acronyms.
>
> **4. I find the Stability condition quite strong. I'd interpret the conditions more as conceptual guidance than as verifiable assumptions.**
>
> We agree, stability is indeed an idealized condition and can be interpreted as conceptual guidance for what ideal behavior looks like.
>
> **5. It seems that the readout head is mostly restricted to linear readout. This significantly reduces the potential impact.**
>
> We focus on linear readouts because they are how these models are trained and used: CLIP and SigLIP use dot-product similarity between image and text embeddings (which is a linear readout). The linear head is part of the architecture, not just an evaluation choice. Extending to non-linear readouts is an interesting direction for future work.
>
> **6. I feel that the authors could have written the limitations part better.**
>
> We agree. In the revision we will more clearly distinguish between findings that are practically useful (e.g. $R^2$ as a diagnostic for compositional capability, the correlation with generalization accuracy) and findings that are theoretical and not yet fully validated in practice (e.g. exact stability).

---

> > ### Author Rebuttal · Reviewer_miQG · 2026-04-04
> >
> > I think the authors did a good job clarifying the nuances of their paper. I will maintain my score as my initial evaluation is already high.

---

### Official Review · Reviewer_XKhF · 2026-03-12

**Soundness:** 2
**Presentation:** 3
**Significance:** 2
**Originality:** 2
**Overall Recommendation:** 4
**Confidence:** 4

**Summary:**

This paper defines the necessary conditions for CLIP-like embedding models to achieve compositional generalization, categorized into dataset requirements, representation properties, and model constraints. The authors claim that the training set must satisfy validity rules and class conditions within the concept space, while the model must ensure divisibility, transferability, and stability. Furthermore, it is argued that linear compositional generalization is feasible when an encoder, linear head, and validity class satisfy specific cross-entropy or contrastive-based objectives. The authors provide empirical validation through whitening, orthogonality, and analysis of factor-specific dimensionalities.

**Compliance With Llm Reviewing Policy:**

Affirmed.

**Final Justification:**

My concerns have all been adequately addressed, and I have therefore increased my score.

**Key Questions For Authors:**

- Please address the points raised in the Weaknesses section.

**Limitations:**

The paper focuses strictly on the interpolation aspect of combinatorial generalization and does not address generalization in the context of generative models. (However, since combinatorial interpolation is not yet clearly defined in existing literature, I have not penalized the final score based on these limitations).

**Strengths And Weaknesses:**

### Strengths
- **(Originality)** The definition and analysis of necessary conditions at both the dataset and model levels for discriminative models to achieve combinatorial generalization is timely and highly relevant to the current state of representation learning.

- **(Significance)** By focusing on the practically impactful CLIP-like models, this research provides a useful framework that can be extended to future studies on CLIP-based model generalization.

---

### Weaknesses (Major Issue)
1. **Scope of Generalization**: The paper only defines cases for interpolation within combinatorial generalization. Without a definition of datasets or models capable of performing extrapolation, it is difficult to conclude that the proposed framework fully addresses "combinatorial generalization." [1].

2. **Injective Encoder Assumption**: How is the injectivity of the encoder verified? The paper assumes embedding models using linear classifiers; however, in such cases, embeddings of the same class tend to converge to nearly identical representations such as simplex ETF. This significantly reduces the likelihood of the encoder being injective.

3. **Justification for Linear Head ($h$)**: The authors state that linear heads are chosen because non-linear heads with injective encoders might rely on memorization. However, the causal relationship between non-linearity and memorization is not clearly explained or substantiated.

4. **Dependency on $|T|$ and $|C|$ within Theoretical Bounds**: In Section 5.1, the $R^2$ score results are likely sensitive to the size of the training support $|T|$. Specifically, $R^2$ values naturally tend to increase as $|T|$ grows, which necessitates a more rigorous sensitivity analysis across varying $|T|/|C|$ ratios.

    -  In Section 4.1 (lines 259–266), the authors state the data point requirement as $1+C \\leq |T| \\leq 1+2^{C-1}$. However, it is unlikely that generalization performance remains consistent across this entire interval.

    - The experiments in Section 5.2 utilize $|T| = 0.9n^k$ (a $9:1$ train-test split). This represents a standard high-data regime where generalization is relatively easy to achieve. To truly validate the theoretical "requirement," the authors must conduct experiments in more extreme, low-data case such as approaching the lower bound $|T| = 1+C$ mentioned in Section 4.1.

    - Empirical evidence showing how the $R^2$ score and generalization stability fluctuate as $|T|$ varies within the proposed theoretical bounds is essential. Without demonstrating that the model holds at the minimal requirement of $1+C$, the claim regarding data point requirements remains unsubstantiated.


5. **Inconsistency in Experimental Setup**: In Section 5.2, the train-test ratio is $9:1$ ($0.9n^k$), whereas Proposition 1 in Section 4.1 specifies $|T| = 2^{k-1}+1$. These conditions are inconsistent. If $0.9n^k$ was chosen to satisfy the theoretical bounds, the authors must also provide results for the minimum case ($1+C$). Performance and trends are likely to vary significantly with training set size.

6. The standard $x, y$-position count for dSprites is 32, yet only 10 are used here. If the dataset was downsampled or truncated, the authors should justify this choice or re-validate the results using the full dataset.

- **Note to Authors**: I am willing to increase my score if at least points 2, 4, 5, 6 are adequately addressed.

---

### Minor Issue

- **Alignment**: The order of figures and tables mentioned in the text does not match the actual document sequence.

- **Content Placement**: Experimental results mentioned in the main body should be moved from the Appendix to the main paper for better accessibility.

- **Notation**: Define all notations upon their first appearance (e.g., $U$).

---

- [1] Visual Representation Learning Does Not Generalize Strongly Within the Same Domain, Schott, L., Kugelgen, J.V., Trauble, F., Gehler, P., Russell, C.,
Bethge, M., Scholkopf, B., Locatello, F., and Brendel, W. International Conference on Learning Representations, 2022

---

> ### Author Rebuttal · Authors · 2026-03-30
>
> Thank you for the thoughtful review. We provide supplementary figures for the new experiments in the [rebuttal PDF](https://anonymous.4open.science/api/repo/anonrebuttal5314413-89BD/file/rebuttal.pdf?v=40554247).
>
> **1. The paper only defines cases for interpolation within combinatorial generalization. Without a definition of datasets or models capable of performing extrapolation...**
>
> Thank you for raising this. Our work studies compositional generalization: recognizing known concept values in novel combinations. Following the terminology of Schott et al. [1] (their Fig. 2), this corresponds to the "composition" setting, which is distinct from "interpolation" and "extrapolation" (generalizing to unseen values of continuous factors). We focus on composition because this is the setting relevant to CLIP-like models: the individual concept values (e.g. "cat", "red") are seen during training, but specific combinations may not be. We will clarify this distinction in the revision.
>
> **2-3. How is the injectivity of the encoder verified? Embeddings of the same class tend to converge to nearly identical representations such as simplex ETF. This significantly reduces the likelihood of the encoder being injective; The causal relationship between non-linearity and memorization is not clearly explained**
>
> We do not assume injectivity in our theoretical results. We mention it only once (Section 3.2) to motivate why we focus on linear heads. The main reason we consider linear heads is that the majority of embedding models in practice use them (CLIP, SigLIP). Injectivity is a side concern and is disconnected from memorization; we mentioned it to highlight that non-linear compositional models may exist by default due to potential injectivity of the encoder. We will make this distinction clearer in the revision.
>
> **4. The $R^2$ score results are likely sensitive to the size of the training support $S$. Specifically, $R^2$ values naturally tend to increase as $S$ grows, which necessitates a more rigorous sensitivity analysis across varying $S$ ratios.**
>
> We agree that understanding the relationship between $R^2$ and $|S|$ is important. Yes, we do expect $R^2$ to be positively correlated with $|S|$: recovering the factors gets easier as $|S|$ grows. We ran an experiment to show this over the models and datasets by retraining the models with different $|S|$ and measuring the $R^2$ score.
>
> Overall, the $R^2$ score is sensitive to the size of the training support, however, the major jump in $R^2$ happens when $|S|$ gets close to $< 0.1$ fraction of the whole dataset, showing that the $R^2$ score is not trivially inflated at high $|S|$ (see **Fig. 3 in the rebuttal PDF**).
>
> **5. The authors must conduct experiments in more extreme, low-data case such as approaching the lower bound $|T|=1+k$ mentioned in Section 4.1.**
>
> We agree that such experiments are important. We ran experiments across a wide range of train fractions (from 0.1% to 90%) over 3 seeds. The positive correlation between $R^2$ and compositional accuracy holds across all regimes, including extreme low-data settings (see **Fig. 1–2 in the rebuttal PDF**).
>
> **6a. In Section 5.2, the train-test ratio is $|S|/N=0.9$, whereas Proposition 1 specifies $|S|>N/2$. These conditions are inconsistent.**
>
> Our intention was to highlight that the proposition applies in two scenarios: when $|S|$ is close to $|T|$ and when $|S|$ is close to $|N|$. As such, the result should be understood as holding in low-data and high-data regimes and applies universally. We will clarify this in the text.
>
> **6b. The standard x-position count for dSprites is 32, yet only 10 are used here.**
>
> - The reason we use only 10 x and y positions was to not give higher importance to concepts that have more values (like positions), since we consider 10 orientations (to avoid symmetric cases) and 10 colors.
> - Also, having 32 positions seemed somewhat unnatural for CLIP tasks since describing 32 positions in text isn't common and probably unseen during pre-training.
> - Nonetheless, we re-ran all experiments with the full 32$\times$32 dataset (denoted as DSprites32) over 3 seeds.
> - All conclusions hold: the correlation between $R^2$ and compositional accuracy persists (**Fig. 1–2 in the rebuttal PDF**, rightmost column), $R^2$ sensitivity is consistent (**Fig. 3 in the rebuttal PDF**, rightmost panel), and the orthogonality results hold (**Fig. 4 in the rebuttal PDF**, rightmost panel).
>
> **7–9. Figure/table ordering, appendix results, notation.**
>
> Thank you for pointing these out. We will fix the figure/table ordering, move key experimental results from the appendix to the main paper, and ensure all notation is defined on first use.

---

> > ### Author Rebuttal · Reviewer_XKhF · 2026-04-03
> >
> > My concerns have all been adequately addressed, and I have therefore increased my score.

---

### Decision · Program_Chairs · 2026-04-30

**Decision:**

Accept (spotlight)

**Comment:**

The authors derive necessary structural conditions for compositional generalization in embedding models. Under standard training - linear classification head, cross-entropy loss, max-margin bias of gradient descent - three practically motivated desiderata (divisibility, transferability, stability) force representations to factorize linearly into additive per-concept components whose concepts are near-orthogonal. Divisibility alone, not assumed orthogonality, yields the elegant d ≥ k lower bound. The authors then survey CLIP and SigLIP families on dSprites, MPI3D, and PUG-Animal, and show that projected R², their measure of linear factorization, correlates positively with compositional accuracy across models and data regimes. All four reviewers recommend acceptance; three raised their scores after the rebuttal, and all four marked concerns fully resolved. Reviewers converge on several strengths: the paper bridges the empirical linear-representation hypothesis to a principled derivation (desiderata → linear factorization → orthogonality), proves an elegant d ≥ k result, explains dense technical material clearly, and sweeps photorealistic PUG-Animal alongside the synthetic benchmarks. The rebuttal added substantive evidence: error bars across seeds and concept pairs, an SVD residual analysis showing that unexplained variance spreads rather than clusters structurally, experiments across train fractions 0.001–0.9 confirming the R²/accuracy correlation even at extreme low-data regimes, and a full dSprites32 re-run confirming robustness to downsampling. Limitations remain, and the authors acknowledge them: the theory assumes exact stability and linear readouts; empirical R² reaches only 0.4–0.6, leaving substantial unexplained variance (consistent with partial satisfaction of a necessary condition, and with the SVD finding no structured residual); Proposition 2's d ≥ k result can look intuitive under orthogonality, so the authors should make the divisibility-only derivation explicit. I recommend Strong Accept. For camera-ready, the authors should clarify the two regimes of Proposition 1 (structured |T| = 1 + k vs. random |T| > N/2), sharpen their use of Soudry et al.'s max-margin result (overparametrization, linear separability, asymptotic direction), spell out Proposition 2's divisibility-only derivation, promote the new error bars, low-data experiments, dSprites32 re-runs, and SVD residuals into the main paper, add a proper limitations section, and fix figure/table ordering and typos.